# Ancestry and somatic profile indicate acral melanoma origin and prognosis

Patricia Basurto-Lozada[1,2], Martha Estefania Vázquez-Cruz[1], Christian Molina-Aguilar[1], Amanda Jiang[3,4,5], Dekker C. Deacon[3,4], Dennis Cerrato-Izaguirre[6], Irving Simonin-Wilmer[1], Fernanda G. Arriaga-González[1,7], Kenya L. Contreras-Ramírez[1], Emiliano Ferro-Rodríguez[1], Jamie Billington[7], Eric T. Dawson[8,9], J. Rene C. Wong-Ramirez[1,10], Johana Itzel Ramos-Galguera[1], Alethia Álvarez-Cano[11], Dorian Y. García-Ortega[12], O. Isaac García-Salinas[1,7], Alfredo Hidalgo-Miranda[13], Mireya Cisneros-Villanueva[13], Peter A. Johansson[14,15], Héctor Martínez-Said[12], Pilar Gallego-García[16], Mark J. Arends[17], Ingrid Ferreira[7], Mark Tullett[18], Rebeca Olvera-León[1,7], Louise van der Weyden[7], Martín del Castillo Velasco-Herrera[7], Rodrigo Roldán-Marín[19], Helena Vidaurri de la Cruz[20], Luis Alberto Tavares-de-la-Paz[21], Diego Hinojosa-Ugarte[22], Rachel L. Belote[23,24], D. Timothy Bishop[25], Marcos Díaz-Gay[16,26,27,28], Ludmil B. Alexandrov[26,27,28], Yesennia Sánchez-Pérez[6], Gino K. In[29], Richard M. White[30], Patrícia A. Possik[7,31], Robert L. Judson-Torres[3,4,5], David J. Adams[7] & Carla Daniela Robles-Espinoza[1,7 ✉]

Acral melanoma, which is not ultraviolet-associated, is the type of melanoma reported most commonly in several non-European-descent populations[1–3], including in Mexican people[4]. Latin American samples are substantially under-represented in global cancer genomics studies[5], which directly affects patients in these regions as it is known that cancer risk and incidence may be influenced by ancestry and environmental exposures[6–8]. To address this, we characterized the genome and transcriptome of 123 acral melanoma tumours from 92 Mexican patients—a population notable because of its genetic admixture[9]. Compared with other studies of melanoma, we found fewer mutations in classical driver genes such as *BRAF*, *NRAS* or *NF1*. Although most patients had predominantly Amerindian genetic ancestry, those with higher European ancestry had increased frequency of *BRAF* mutations. The tumours with activating *BRAF* mutations had a transcriptional profile more similar to cutaneous non-volar melanocytes, indicating that acral melanomas in these patients may arise from a distinct cell of origin compared with other tumours arising in these locations. Transcriptional profiling defined three expression clusters; these characteristics were associated with recurrence-free and overall survival. Our study enhances knowledge of this understudied disease and underscores the importance of including samples from diverse ancestries in cancer genomics studies.

Melanoma is classified into several clinicopathological subtypes on the basis of tumour site of presentation and histopathological features. Acral melanoma is an understudied melanoma subtype due to its low incidence globally, and because it represents a small proportion of melanoma cases in European-descent populations[2,10]; however, acral melanoma represents the vast majority of melanoma cases in some Latin American, African and Asian countries due to the lower incidences of ultraviolet-induced melanoma subtypes[11]. Furthermore, the causes of this type of disease are unknown, with patients managed in a similar way to ultraviolet-associated cutaneous melanoma. However, its site of presentation and genomic characteristics are vastly different[12].

Acral melanoma arises on the glabrous (non-haired) skin of soles, palms and in the nail unit (subungual location), and its genome differs substantially from other cutaneous melanoma subtypes[13]. In contrast to ultraviolet-linked subtypes, acral melanoma has a lower burden of single nucleotide variants (SNVs), a higher burden of structural variants and a low prevalence of mutational signatures SBS7a/b/c/d, which are associated with ultraviolet irradiation[14–18]. Genes that are mutated frequently in cutaneous melanoma such as *BRAF*, the *RAS* genes and *NF1*, are reported to be altered at a significantly lower frequency in acral melanoma. This, coupled with the comparatively lower number of studies of acral melanoma when compared with other cutaneous melanoma subtypes, has translated into limited available therapies for acral melanoma management.

It is known that cancer risk and incidence, as well as tumour genomic profiles, vary with ancestry and geographical location[6,19,20]. As most genomic studies on acral melanoma have been performed on patients of European or Asian ancestry, we considered it necessary to examine the genomics of this subtype of melanoma in Latin American people. Specifically, Latin American populations have been substantially

under-represented in cancer genomic studies, with only about 1% of all samples in cohorts such as the Pan-Cancer Analysis of Whole Genomes, The Cancer Genome Atlas (TCGA) and other repositories, and those contributing to cancer genome-wide association studies, being of Latin American origin[5,21,22]. Identification of differences in the genomic profile among populations can potentially aid the discovery of germline/inherited or environmental factors related to acral melanoma aetiology, as well as identify optimal therapeutic strategies for all patients.

In this study, we analysed 123 acral melanoma samples from 92 Mexican patients through genotyping, exome sequencing, SNV and insertion/deletion (indel) variant calling, copy number estimation and gene expression profiling, and examined the correlation of these molecular characteristics with clinical variables. We reveal a significant correlation between genetic ancestry and *BRAF* somatic mutations, as well as a distinct transcriptomic profile in tumours with *BRAF*-activating mutations compared with samples without activating mutations in *BRAF*. We also identify significant differences in recurrence-free and overall survival among patients with tumours with distinct gene expression profiles.

## Ancestry and clinical characteristics

A total of 123 uniformly ascertained samples from 92 patients from a large Mexican tertiary referral hospital were analysed in this study (Supplementary Table 1; Methods). Of these tumours, 89 were primaries, 27 were metastases, five were recurrences, one was a lesion in transit and one was unknown (Supplementary Table 1). Latin American genomes are generally a mixture of European, African and Amerindian ancestry. Of note, 90% of genotyped samples ($n = 80$) in this study had predominantly Amerindian ancestry (median 81%) (Supplementary Fig. 1 and Supplementary Table 2) with European and African ancestries contributing a median of 13.6% and 2.5%, respectively. The median age of the patients in this cohort was 60, with 59% of the patients being female. Most patients were stage III (American Joint Committee on Cancer, 8th edn)[23] at diagnosis, and the most common primary site was the foot—most frequently the sole. The median Breslow thickness was 4.0 mm and most primary tumours were ulcerated (68%) (Table 1). It should be noted that only four patients received immune checkpoint inhibitors or targeted therapy, due to lack of access.

## Sex and ancestry link to somatic profile

Considering all 123 samples, acral melanoma tumours showed a SNV+indel (hereafter referred to as tumour mutational burden (TMB)) mean of 0.95 mutations per megabase and a median of 0.87 mutations per megabase (range, 0–3.49 mutations per megabase). When including only one sample per patient, with primaries being selected preferentially, the most frequently mutated genes were *NRAS* (14% of samples; $Q$-value < $4.97 \times 10^{-10}$), *KIT* (14% of samples not counting deletions, as they are unlikely to be activating; $Q$-value = $4.97 \times 10^{-10}$), *BRAF* (13%, $Q$-value = $3.86 \times 10^{-7}$) and *NF1* (9%, $Q$-value = 0.0001) (Fig. 1a and Supplementary Table 3). Two of these samples had homozygous *NF1* deletions, in addition to a further two secondary samples from other patients (Extended Data Fig. 1a). For *BRAF*, all mutations except one were V600E, with one L597R (Fig. 1b). These genes, which represent known drivers, were identified as being under positive selection (Methods) and exhibit mutual exclusivity (only one patient has tumours with mutations in more than one of these genes), which reflects their functional redundancy in activating the MAPK pathway. Separate capillary sequencing of the *TERT* promoter in 76 samples belonging to 64 patients identified that six carry the −124 promoter mutation (9.3%) and two out of 59 patients for which the −146 position was amplified successfully carry a mutation in this position (3.4%) (Supplementary

### Table 1 | Clinical information for patients included in this study

| Number of patients | | 92 |
|---|---|---|
| **Age** | | |
| | Median | 60 |
| | Mean | 61.22 |
| | Minimum | 32 |
| | Maximum | 98 |
| **Sex** | | |
| | Female | 54 |
| | Male | 38 |
| **Stage** | | |
| | IV | 9 |
| | III | 47 |
| | II | 25 |
| | I | 10 |
| | In situ | 1 |
| **Ulceration** | | |
| | Yes | 61 |
| | No | 29 |
| | Not available | 2 |
| **Site** | | |
| | Feet | 76 |
| | Hands | 4 |
| | Subungual | 12 |
| **Breslow thickness** | | |
| | Median | 4 |
| | Mean | 6.61 |
| | Minimum | 0.45 |
| | Maximum | 50 |

Table 4). In total, we estimate that about 10.5% of patients have an activating *TERT* promoter mutation, which is similar to estimates in other studies[15,24]. All samples from all patients that had several samples sequenced and that could be assessed had a concordant *TERT* genotype, in agreement with an early emergence of this mutation during tumour evolution[24]. Other genes reported previously as mutated in other melanoma subtypes, as well as other cancer types are also mutated in this cohort, such as *TP53*, *HRAS* and *KRAS* (Fig. 1a and Extended Data Fig. 2). In summary, the 'classic' melanoma driver genes (*N/H/KRAS*, *BRAF* and *NF1*) are mutated in 40% of Mexican acral melanoma samples, with most of the samples in this cohort therefore being classified as 'triple wild-type' melanomas. Apart from the known *HRAS*, *SPRED1*, *TP53* and *KRAS* driver genes, we also find mutations in *PTPRJ*, *ATM*, *NF2* and *RDH5* (Extended Data Fig. 2a,b). Specifically, in those tumours without mutations in any of the abovementioned four driver genes (*BRAF*, *NRAS*, *KIT*, *NF1*, 'quadruple wild type' (QWT)), we find two tumours each from different patients with deleterious mutations in *ATM* and *RDH5* (Extended Data Fig. 2b). The mutations in these genes are all protein-changing and deleterious. All these genes have been linked previously to tumour suppressor activities in either acral or mucosal melanomas[25–30], as well as other cancer types, and may represent low-frequency drivers. We also observe a significantly higher proportion of women versus men carrying mutations in driver genes (two-tailed Fisher's test $P$ value = 0.003) (Supplementary Table 5). After adjusting for date of diagnosis, age at diagnosis, ancestry and tumour stage, the odds ratio of having a mutation in a driver gene in female patients (compared with men) was estimated to be 3.83 (95% confidence interval, 1.32, 11.03) (multivariate logistic regression, $P$ value = 0.013).

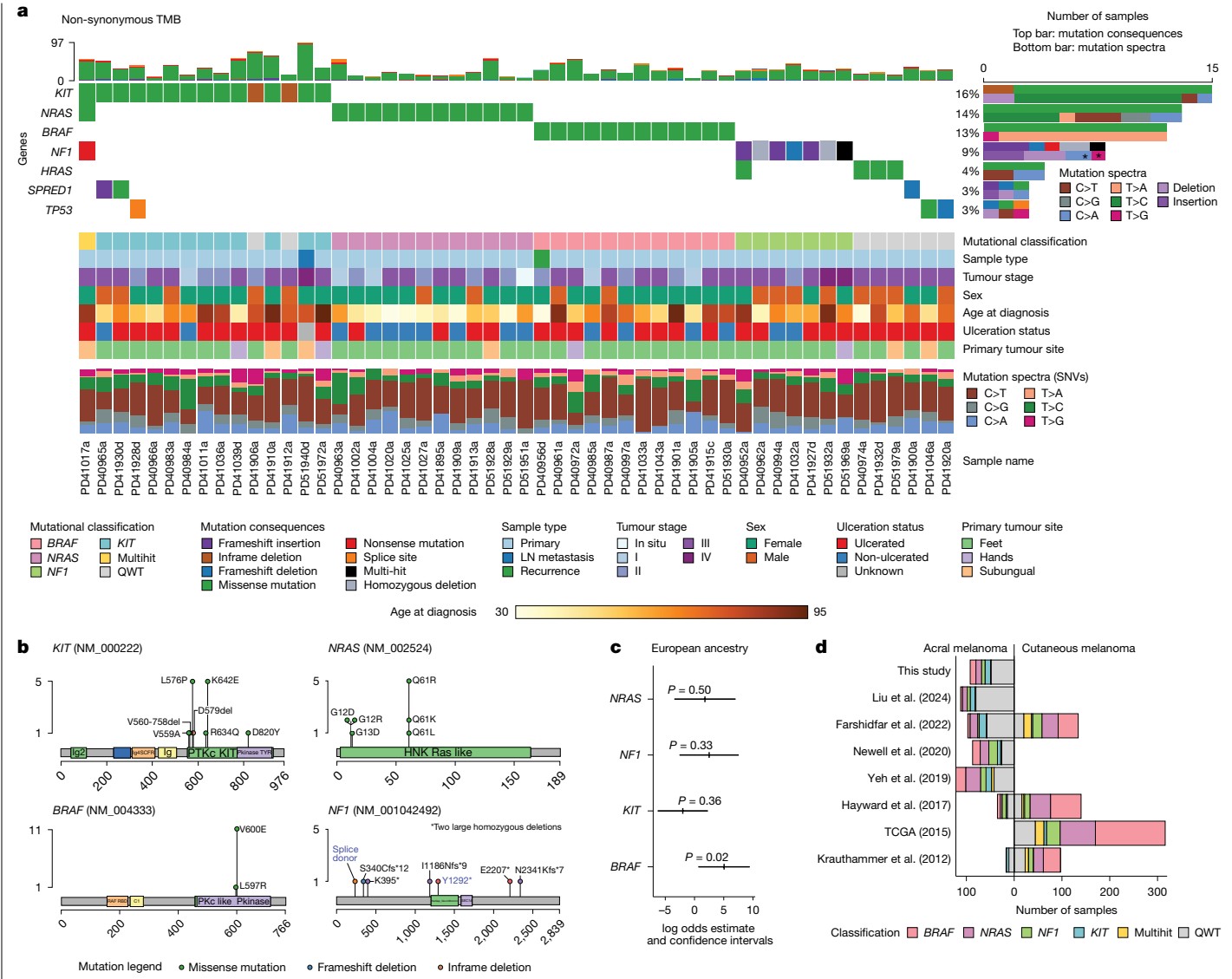

**Fig. 1 | Somatic landscape of acral melanoma in Mexican patients. a**, Oncoplot depicting the seven most mutated genes according to dNdScv and their status in the samples with mutations in these (52 samples out of 92, one per patient). Mutational classification, sample type, tumour stage, sex, age at diagnosis, ulceration status, tumour site and mutational spectra are shown by sample. In the mutational spectra plot, asterisks indicate that these mutations occurred in the same sample. LN metastasis, lymph node metastasis. **b**, Mutations found in *KIT*, *NRAS*, *BRAF* and *NF1*, which are the most significantly mutated genes. For

*NF1*, the two mutations in blue font occurred in the same sample. **c**, A logistic regression model controlling for age, sex and total TMB was fitted to predict the presence or absence of a mutation on the acral melanoma samples using the inferred ADMIXTURE cluster related to the European ancestry component. The log odds estimate and confidence intervals are depicted for the four driver genes. The estimate and its two-sided *P* value were obtained from the summary of the model. **d**, Barplot depicting the number and mutational classification of samples in different acral and cutaneous melanoma studies[14,15,18,31–34].

When examining the relationship between ancestry and somatic profile, we identified significantly higher odds (*P* value = 0.02) of carrying a *BRAF* somatic mutation with increasing European ancestry in a linear model controlling for age at diagnosis, sex and total TMB (Fig. 1c). Patients with mutations in *KIT* showed a tendency for higher Amerindian ancestry (Supplementary Fig. 2). We also found that patients with *NRAS* mutations were younger at diagnosis (median and mean age of diagnosis for patients with *NRAS* mutations was 49 years and 50.84 years versus 63 years and 62.9 years without, respectively), but this effect is probably mediated by ulceration status, as patients with *NRAS* mutations have a significantly lower rate of ulceration (two-tailed Fisher's exact test *P* value = 0.016).

Out of 22 patients for whom we sequenced at least two different samples (for example, a primary and a metastasis), 13 were classified as *NRAS/KIT/BRAF/NF1*-mutated. For *BRAF*, all four patients have

mutations across all samples and, for *NRAS*, three out of four patients have a *NRAS* mutation in both the primary and lymph node metastasis. These data indicate that these mutations appear as an early event in tumour evolution. For patients with *KIT* mutations, about half of the times the mutation is found in the primary and not in the metastasis. These data agree with those of Wang et al.[24], and indicate that metastases are seeded before the appearance of these mutations (Supplementary Table 6).

Collectively, these results are similar to those reported in other acral melanoma studies[14,15,18,31–34], with some important differences: first, the genetic composition of the patients in our study includes a high proportion of Native American ancestry, which is severely under-represented in already published cancer genomics studies and permits the identification of relationships of specific ancestries with somatic characteristics. In addition, the fraction of activating *BRAF* mutations is lower

than in the studies with predominantly European-descent patients, and more similar to those with Asian patients, probably due to the positive relationship between *BRAF* mutation and European ancestry (Fig. 1d).

## Somatic copy number landscapes

Somatic copy number alteration (SCNA) analysis across all samples showed a higher burden of amplifications than deletions (Fig. 2a). Examination of 47 samples, one per patient, that passed our stringent quality filtering for this type of analysis (Methods), showed that 18 regions were significantly amplified, and six regions were frequently deleted (Supplementary Table 7). About a quarter (11, 23%) of these 47 samples had whole genome duplication events (Supplementary Table 1). Potential driver genes in frequently amplified regions include *TERT* (43% of samples), *CRKL* (36%), *GAB2* (30%) and *CCND1* (28%) (Supplementary Tables 4 and 7–9). Regions that showed recurrent deletions contained genes such as *CDKN2A*, *CDKN2B*, *ATM* and *TP53*. Specifically, *CDKN2A* and *CDKN2B* had deletions in 59.6% of samples, with eight of them (out of 28) being homozygous deletions (Extended Data Fig. 1b; Methods), whereas *ATM* and *TP53* both had deletions in 38% and 34% of samples, respectively. No association was found between ancestry and any of the significantly altered CNA regions.

SCNA profiles varied depending on whether samples had mutations in driver genes or were QWT. Specifically, samples with mutations in driver genes ($n = 23$) showed preferential amplification of *NOTCH2* ($P$ value = 0.036, two-tailed exact Fisher test) and 1q21.3, containing several genes ($P$ value = 0.02), whereas *CCND1* ($P$ value = 0.049), and *ARF6* and *SOS2* (both in same amplification peak, $P$ value = 0.048) were preferentially amplified in QWT tumours ($n = 24$). The 8p12 region, containing genes such as *FGFR1* and *TACC1*, was found amplified only in five QWT samples, whereas several regions were found altered only in mutated tumours (Supplementary Tables 10–15 and Extended Data Figs. 3 and 4).

When stratifying samples by mutational status (considering *BRAF*-, *NRAS*-, *NF1*-, *KIT*-mutated and multi-hit, which included one sample with mutations in more than one of these drivers), we did not observe any significant differences in SCNA among groups (measured by global copy number alteration score (GCS); Methods) (Fig. 2b,c). Considering all samples, those with *NRAS*, *BRAF* and *NF1* mutations had the lowest median total TMB, whereas *KIT*-mutated and multi-hit tumours had the highest median total TMB (Supplementary Fig. 3). We found a significant correlation between GCS score and total TMB (Pearson's product-moment correlation coefficient = 0.72; $P$ value < 0.0001) (Fig. 2d). Tumours from the subungual region also had a higher median GCS score than those found on the feet (Fig. 2e).

## Mutational signature analyses

Single-base substitution mutational signature analysis across 116 samples that carried at least one SNV revealed previously reported COSMICv.3.4 signatures SBS1, SBS5 and SBS40a (Extended Data Fig. 5). Apart from clock-like signatures SBS1 and SBS5 (ref. 35), SBS40a was also prevalent across the cohort, contributing 53.38% of mutations to the total (Supplementary Fig. 4). SBS40a is of unknown origin but was identified originally in kidney cancer and is present in many cancer types[36]. Copy number signature analysis identified a number of previously reported signatures across different samples ($n = 60$ samples; Methods)[37,38]. CN1, which has been associated with a diploid state and CN9, which is potentially caused by local loss of heterozygosity on a diploid background, dominated the copy number landscape (Extended Data Fig. 5; Methods). As expected, signatures related to chromothripsis (CN7, CN8) were also found in several samples across the cohort. The number of indels in the samples was too low to add meaningful information (average, 2.52 indels per sample), so signature analysis for indels was not performed. We similarly did not find any significant associations between ancestry and any mutational signature.

## Similarity of *BRAF* acral to cutaneous melanomas

It has been postulated previously that *BRAF*-mutated acral melanomas might be more biologically similar to melanomas from non-acral sites than to other acral melanomas[18,39]; because of the observed correlation of European ancestry with *BRAF* mutation rate, we decided to investigate this hypothesis. We successfully extracted and sequenced RNA from 77 primary tumours from different patients (Supplementary Table 1; Methods). We then generated a gene signature-based score for identifying acral- versus cutaneous-derived melanomas. For this, we sourced a list of candidate genes from acral melanoma and cutaneous melanoma datasets (Supplementary Table 16; Methods) and identified 20 genes with high classification accuracy in a training cohort of ten primary acral melanomas (used to derive a v-mel score, or 'A' for acral) and ten primary cutaneous melanomas (used to derive a c-mel score, or 'C' for cutaneous) recruited at the University of Utah (Fig. 3a,b). We then obtained scores (v-mel/c-mel, or A:C) for samples in our dataset of acral melanomas, separating primary tumours with *BRAF*-activating mutations ($n = 10$) versus *BRAF*-wild-type ($n = 67$) tumours. We observed a difference between *BRAF*-activating and *BRAF*-wild-type tumours ($P$ value = 0.045), with *BRAF*-activating tumours having a score closer to cutaneous melanomas (Fig. 3c, left panel). We then replicated this analysis in an independent cohort of 63 acral melanomas from Newell et al.[15] (*BRAF*-activating $n = 10$, wild type $n = 53$), which further confirmed these results ($P$ value = 0.039) (Fig. 3c, right panel). Therefore, we explored the possibility that this difference could be due to downstream mutated *BRAF* signalling. First, we replicated this analysis in the TCGA cutaneous melanoma data, finding no significant differences among *BRAF* and non-*BRAF*-mutated samples (Extended Data Fig. 6a). We then examined datasets in which mutant *BRAF* was introduced into primary melanocytes in a doxycycline-inducible manner[40]. We found that the c-mel signature genes were not activated downstream of mutant *BRAF*, further indicating that the classifier does not simply reflect *BRAF*-driven transcriptional changes (Fig. 3d). Using a recently developed method for assessing gene signature similarity[41], we compared the c-mel gene signature from our classifier to a previously published set of genes directly activated by mutant *BRAF* in melanoma cells[42]. We found no significant correlation between these signatures (Extended Data Fig. 6b).

In conclusion, in these comparisons, *BRAF*-activating tumours expressed a more 'cutaneous melanoma-like' transcriptional program. This result indicates that *BRAF*-mutated melanomas that occur at acral sites are transcriptionally closer to non-acral cutaneous melanomas, a transcriptional program that is not explained by *BRAF* downstream signalling, and are associated with increasing European genetic ancestry.

## Gene expression and prognostic factors

We then applied a more stringent quality filter, including coverage and alignment features, to primary tumours in this collection with 44 samples remaining for further analyses (Supplementary Table 1; Methods). Consensus clustering of gene expression identified three sample groups, expressing different gene modules (Fig. 4a,b and Supplementary Table 17; Methods). Cluster 1 was characterized by an epidermal/immune profile module, with high expression of keratin genes (for example, *KRT1* and *KRT9*), cytokines (for example, *CXCL16*, *IL7*, *IL4R*, *IL1R*, *IL15RA* and *CXCL14*) and processes such as epidermis development, cell–cell junction organization and wound healing (Fig. 4b, Extended Data Fig. 7 and Supplementary Table 18). Cluster 2 expressed higher levels of a mitotic/proliferative-related signature, with high expression of genes such as *MITF* and *TYR*, and processes such as chromosome segregation, nuclear division and mitochondrial translation (Fig. 4b, Extended Data Fig. 8 and Supplementary Table 19). Cluster 3 showed expression of a gene module characterized by respiration and oxidative phosphorylation-related genes (Fig. 4b, Extended Data Fig. 9 and Supplementary Table 20). Cluster 1 was associated with better prognostic

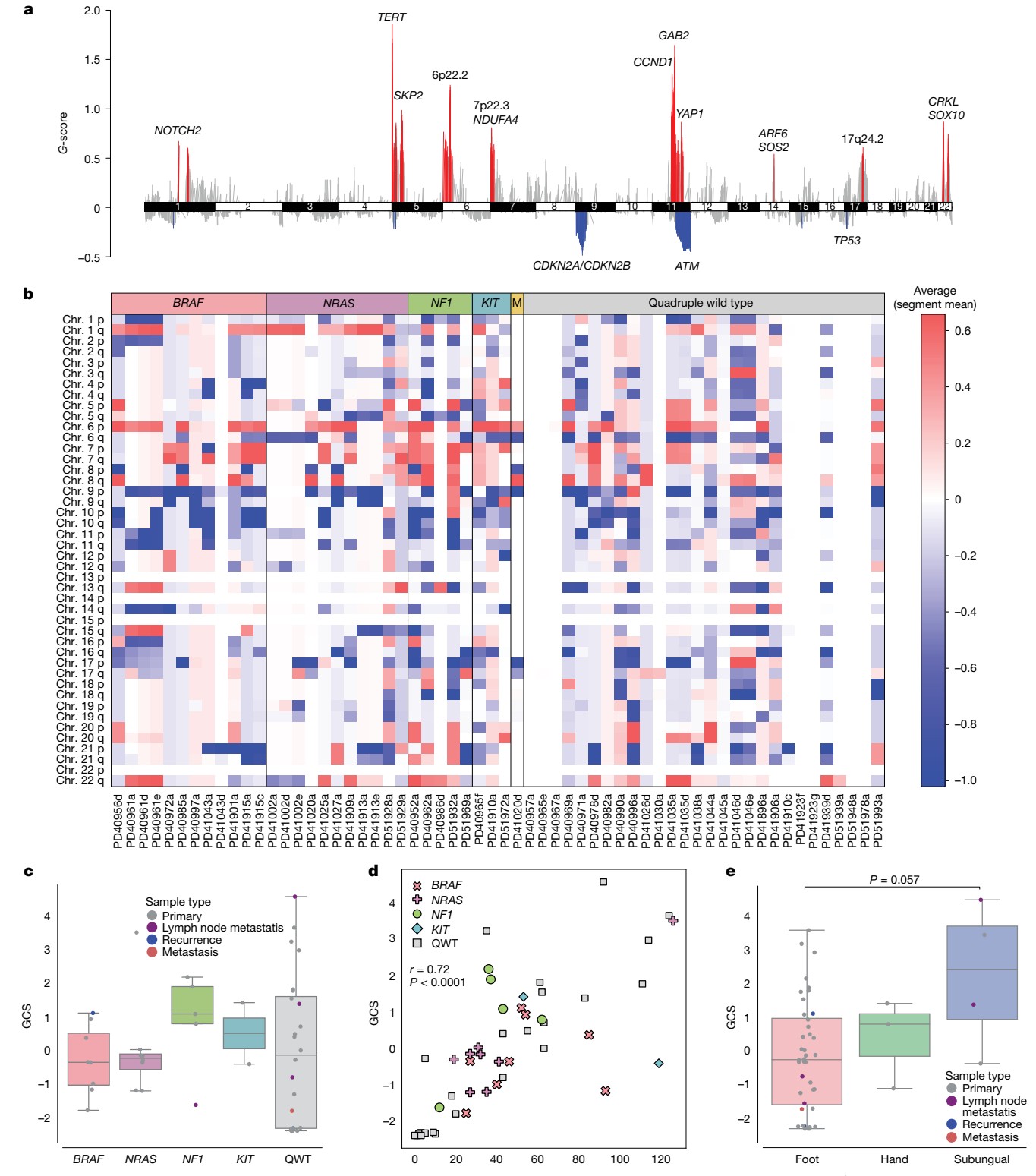

**Fig. 2 | DNA copy number landscape of acral melanoma and molecular and clinical correlates in Mexican patients. a**, Regions of amplification (red) and deletion (blue) in 47 acral melanoma samples, one per patient, as identified by GISTIC2. Known drivers, or the chromosomal regions, are shown. This analysis shows alterations with respect to the normal sample, that is, with respect to a ploidy of two. **b**, Heatmap showing regions of amplification (red) and deletion (blue) by sample and chromosomal arm in all 60 samples classified into genomic subgroups. This analysis shows alterations with respect to the estimated tumour sample ploidy. **c**, Box plot of GCS of 47 samples, one per patient, classified by genomic subgroup. **d**, Scatter plot of total TMB (referring to total number of

mutations, *x* axis) and GCS (*y* axis) for 47 samples, one per patient. Dots represent samples, coloured by genomic subtype. Pearson's product-moment correlation coefficient and associated two-tailed *P* value (*P* = 8.5 × 10⁻¹⁰) is shown. **e**, Box plot of GCS of 47 samples, one per patient, classified by tumour site. GCS scores are calculated with respect to tumour sample ploidy. For box plots, the central line within each box represents the median value, the box boundaries represent the interquartile range (IQR), and the whiskers extend to the lowest or highest data point still within 1.5× IQR. Statistical significance was assessed using two-sided Wilcoxon−Mann−Whitney tests.

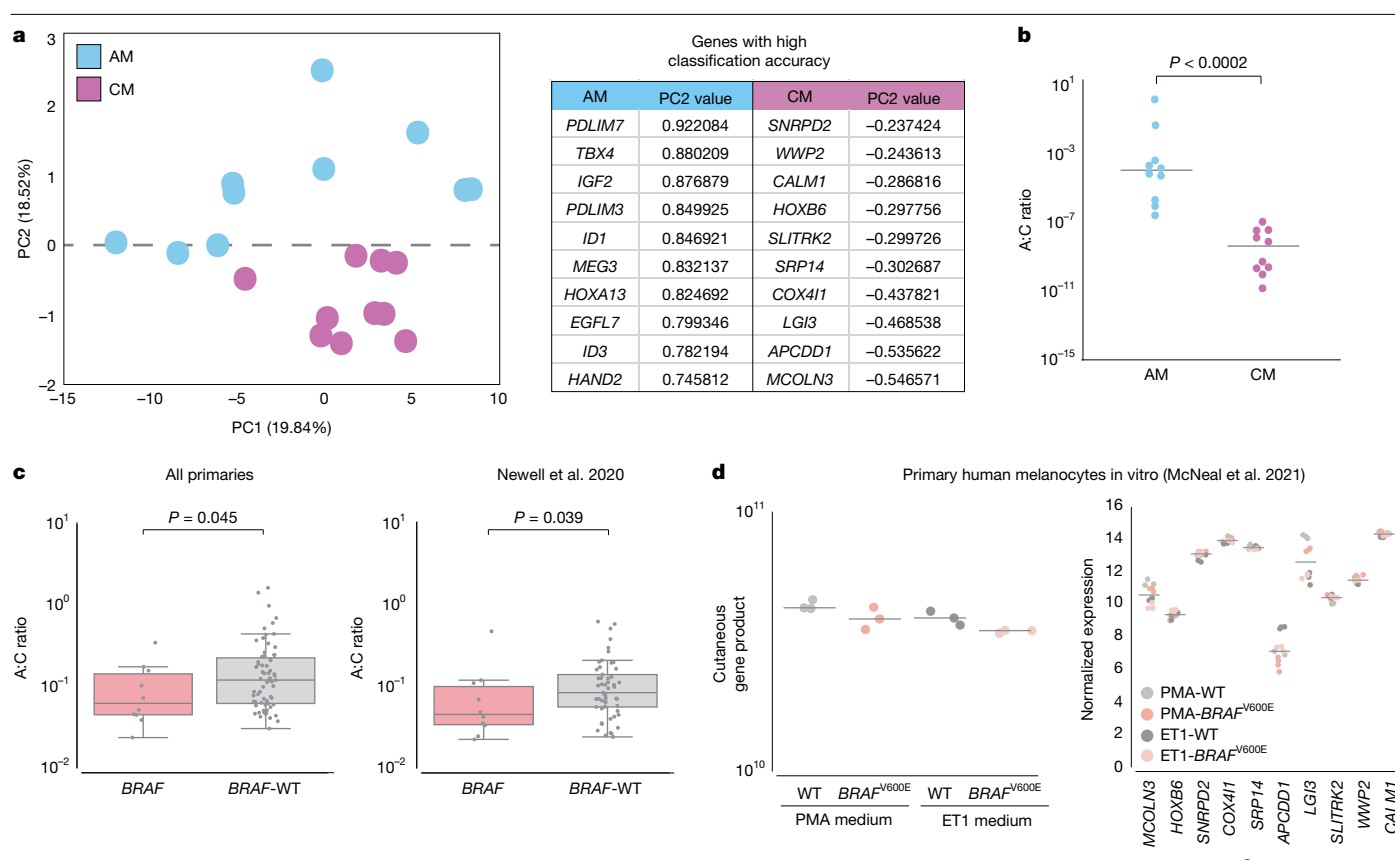

**Fig. 3 | Comparisons of the transcriptional profile of *BRAF*-activating and *BRAF*-wild-type acral melanoma tumours.** **a**, Elucidation of genes used to classify acral melanoma (AM) versus cutaneous melanoma (CM) samples. Left, principal component analysis (PCA) of acral melanoma (blue) and cutaneous melanoma (purple) samples. Right, loadings on PC2 were used to identify the top differentially expressed genes contributing to the variance between acral melanomas and cutaneous melanomas. **b**, Scatter plot showing the distribution of the acral:cutaneous (A:C) gene expression ratios between test acral and cutaneous melanoma samples. Acral melanoma samples (*n* = 10) are represented by blue dots and cutaneous melanoma samples (*n* = 10) are represented by purple dots (*P* = 0.00018, two-sided Wilcoxon–Mann–Whitney test). **c**, Left, comparison of A:C gene expression ratio in acral melanoma samples with different mutation status. Box and whiskers plot comparing two groups: *BRAF*-wild-type (*BRAF*-WT; *n* = 67) and *BRAF*-activating mutated tumours (*n* = 10). Right, comparison of A:C gene expression ratio in acral melanoma

clinical characteristics, such as a smaller proportion of ulcerated samples (53% versus 93% in cluster 2 and 57% in cluster 3), a tendency for earlier clinical stages and lower mitotic rates (Mann–Whitney *U* cluster 1 versus cluster 2 *P*-value: 0.041, cluster 1 versus cluster 3 *P*-value: 0.052) (Fig. 4c). Deconvolution of gene expression profiles also indicated differences in immune cell infiltration composition, with cluster 1 having a higher proportion of CD4⁺ T cells and cancer-associated fibroblasts (CAFs) and cluster 2 having a higher proportion of B cell infiltration (Fig. 4d–f, Extended Data Fig. 10 and Supplementary Table 21).

## Genomic profiles and survival outcomes

Next, we evaluated whether the genomic and transcriptomic characteristics had any impact on patient overall or recurrence-free survival. We included in the analysis those participants whose primary could be analysed (*n* = 85; Methods). The mean time between diagnosis and recruitment was 2.01 years, including 21 participants recruited within 6 months; the range was from a few days to more than 10 years. Among these participants, 12 primary tumours had an *NRAS* mutation, 11 had

samples with *BRAF*-activating mutations (*n* = 10) and *BRAF*-WT tumours (*n* = 53) from Newell et al.[15]. The central line within each box represents the median value, the box boundaries represent the IQR and the whiskers extend to the lowest or highest data point still within 1.5× IQR. Individual data points are plotted as dots. Statistical significance was assessed using individual one-sided Wilcoxon–Mann–Whitney tests. **d**, Left, comparison of the product of the cutaneous genes in normal human melanocytes. Melanocytes were cultured in PMA- or ET1-containing medium with or without doxycycline-induced *BRAF*^V600E expression. Each dot is an individual biological replicate (*n* = 3) with horizontal lines indicating median values. Right, relative expression levels of cutaneous genes across individual normal human melanocytes. Melanocytes were cultured in PMA- or ET1-containing medium with or without doxycycline-induced *BRAF*^V600E expression. Each point represents a biological replicate (*n* = 3 per condition) with horizontal lines indicating median values. Expression data for **d** are derived from McNeal et al.[40].

a mutation in *KIT*, 11 had a *BRAF* mutation, seven had *NF1* mutations, one had multiple hits and 43 were classified as QWT.

Carrying any driver mutation was not associated with age at diagnosis or tumour stage (data not shown). Having a tumour with a driver mutation was, however, associated with a reported recurrence, with 66.7% of mutated tumours having a recurrence as compared with 37.2% of QWT tumours (Pearson two-tailed Chi-squared test *P* value = 0.007). After adjusting for date of diagnosis, sex, age at diagnosis, ancestry and tumour stage (*n* = 73, primaries with ancestry information available), the odds ratio for a mutated tumour having a recurrence compared with QWT tumours was 5.31 (95% confidence interval, 1.56, 18.12), (multivariate logistic regression, *P* value = 0.008) (Fig. 5a and Supplementary Table 22). Notably, among the mutated tumours, for each different gene, tumour recurrence was increased over QWT tumours (Fig. 5a), most notably for *NF1*, where all seven of the mutated tumours recurred.

Overall, 44 of the tumours could be classified transcriptomically into one of the three clusters. There was no association between tumour driver mutation and transcriptomic cluster (data not shown). There was, however, evidence of differences in recurrence frequency by

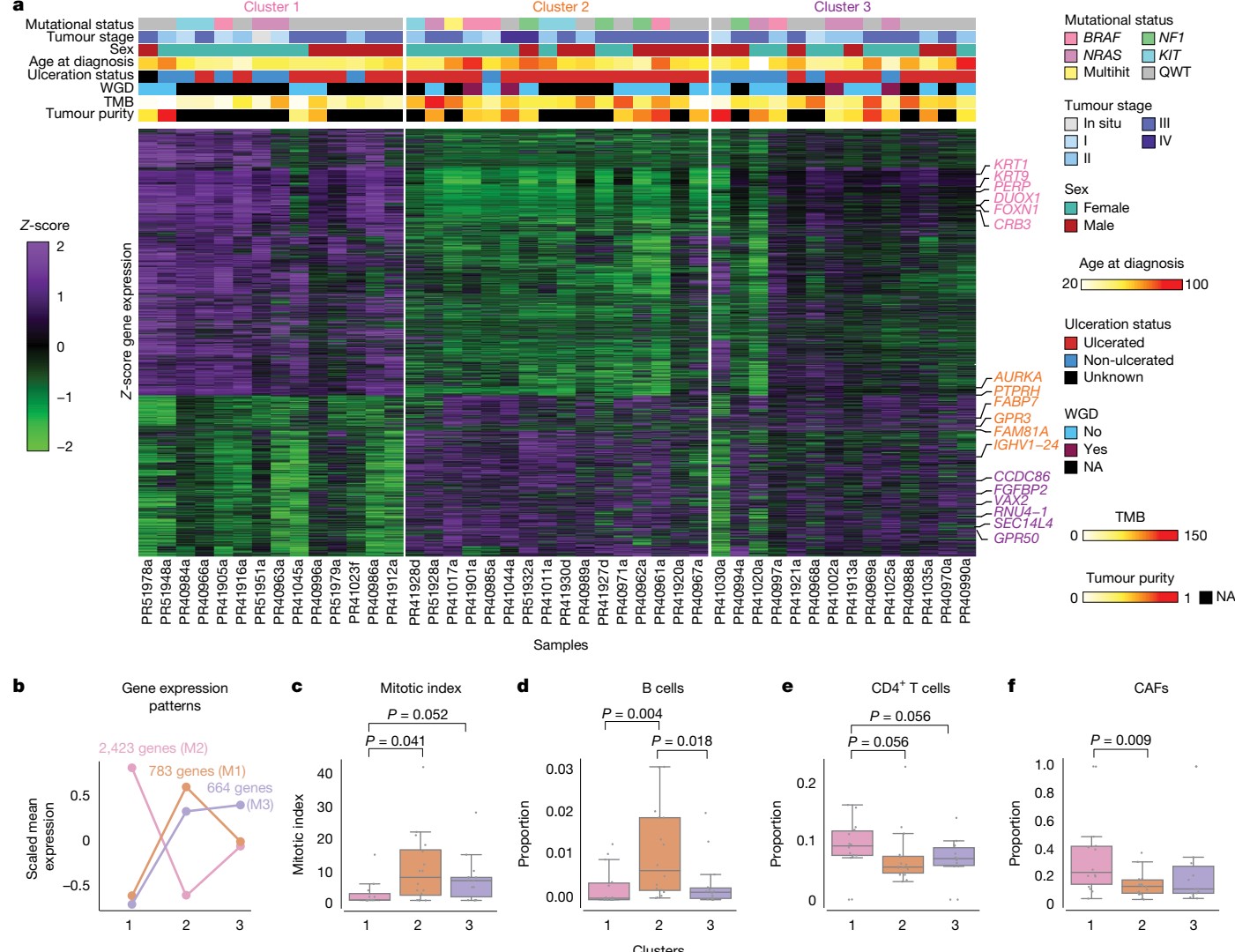

**Fig. 4 | Unsupervised gene expression clustering of primary acral melanoma samples from Mexican patients identifies three main groups. a**, Gene expression heatmap showing the 3,870 genes identified as differentially expressed among sample clusters; *x* axis, samples; *y* axis, genes. Mutational status and clinical covariates by sample are shown above the heatmap. WGD, whole genome duplication. **b**, Scaled mean expression patterns per cluster for the three gene modules defining each cluster. The name of the gene module is indicated after the number of genes. **c**, Box plot of mitotic index (*y* axis) per sample classified by transcriptional cluster. **d**, Box plot of B cell proportion (*y* axis), as calculated by deconvolution, per sample classified by transcriptional cluster. **e**, Box plot of CD4+ T cell proportion (*y* axis), as calculated by deconvolution, per sample classified by transcriptional cluster. **f**, Box plot of CAFs (*y* axis), as calculated by deconvolution, per sample classified by transcriptional cluster. The central line within each box represents the median value, the box boundaries represent the IQR and the whiskers extend to the lowest or highest data point still within 1.5× IQR. Individual data points are plotted as dots. For **c**–**f**, two-sided Wilcoxon–Mann–Whitney paired tests were performed. For **c**–**f**, 14 samples are included in cluster 1, 16 samples in cluster 2 and 14 in cluster 3.

cluster, with 35.7% of cluster 1 tumours, 81.2% of cluster 2 tumours and 57.1% of cluster 3 tumours having a recurrence (Fisher's exact test, *P* value = 0.04 for homogeneity). Logistic regression adjusting for age at diagnosis, sex, diagnosis date and stage at presentation showed that those tumours from cluster 2 had a higher rate of recurrence as compared with cluster 1 (odds ratio = 6.68; 95% confidence interval, 0.97, 46.27), multivariate logistic regression, *P* value = 0.054; Supplementary Table 23), whereas cluster 3 had intermediate rates of recurrence (Fig. 5b).

Fifteen participants (17.6%) died during the study period; 9% of participants with QWT tumours and 26.2% of participants with tumours with driver mutations died (*P* value = 0.042 for homogeneity). Log rank analysis of time to death from diagnosis showed a tendency for an increased risk of death among those with any mutation (*P* value = 0.077) (Supplementary Table 24), whereas similar analysis by specific mutation showed more extreme significance (*P* value = 0.0006)

(Supplementary Table 25). Cox proportional hazards analysis adjusting for age, sex, tumour stage and ancestry indicated participants whose tumour had any mutation in a driver gene had a tendency for an increased risk of death (hazard ratio = 3.19, 95% confidence interval, 0.8, 12.74; *P* value = 0.1), although this was not conventionally significant (Supplementary Table 26) (Fig. 5c).

Finally, survival analysis based on the 44 tumours with transcriptomic classification showed significant variation between the clusters, with 0% of cluster 1, 43.7% of cluster 2 and 21.4% of cluster 3 having died (log-rank *P* value = 0.011, Pearson two-tailed Chi-squared *P* value = 0.017), again in keeping with the analysis of recurrence, a known main risk factor for survival (Fig. 5d and Supplementary Table 27). Cox proportional hazards analysis adjusting for age at diagnosis, sex, stage at presentation and ancestry did not provide significant evidence of differences between the clusters in terms of mortality rates, in keeping with the limited sample size.

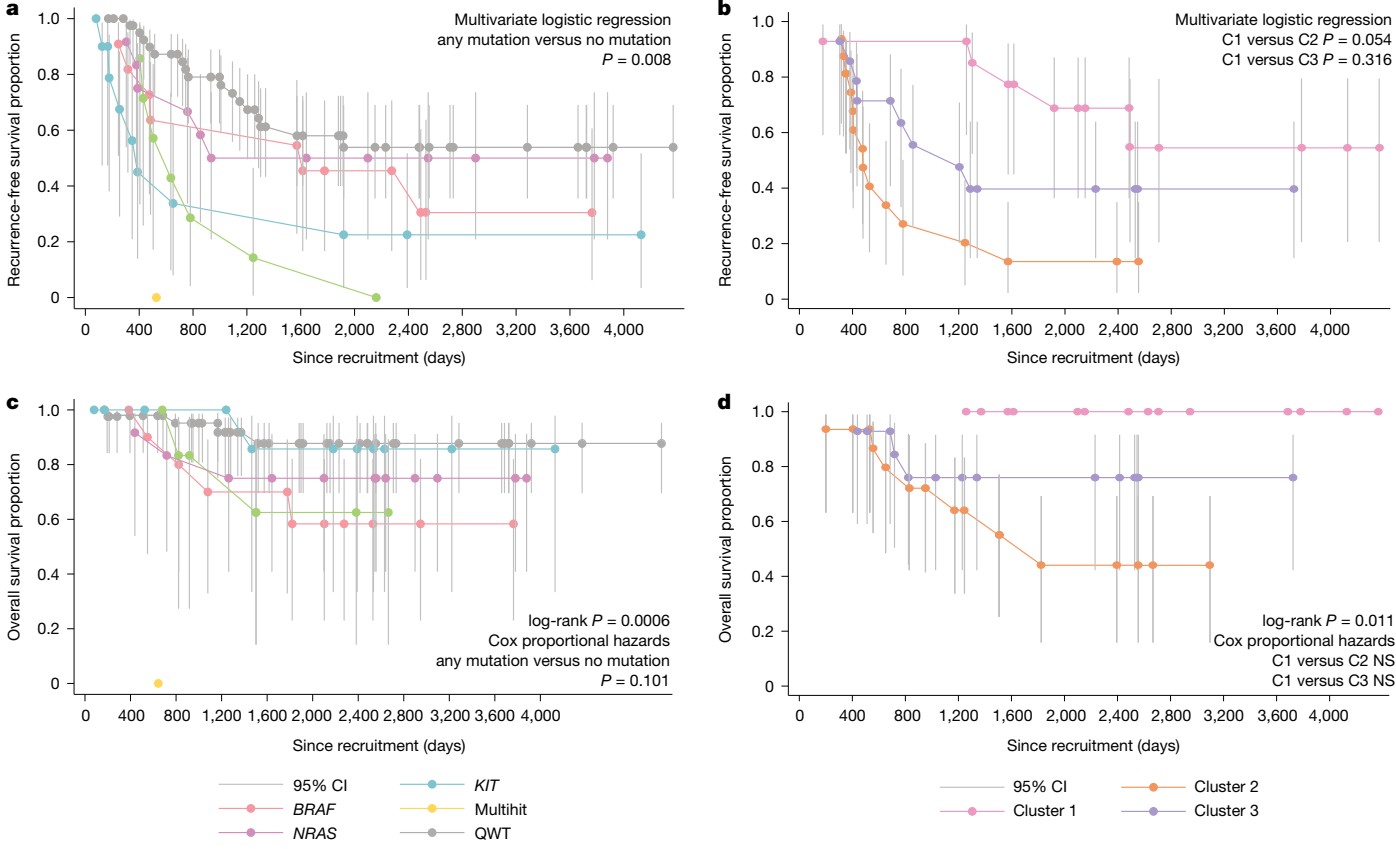

**Fig. 5 | Kaplan–Meier plots of overall and recurrence-free survival for patients by tumour mutational and transcriptional status. a,** Recurrence-free survival of patients ($n = 85$, all those participants whose primary could be analysed) with and without driver mutations, depicted by each category of the mutational classification. **b,** Recurrence-free survival for patients with tumours in each of the three transcriptional clusters ($n = 44$). **c,** Overall survival of patients with and without driver mutations, depicted by each category of the mutational classification ($n = 85$). **d,** Overall survival for patients with tumours in each of the three transcriptional clusters ($n = 44$). Each panel indicates the crude survival curves as indicated. Parametric analyses are derived from the same sample adjusted for the relevant covariates. The centre of each error bar is the estimated cumulative survival (recurrence-free or alive proportion) to that time point, and the bars represent its 95% confidence interval (CI). All reported $P$ values are two-sided. NS, not significant.

## Discussion

In this study, we report the analysis of the somatic and transcriptomic profile of 123 acral melanoma samples from Mexican patients – one of the largest cohorts reported for this type of cancer. In our view, this study helps address several research gaps: (1) the underrepresentation of samples of Latin American ancestry in cancer sample repositories[5]: as shown previously, genetic ancestry and environment influence the somatic profile of tumours, with potential impacts on patient management and treatment[6,19,20]; (2) the relative lack of studies of acral melanoma, when compared with other types of the disease, as this type of melanoma constitutes most reported cases in some non-European populations[11], and (3) the relative paucity of genomic studies performed and directed from low- and middle-income countries such as Mexico.

Most patients in this study had predominantly Amerindian genetic ancestry, which allowed us to perform an analysis of genetic ancestry correlates with somatic mutation profile. We identified a positive correlation between European ancestry and *BRAF* mutation rate (Fig. 1c). A possible link between European ancestry and *BRAF^V600E* mutation had been described previously[18], and this study provides further confirmatory evidence. Other similar correlations have been described recently for other types of cancer, such as a positive relationship between Native American ancestry and *EGFR* mutation rate in lung cancer[20], and an increased rate of somatic *FBXW7* in African patients compared with European patients[6]. In accordance with this observation, other cohorts of acral melanoma, which studied patients with predominantly European ancestry, have a higher *BRAF* mutation rate than that in this study (for example, 23% in Australian patients with predominantly European ancestry[15]). These observations should provide the basis for future studies exploring the relationships between ancestry and somatic mutation rate.

We were intrigued to discover that acral melanomas with *BRAF*-activating mutations exhibit a more 'cutaneous melanoma-like' transcriptome than other genetic subtypes of acral melanoma. One possible explanation is that this gene signature is uniquely downstream of a *BRAF* missense mutation. However, in further analyses, we do not see evidence for this explanation (Fig. 3c,d and Extended Data Fig. 6). An alternative explanation involves the distinct origins of acral melanomas with *BRAF*-activating mutations compared with other acral melanomas. In our previous work[43], we identified distinct subclasses of human epidermal melanocytes: a common type enriched in limbs (c-type) and a rare type enriched in volar regions (v-type). We observed that most acral melanomas generally retained a transcriptional signature such as v-type melanocytes, whereas a subset seemed more akin to c-type melanocytes[43]. The current work indicates that these tumours are more likely to belong to the *BRAF*-activating genetic subtype, indicating that a subset of volar melanomas might be classified more accurately by cell of origin and/or genetic profile as non-acral cutaneous melanoma, rather than bona fide acral melanomas. It is important to clarify that the hypothesis that acral melanomas may arise from different cells of origin is not based solely on this study but is also supported by previous work. Our previous research has demonstrated transcriptional diversity among melanocytes in different anatomic

locations, including distinct populations of epidermal melanocytes in the palms and soles[43]. Furthermore, our zebrafish model studies have shown that acral melanoma-associated drivers preferentially (although not exclusively) induce tumours in the limbs (fins), whereas cutaneous melanoma-associated *BRAF* mutations lead preferentially (but not exclusively) to tumours in the trunk[44]. Furthermore, we have demonstrated that *BRAF* mutations selectively drive hyperproliferation in less-pigmented primary human melanocytes[40]. Therefore, although our additional analyses do not strongly support an oncogene signature as the explanation for the differences in transcriptional scores, thus favouring the cell-of-origin hypothesis, it is possible that, in some cases, these two phenomena could be intertwined. For example, recent data have shown that some acral melanomas harbouring amplifications of the *CRKL* oncogene depend on *HOX13* positional identity programs already present in the cell of origin, indicating that oncogenes and cell-of-origin programs can synergize[44]. Future studies could explore the diagnosis of cutaneous melanoma as acral versus non-acral based on molecular signatures rather than solely on anatomic location. The fact that *BRAF*-mutated tumours occur less frequently on patients of non-European ancestry highlights the need to study a diverse set of samples to maximize clinical benefit to all patients.

Patients with cluster 1 tumours showed better prognosis than other patients, which is not surprising given their associated clinical characteristics (lower Breslow thickness, a tendency for earlier stages at diagnosis and lower mitotic indexes). However, what may be surprising is the gene expression profile characteristic of this cluster. More CAFs and CD4$^+$ T cells were found by deconvolution to be associated to cluster 1, signatures commonly associated with immunosuppression. A possible explanation is that early-stage tumours are associated with immunosuppressive microenvironments—a balance which, in later tumours, may have been tilted in favour of tumour cell growth. Another potential explanation may involve the recently described roles of CAFs in immunostimulation[45]. Patients with cluster 2 tumours, with a proliferative signature and expression of pigmentation genes showed the worst survival. It has been observed previously in a zebrafish model and in TCGA samples that a pigmentation signature also predicts worse survival[46] and, in a recent report by Liu and collaborators[31], acral melanoma tumours with a proliferative signature also were associated with worse survival than other tumours. This study both extends and replicates these findings in acral melanoma.

This study has several limitations, regarding the nature of the samples (formalin-fixed paraffin-embedded (FFPE)), and the stringent mutation and copy calling methodology, selected to mitigate artefacts at the cost of sensitivity. This study was also done using whole-exome data, which limits our ability to call mutations in non-coding regions of the genome. There are other challenges of setting up such a study. For instance, the fact that the year of diagnosis preceded the date of recruitment by up to 10 years means that somatic mutations associated with higher mortality rates would be under-represented among those recruited, whereas survival is probably extended over a similarly sized cohort of prospectively recruited cases. To assess the impact of any biases on our interpretation of the impact of mutations, we performed an analysis with only those tumours diagnosed after mid-2016, that is, those closest to the time of recruitment (Methods), obtaining similar results.

All tumours included in this study were confirmed as melanomas arising from glabrous skin. Our data indicate that tumours harbouring *BRAF* mutations may constitute a distinct subtype, sharing characteristics with superficial spreading and acral melanoma. These findings would have implications for patient selection in clinical trials evaluating new therapies for acral melanoma. Overall, we were able to identify new associations of the germline and somatic profile in acral melanoma, genomic-clinical correlates of overall and recurrence-free survival, as well as transcriptional differences in *BRAF*-mutated acral melanomas. This study shows the value of studying diverse populations, allowing us to uncover previously unreported relationships and better understand tumour evolution.

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

[1]Laboratorio Internacional de Investigación sobre el Genoma Humano, Universidad Nacional Autónoma de México, Santiago de Querétaro, Mexico. [2]Posgrado en Ciencias Biológicas, Universidad Nacional Autónoma de México, Mexico City, Mexico. [3]Huntsman Cancer Institute, University of Utah Health Sciences Center, Salt Lake City, UT, USA. [4]Department of Dermatology, University of Utah, Salt Lake City, UT, USA. [5]Department of Oncological Sciences, University of Utah, Salt Lake City, UT, USA. [6]Subdirección de Investigación Básica, Instituto Nacional de Cancerología (INCan), Mexico City, Mexico. [7]Wellcome Sanger Institute, Hinxton, UK. [8]EarthFrame Corporation, Bryan, TX, USA. [9]Phileal LLC, Bryan, TX, USA. [10]Research Programs Unit, Research Program in Systems Oncology, Faculty of Medicine, University of Helsinki, Helsinki, Finland. [11]Departamento de Cirugía Oncológica, Hospital NOVA, Universidad de Monterrey UDEM, Monterrey, Mexico. [12]Surgical Oncology, Skin, Soft Tissue and Bone Tumors Department, National Cancer Institute, Mexico City, Mexico. [13]Laboratorio de Genómica del Cáncer, Instituto Nacional de Medicina Genómica (INMEGEN), Mexico City, Mexico. [14]QIMR Berghofer, Brisbane, Queensland, Australia. [15]University of Queensland, Brisbane, Queensland, Australia. [16]Digital Genomics Group, Cancer Genomics Program, Spanish National Cancer Research Center (CNIO), Madrid, Spain. [17]Edinburgh Pathology, Cancer Research UK Scotland Centre, Institute of Genetics and Cancer, University of Edinburgh, Edinburgh, UK. [18]Department of Histopathology, University Hospitals Sussex, St Richard Hospital, Chichester, UK. [19]Dermato-Oncology Clinic, Research Division, Faculty of Medicine, Universidad Nacional Autónoma de México, Mexico City, Mexico. [20]Pediatric Dermatology Service, Hospital General de México Dr. Eduardo Liceaga, Ministry of Health, Mexico City, Mexico. [21]Surgical Oncology, Bajio Regional High Specialty Hospital, Leon, Mexico. [22]Division of Surgery, Instituto Nacional de Ciencias Médicas y Nutrición Salvador Zubirán, Mexico City, Mexico. [23]Department of Molecular Genetics, The Ohio State University, Columbus, OH, USA. [24]Department of Dermatology, The Ohio State University, Columbus, OH, USA. [25]Leeds Institute of Medical Research, University of Leeds, Leeds, UK. [26]Department of Cellular and Molecular Medicine, University of California San Diego, La Jolla, CA, USA. [27]Department of Bioengineering, University of California San Diego, La Jolla, CA, USA. [28]Moores Cancer Center, University of California San Diego, La Jolla, CA, USA. [29]Division of Oncology, Keck School of Medicine, Norris Comprehensive Cancer Center, University of Southern California, Los Angeles, CA, USA. [30]Nuffield Department of Medicine, Ludwig Institute for Cancer Research, University of Oxford, Oxford, UK. [31]Division of Basic and Experimental Research, Brazilian National Cancer Institute, Rio de Janeiro, Brazil. ✉e-mail: drobles@liigh.unam.mx

## Methods

A flowchart describing the analyses, steps and number of samples used in each individual section can be found in Supplementary Fig. 5.

### Patient recruitment and sample collection

The protocol for sample collection was approved by the Mexican National Cancer Institute's (Instituto Nacional de Cancerología, INCan, México) Ethics and Research committees (017/041/PBI;CEI/1209/17) and the United Kingdom's National Health Service (18/EE/00076). Patient samples collected for the Utah cohort analysis were derived as described previously[47].

Recruitment of patients and sample collection took place from 2017 to 2019. Patients attending follow-up appointments at INCan who had previously been diagnosed with acral melanoma were offered the chance to participate in this study and, upon signing a written consent form, were asked to provide access to a FFPE sample of their tumour tissue that had been kept at the INCan tumour bank, as well as a saliva or normal adjacent tissue sample. Patients provided samples and their clinical data in Excel format with written informed consent. FFPE samples underwent inspection by a medical pathologist to establish whether sufficient tumour tissue was available for exome sequencing. Saliva samples were collected using the oragene DNA kit (DNAGenotek, catalogue no. OG-500).

### DNA and RNA extraction

DNA extraction from all saliva samples was performed at the International Laboratory for Human Genome Research from the National Autonomous University of México (LIIGH-UNAM) using the reagent prepITL2P (DNAGenotek, catalogue no. PT-L2P) and the All-Prep DNA/RNA/miRNA Universal Kit (Qiagen, catalogue no. 80224). DNA and RNA extraction from FFPE samples was performed at the Wellcome Sanger Institute (UK) using the All-prep DNA/RNA FFPE Qiagen kit. Samples with >0.1 ng μl⁻¹ were sequenced through the Sanger Institute's standard pipeline. Saliva and adjacent tissue samples were used for whole-exome sequencing, and only saliva samples were used for genotyping.

### Genotyping

Genotyping was performed using Illumina's Infinium Multi-Ethnic AMR/AFR-8 v.1.0 array at King's College London and Infinium Global Screening Array v.3.0 at University College London. Sufficient germline DNA was available for genotyping for 80 out of 92 samples (86.9%). Ancestry estimation was performed using PLINK v.1.9, and ADMIXTURE[48] v.1.3.0 for unsupervised analysis together with the superpopulations of the 1000 Genomes dataset[49]. Five superpopulations were identified, corresponding to AFR (Q1), AMR (Q2), SAS (Q3), EAS (Q4) and EUR (Q5) (Supplementary Table 2 and Supplementary Fig. 1).

### Exome sequencing and data quality control

FFPE samples, saliva and normal adjacent tissue underwent whole-exome sequencing as follows: Exome capture was performed using Agilent SureSelect AllExon v.5 probes and paired-end sequencing was performed at the Wellcome Sanger Institute (UK) in Illumina HiSeq 4000 machines. Control and tumour samples were sequenced to a mean coverage of 43.72×. Alignment was done using BWA-mem[50] v.0.7.17, using the GRCh38 reference genome. Sequencing quality filters were performed using Samtools v.1.9 stats[51] and fastqc v.0.11.3 (ref. 52). Sample contamination was estimated using the GATK v.4.2.3.0 tool CalculateContamination[53]. Concordance between sample pairs was estimated using the Conpair v.0.2 tool[54]. Samples that had less than 90% similarity with their pair (tumour–normal) or showed a level of contamination above 5% were excluded from the study. After this step, 123 samples remained for further analysis.

### Somatic SNV calling and identification of driver genes and mutations

The nature of our samples (FFPE) may introduce artefacts that affect our ability to identify SNVs and indels accurately. Therefore, to mitigate this risk and increase specificity, we used three different variant calling tools, albeit at the cost of reduced sensitivity. As formalin fixation can generate DNA fragmentation, this may also affect copy number estimation analyses and, consequently, copy number mutational signature analysis. To mitigate this, we stringently filtered the samples used for this analysis, which affected our statistical power due to a reduced sample size.

Somatic variant calling was done using three different tools (cgp-CaVEMan[55] v.1.15.2, Mutect2 (ref. 56) v.4.1.0.0 and Varscan2 (ref. 57) v.2.3.9), keeping only the variants identified by a minimum of two out of the three tools. SmartPhase v.1.2.1 (ref. 58) was used for variant pairs phasing. VCF handling was done using bcftools v.1.9 (ref. 59). For $BRAF^{V600E}$ mutations, we kept these variants even if they were identified by only one of the tools as its oncogenicity and relevance in melanoma is well known. When available within the variant calling tool, strand bias filters were applied. A minimum base quality score of 30 on the Phred scale was used. Indel calling was performed using cgpPindel[60,61] v.1.15.2. When selecting one sample per patient, preference was given to primaries, and metastases or recurrences were chosen only when a primary had not been collected.

Significantly mutated genes were identified using the tool dNdScv[62] v.0.0.1.0 with default parameters using SNVs identified by two of the three tools used for variant calling and indels identified by pindel as input data. Positive selection was considered for genes that had global $Q$ values below 0.1 according to the dNdScv tool recommendations. Visualization of variants was done using Maftools v.2.2.10 (ref. 63).

Two lymph node metastasis samples (one from a patient that had a $BRAF^{V600E}$ mutation and another one with an $NRAS$ mutation) and their primaries were annotated as having the same mutation for follow-up analysis after manual inspection using IGV[64].

### Analysis of correlation between driver mutations and clinical covariates and ancestry

Statistical tests were performed to identify potential clinical and ancestry covariates that correlated with driver mutational status. For tumour stage, sex, ulceration status and tumour site, which are discrete variables, association was tested with contingency Chi-squared tests.

For each of the four driver genes, a logistic regression model was fitted to predict the presence or absence of a mutation on the acral melanoma samples using the inferred ADMIXTURE[48] cluster related to the European ancestry component from the 1000 Genomes Project, correcting for age, sex and total TMB (total TMB, SNVs + indels), as such: driver gene status ~ EUR related cluster proportion + age + sex + total TMB. The log odds related to the EUR cluster were then plotted with their respective confidence intervals. The models were constructed using 80 samples out of the 92, which were those with available genotyping information and with all tested covariables available.

### Somatic DNA copy number calling

Sequenza[65] v.3.0.0.0 and ASCAT[66] v.3.1.2 were used to estimate ploidy and purity values for each sample. These values per sample were compared between the two tools, and samples that had a high discrepancy in their purity estimates (less than 0.15 versus 1, respectively) were filtered out (Supplementary Table 28). Samples with an estimated goodness of fit below 95 were also filtered out. Subsequently, copy number, cellularity and ploidy values estimated by ASCAT were used in follow-up analyses. Whole genome duplication events were assigned as reported by ASCAT. Regions significantly affected by CNAs were identified using GISTIC2 (ref. 67) v.2.0.23. Amplifications were classified as low-level amplifications when regions had a copy number gain above

0.25 and below 0.9, and as high-level amplifications when regions had a copy number gain above 0.9 according to GISTIC2 values; deletions were classified as low-level deletions when regions had a copy number change between −0.25 and −1.3 and as high-level deletions when regions had a copy number change lower than −1.3. Only peaks with residual $Q$ values < 0.1 were considered as significantly altered. For the analyses of differences in CNA burden by sample group (mutational status or site of presentation), we used the CNApp tool[68] to generate GCS, focal copy number alteration scores and broad copy number alteration scores, calculating segment means (seg.mean) as $\log_2$(cn/ploidy) and using default parameters. GCS is a number quantifying the copy number aberration level in each sample provided by the CNApp tool[68]. Higher GCS scores indicate a higher burden of copy number aberrations compared with all other samples in the cohort. GCS is the sum of the normalized broad copy number alteration score and focal copy number alteration score, which are calculated considering broad (chromosome and arm-level) and focal (weighted focal CNAs corrected by the amplitude and length of the segment) aberrations per sample. These values are calculated using as input the number of DNA copies normalized by sample ploidy. A more detailed explanation can be found in the original publication[68]. GCS values were used for comparisons between sample groups. All paired comparisons between groups were evaluated with a Mann–Whitney test.

To further scrutinise the presence of deletions in tumour suppressors *NF1* and *CDKN2A*, we used CNVkit[69] v.0.9.10. We called copy number alterations against a pooled reference generated from the highest quality normal samples, and generated bin-level and segmented level $\log_2$ ratios. We calculated the $\log_2$ ratio estimated for homozygous deletions for each sample based on ASCAT's estimation of ploidy and purity as published previously[25]. For *NF1*—a large gene—we considered homozygous focal deletions when at least two contiguous bins had $\log_2$ ratios at or below the calculated threshold for that sample, or at least one full exon has read coverage equal to zero. For *CDKN2A*—a small gene—we considered a sample as having a homozygous deletion if it had at least one bin below the threshold, at least two other bins close to the threshold and a noticeable difference in $\log_2$ ratios for bins falling in *CDKN2A* in comparison with its neighbours by manual scrutiny.

## Mutational signature analysis

Mutational matrices were generated using SigProfilerMatrixGenerator[70] v.1.2.20. These matrices, with single nucleotide mutations found by at least two of the three variant callers and all insertions and deletions identified by cgpPindel, were used as input for mutational signature extraction using SigProfilerExtractor[71] v.1.1.23 and decomposition to COSMICv.3.4 (ref. 72) reference mutational signatures[72]. For single base substitutions, the standard SBS-96 mutational context was used, with default parameters and a minimum and maximum number of output signatures being set as one and five, respectively. A total of 116 samples with an SNV count > 0, were used for this analysis. For copy number signature analysis, all 60 samples with available copy number data were used with default parameters, and using the standard CN-48 context from COSMICv.3.4.

## RNA sequencing and data quality control

Total RNA library preparation followed by exome capture using Agilent SureSelect AllExon v.5 was performed on Illumina HiSeq 4000 machines. Reads were aligned to the GRCh38 reference genome using the splice-aware aligner STAR[73] v.2.5.0. Of these, we focused on the 77 samples that came from different patients, that had matching DNA and were primaries for the score analysis (see below). We then applied further quality control filters for the consensus clustering analysis: samples were excluded if total read counts were fewer than 25 million, or if the sum of ambiguous reads and no feature counts was greater than the sum of all gene read pair counts. Forty-four samples remained for downstream analysis. Counts were generated with HTSeq[74] v.0.7.2.

Transcripts per million normalization was performed and values were $\log_2$(transcripts per million + 1) transformed.

## Acral versus non-acral cutaneous tumour score

Patient samples collected for the Utah cohort analysis were derived as described previously[47]. Invasive acral and non-acral cutaneous melanomas were identified and collected as part of the University of Utah IRB umbrella protocol no. 76927, Project no. 60, and RNA was extracted and quantified as described previously[47]. A custom NanoString nCounter XT CodeSet (NanoString Technologies) was designed to include genes differentially expressed between glabrous and non-glabrous melanocytes[43,44]. Sample hybridization and processing were performed in the Molecular Diagnostics core facility at Huntsman Cancer Institute. Data were collected using the nCounter Digital Analyzer. Raw NanoString counts were normalized using the nSolver Analysis Software (NanoString Technologies). Normalization was carried out using the geometric mean of housekeeping genes included in the panel (Supplementary Table 16). Background thresholding was performed using a threshold count value of 20. Fold change estimation was calculated by partitioning by acral versus cutaneous melanoma. The $\log_2$ normalized gene expression data were subjected to principal component analysis (PCA) using the PCA function in Prism v.10.2.1 (GraphPad Software). PCA was performed to identify the main sources of variability in the data and to distinguish between acral and cutaneous samples.

To determine the top differentially expressed genes contributing to the variance between acral melanomas and cutaneous melanomas, the loadings of the second principal component (PC2) were examined. Genes with the highest positive and negative loadings on PC2 were selected as the top ten and bottom ten genes, respectively; $\log_2$ expression values of these genes were used to generate a multiplicative score, producing the ratio of acral to cutaneous melanocyte genes. Statistical analyses were performed using Prism v.10.2.1 (GraphPad Software). Differences in acral to cutaneous ratios were assessed using the Mann–Whitney $U$ test.

The acral:cutaneous (A:C) ratio was calculated for each of the 77 primary acral tumours using the method described above after batch correction (limma v.3.64.1, ref. 75) on normalized and transformed expression data processed by the R package DESeq2 v.1.48.1 (ref. 76). Differences in the A:C gene expression ratio scores between *BRAF*-activating mutation-positive and *BRAF*-wild-type acral melanoma samples were assessed using a Mann–Whitney $U$ test. The same normalization, scoring method and statistical testing was applied to the 63 transcriptomes from acral melanoma tumours considering *BRAF*-activating ($n = 10$) and wild type ($n = 53$) in Newell et al.[15]. All available samples in this cohort were used, as only one primary had a *BRAF* mutation. Only samples with *BRAF*-activating mutations (V600E and L597R for the Mexican acral melanoma set) were included in the *BRAF* group.

To determine whether the cutaneous melanoma classifier genes are induced by $BRAF^{V600E}$ signalling in melanocytes, we analysed RNA sequencing (RNA-seq) data from McNeal et al.[40], which consisted of bulk-RNA-seq of primary human melanocytes transduced with $BRAF^{V600E}$ and cultured under two conditions: phorbol 12-myristate 13-acetate (PMA) and endothelin-1 (ET1). We extracted normalized expression values for cutaneous melanoma classifier genes across four conditions: PMA, PMA + $BRAF^{V600E}$, ET1 and ET1 + $BRAF^{V600E}$. Normalized expression levels were compared using the Mann–Whitney $U$ test in Prism v.10.2.1 (GraphPad Software).

We evaluated the A:C classifier in clinical melanoma samples using RNA-seq data from TCGA Skin Cutaneous Melanoma Firehose Legacy cohort. Normalized gene expression data were downloaded from cBioPortal[77]. Samples were classified as *BRAF*-activating or *BRAF*-wild type in the same way as for Fig. 3c. We calculated the product of the expression of cutaneous melanoma classifier genes for each category. Differences were assessed using the Mann–Whitney $U$ test.

We used an interactive Shiny application, What Is My Melanocytic Signature (WIMMS; https://wimms.tanlab.org)[41] to compare transcriptional programs associated with distinct melanocytic cell states. WIMMS classifies melanocytic gene expression profiles by aggregating previously published gene expression signatures and clusters them into seven principal cell state categories. We input our gene signature into WIMMS to assess correlation with these reference states. The resulting dendrogram (Extended Data Fig. 6b) represents the relationship between our classifier-derived cutaneous melanoma genes and known signatures.

## Consensus clustering and deconvolution based on gene expression

To identify molecular subgroups based on transcriptome data, we performed consensus clustering using the Cola R v.2.10.1 package[78]. Standard preprocessing of the input matrix was performed, including removal of rows in which more than 25% of the samples had 'NA' (not available) values, imputation of missing values, replacement of values higher than the 95th percentile or less than 5th by corresponding percentiles, removal of rows with zero variance and removal of rows with variance less than the 5th percentile of all row variances. Consensus clustering was performed using several algorithms (k-means, partitioning around medoids, hierarchical clustering) and feature selection methods (s.d., median absolute deviation, coefficient of variation) to ensure robust partition identification. The optimal number of clusters was determined using several stability metrics including 1-PAC (Proportion of Ambiguous Clustering) score, concordance and index, Jaccard index, coefficient and visual inspection of the consensus matrix through heatmaps visualizations. The best-performing method (s.d.:partitioning around medoids with $k = 3$) was selected based on highest consensus scores and biological interpretability.

**Two-level signature analysis.** Following sample clustering, we performed a two-level signature analysis to characterize both sample clusters and gene co-expression modules.

**Sample cluster characterization.** Characterization of sample clusters was performed by identifying genes with significantly different expression across the three identified sample groups using F-tests with false discovery rate correction ($P < 0.05$). For each differentially expressed gene, we determined the sample cluster with highest mean expression to characterize the molecular profiles of patient subgroups.

**Gene module identification.** To understand co-expression patterns within signature genes, we applied k-means clustering ($k = 3$) to group signature genes based on their scaled expression profiles across the three sample clusters. This identified gene modules (M1, M2, M3) representing distinct biological programs that may be co-ordinately regulated across different sample subtypes (Supplementary Tables 17–20).

**Functional enrichment analysis.** Functional enrichment was performed separately for each gene module using over-representation analysis with the clusterProfiler R package v.4.12.6 (ref. 79). Gene Ontology Biological Process terms were tested using the hypergeometric test with Benjamini–Hochberg false discovery rate correction ($Q < 0.05$). Ensembl gene identifiers were mapped to gene symbols using the org. Hs.eg.db v.3.19.1 (ref. 80) annotation package before enrichment analysis (Supplementary Tables 17–20).

The EPIC algorithm[81] v.1.1.7 was used in the R programming environment to perform deconvolution to infer immune and stromal cell fractions within acral melanoma tumours. We used the TRef signature method with default parameters, which includes gene expression reference profiles from tumour-infiltrating cells. The algorithm generated an absolute score that could be interpreted as a cell fraction.

## Survival analyses

Consenting and recruitment of patients started in 2017 and ended in 2019. Because of the challenges of recruiting sufficient numbers of participants with acral melanoma, patients diagnosed in earlier years who were still attending follow-up clinics for primary or recurrent disease were recruited. To ensue data consistency, only participants with a primary available for analysis were the subject of focus on analyses of recurrence and/or death ($n = 85$ patients). In total, 73 participants were recruited whose primary and ancestry data were available for analysis on driver mutations. A total of 44 patients with primary and RNA cluster data available were used for analysis on clusters regardless of their ancestry data availability. Analyses were performed with Stata v.19.5 (ref. 82).

To compare the effect of distinct driver mutations and the RNA clusters, we examined two measures of disease severity: (1) the recurrence of the primary tumour and (2) overall survival. For recurrence, we examined time to recurrence using a life-table approach from date of primary diagnosis onwards as a descriptor, and based analyses of differences between mutations (and clusters) on logistical regression adjusting for date of diagnosis, age at diagnosis, sex, stage at diagnosis (advanced/early), ancestry (only for mutations) and time from diagnosis through either death or last known to be alive. Primary tumours were either classified as QWT or mutated for known drivers; tumours with several driver mutations were classified as 'multi-hit'. We also conducted analyses based on a binary exposure of 'QWT' or 'Mutated' tumour based on the existence of one or more mutations in a known driver gene.

For survival analysis, we also included a life-table approach, again as a descriptor, based on time from diagnosis through to date of death or date last known to be alive. Statistical assessment of the effect of each mutation and/or cluster were based on Cox proportional hazard analysis with follow-up starting from date of recruitment through to date of death (or last date alive) adjusting for date of diagnosis, age at diagnosis, sex and ancestry (as a sensitivity analysis). Analysis of combined mutations were also conducted as for the recurrence analysis.

To assess the impact of any biases on our interpretation of the impact of mutations, we restricted attention to the 40 tumours diagnosed since the beginning of 2017, that is, those closest to the time of recruitment. Analysis of survival gave qualitatively and quantitatively similar results to those reported above; in total, 7 of 23 (30.4%) of cases with mutated tumours had died, as opposed to 1 of 17 among cases with QWT tumours ($P$ value = 0.055). In the analysis of recurrence, again results matched with 70.0% of cases with mutated tumours recurring (16 of 23 cases) compared with 17.7% of QWT tumours (3 of 17; $P$ value = 0.001). Results for the RNA clusters were similar to the results quoted above for both survival and recurrence.

Data formatting and handling were performed using Python and R.

## Reporting summary

Further information on research design is available in the Nature Portfolio Reporting Summary linked to this article.

## Data availability

Sequencing data are available at the European Genome-Phenome Archive (EGA). DNA sequencing data are available under ENA accession number EGAD00001015755 and RNA-seq data under ENA accession number EGAD00001015756. The 1000 Genomes Project datasets can be downloaded from https://www.internationalgenome.org/data. The GRCh38 reference genome can be downloaded from https://www.ncbi.nlm.nih.gov/datasets/genome/GCF_000001405.40/. Sequencing data for the Newell et al.[15] study is available from the EGA under study accession EGAS00001001552 and dataset accession EGAD00001005500. Access to the data can be gained through application to the Data Access Committee for the dataset. Information on how to apply for access is available at the EGA dataset link: https://ega-archive.org/datasets/EGAD00001005500. TCGA Skin Cutaneous Melanoma Firehose Legacy cohort data can be downloaded from cBioPortal (https://www.cbioportal.org/). RNA sequencing data from McNeal et al.[40] is available from GEO under accession number GSE150849.

## Code availability

Code for reproducing the analyses in this paper is available at https://github.com/CGBio-Lab/Mex-acral-exomes-transcriptomes.

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

**Acknowledgements** We are deeply grateful to patients and their families for agreeing to form part of this study and providing access to their samples. We are also thankful to members of the CGBio laboratory team at LIIGH-UNAM for valuable discussions regarding the findings in this article. We wish to thank L. A. Aguilar, A. de León and A. Avalos from the Laboratorio Nacional de Visualización Científica Avanzada and J. S. García Sotelo, A. Hernández, E. Lomelín, I. Martínez, R. Muciño, M. A. Ávila, A. Castillo and C. Uribe Díaz from the International Laboratory for Human Genome Research, National Autonomous University of Mexico. We are grateful to the International Cancer Genome Consortium Data Access Committee for granting access to ICGC controlled data. We are also thankful to K. Wong and D. Desposorio for useful discussions. Work included in this paper has been funded by Wellcome Trust (204562/Z/16/Z and 227228/Z/23/Z to C.D.R.-E.), the Melanoma Research Alliance (Pilot Award #825924, to C.D.R.-E.), the Mexican National Council of Humanities, Science and Technology (CONAHCYT/ SECIHTI, FOSISS A3-S-31603, to C.D.R.-E.), Programa de Apoyo a Proyectos de Investigación e Innovación Tecnológica (PAPIIT UNAM) (IN209422 to C.D.R.-E.), the Academy of Medical Sciences through a Newton Advanced Fellowship (NAF/R2/180782) and the Wellcome Sanger Institute through an International Fellowship. This project was also supported by the MRC Dermatlas project; MR/V000292/1. C.D.R.-E. is grateful to the William Guy Forbeck Research Foundation for their generous support and for promoting a collaborative and rich environment that helped advance the ideas underlying this study. A.J., D.C.D. and R.L.J.-T. are supported by the Department of Dermatology and the Huntsman Cancer Foundation. This work was funded in part by the Melanoma Research Alliance Dermatology Fellows award to D.C.D., the Harry J Lloyd Charitable Trust Melanoma Research Grant to R.L.J.-T., a National Cancer Institute R01 (R01CA229896) to R.L.J.-T. and pilot funds from the Huntsman Cancer Institute Melanoma Center. We used the Shared Resources for Research Informatics and High-Throughput Genomics and Bioinformatics Analysis, each supported by the National Cancer Institute of the National Institutes of Health under Award Number P30CA042014. M.D.-G. and P.G.-G. were awarded fellowships within the 'Generación D' initiative, Red.es, Ministerio para la Transformación Digital y de la Función Pública, for talent attraction (C005/24-ED CV1), funded by the European Union NextGenerationEU funds, through PRTR. This work was in part supported by the US National Institute of Health grants R01ES032547, R01ES036931, R01CA269919, R01CA296974, P01CA281819 and U01CA290479 to L.B.A. as well as by L.B.A.'s Packard Fellowship for Science and Engineering and the UC San Diego Sanford Stem Cell Institute. P.B.-L. is a PhD student from Programa de Doctorado en Ciencias Biológicas, Universidad Nacional Autónoma de México (UNAM), and was supported by Consejo Nacional de Humanidades, Ciencia y Tecnología (CONAHCyT, now known as SECIHTI) (holder no. 562546, scholarship no.762536). P.B.-L. is grateful to the Posgrado en Ciencias Biológicas for the support received during her doctoral studies. This paper is part of P.B.-L.'s requirements for obtaining a Doctoral degree at the Posgrado en Ciencias Biológicas, UNAM.

**Author contributions** P.B.-L., M.E.V.-C., D.C.-I., E.F.-R., J.B., P.A.J., I.S.-W., J.R.C.W.-R., K.L.C.-R., A.J., D.C.D., J.I.R.-G., O.I.G.-S. and M.d.C.V.-H. performed bioinformatic and statistical analyses. C.M.-A., F.G.A.-G., M.C.-V., R.O.-L. and L.v.d.W. did sample cataloguing and nucleic acid extraction. E.T.D. provided computational resources and advice on statistical analyses. A.A.-C., D.Y.G.-O., H.M.-S., R.R.-M., H.V.d.l.C., L.A.T.-d.-l.-P. and D.H.-U. assessed patients and provided access to biological samples. A.H.-M. provided facilities for sample processing and supervised that part of the work. M.J.A., I.F. and M.T. performed sample histopathology. P.G.-G., M.D.-G. and L.B.A. supervised the mutational signatures analysis. Y.S.-P. provided access to patient clinical information and supervised that part of the work. G.K.I., R.L.B. and R.M.W. provided data and information that crucially helped the interpretation of the results in this manuscript. D.T.B. performed survival statistical analyses. P.A.P., R.L.J.-T., D.J.A. and C.D.R.-E. jointly supervised this work. C.D.R.-E. wrote the manuscript with assistance from P.B.-L., P.A.P., R.L.J.-T. and D.J.A.

**Competing interests** L.B.A. is a co-founder, CSO, scientific advisory member and consultant for io9 (now Acurion), has equity and receives income. The terms of this arrangement have been reviewed and approved by the University of California, San Diego in accordance with its conflict-of-interest policies. L.B.A. is a compensated member of the scientific advisory board of Inocras. L.B.A.'s spouse is an employee of Hologic, Inc. L.B.A. declares US provisional applications with serial numbers: 63/289,601; 63/269,033; 63/366,392; 63/412,835 as well as international patent application PCT/US2023/010679. L.B.A. is also an inventor of a US Patent 10,776,718 for source identification by non-negative matrix factorization. L.B.A. and M.D.-G. further declare a European patent application with application number EP25305077.7. All other authors declare no competing interests.

**Additional information**
**Correspondence and requests for materials** should be addressed to Carla Daniela Robles-Espinoza.

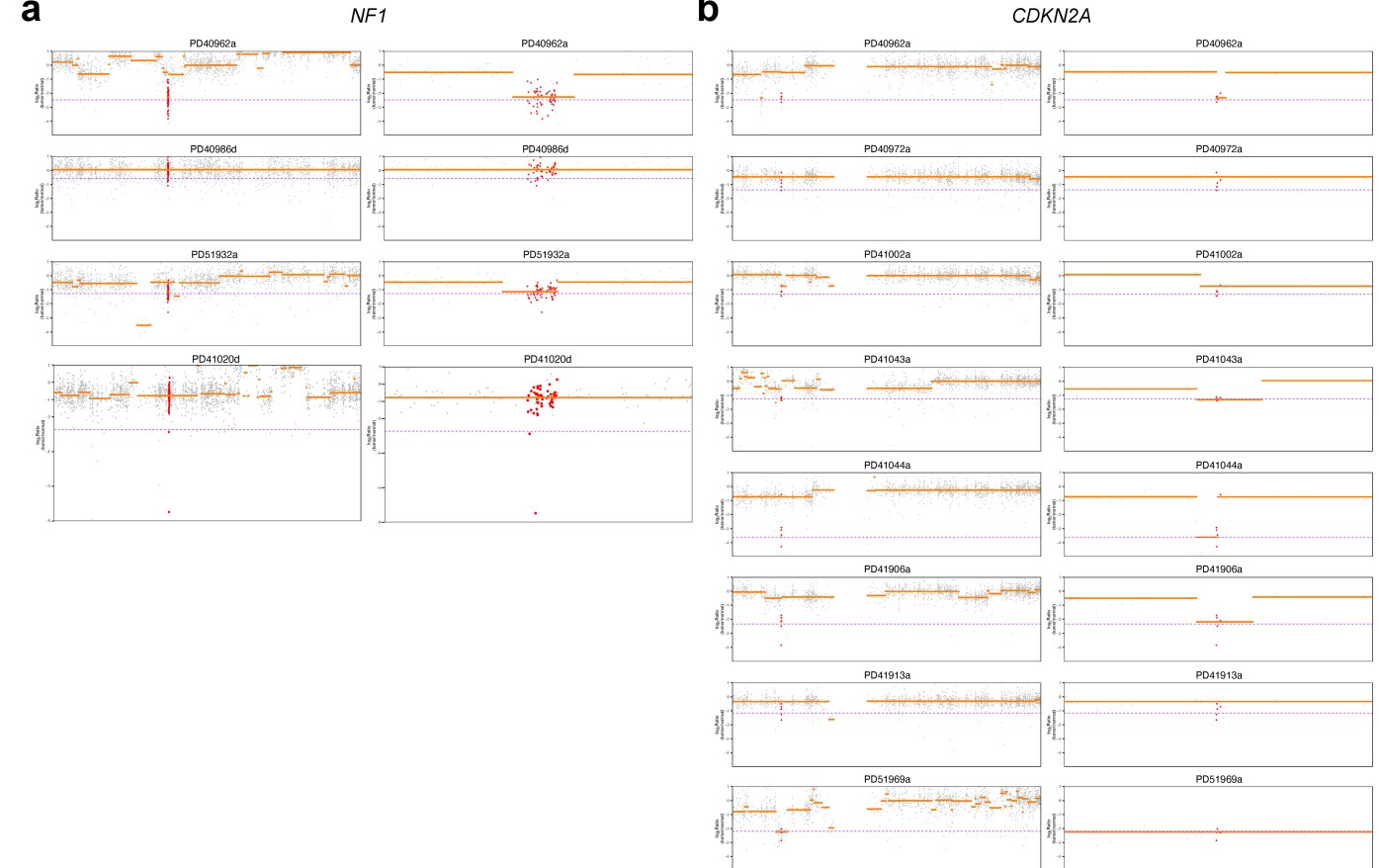

**Extended Data Fig. 1 | Analysis of the *NF1* and *CDKN2A* regions through re-segmented bin-level copy number data for acral melanoma samples with homozygous deletions in these genes.** Each row represents one sample, with the left panel displaying the whole chromosome and the right panel a close up view at the gene region. Each point represents a bin as calculated by the CNVkit algortihm, the x-axis represents chromosomal position and the y-axis the log$_2$ ratio calculated with CNVkit. Red dots indicate bins falling in the gene region. The purple dotted line represents the expected log$_2$ ratio for a homozygous deletion (Methods). The orange lines represent segmented data. a) *NF1* gene. PD40986d and PD41020d are secondary samples and a different sample was selected as representative for these patients. b) *CDKN2A* region.

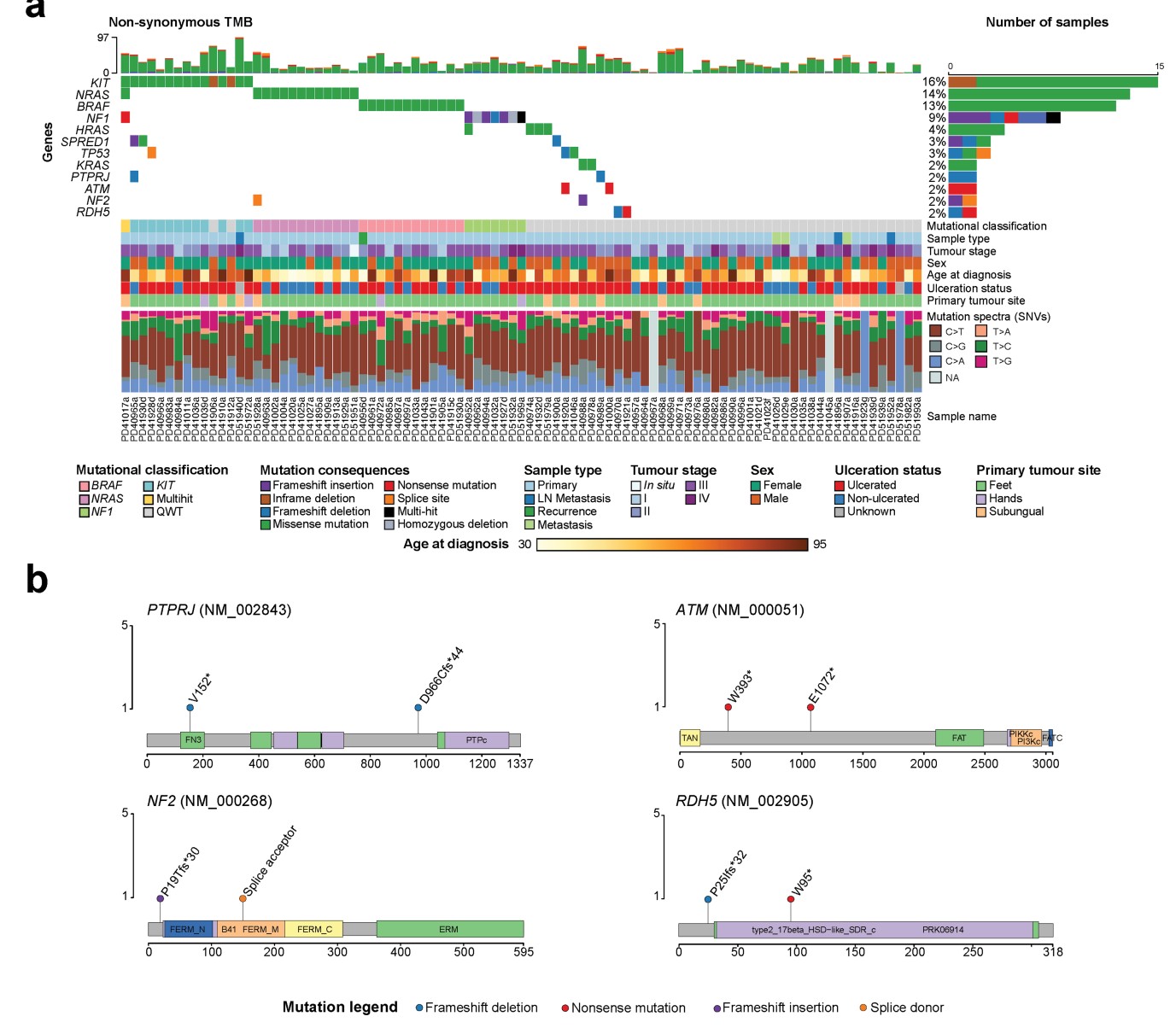

**Extended Data Fig. 2 | Somatic landscape of all acral melanoma samples, one selected per patient (n = 91).** a) Oncoplot depicting the seven most mutated genes according to dNdScv and five selected genes based on mutational frequency and biological function. Mutational classification, sample type, tumour stage, sex, age at diagnosis, ulceration status, tumour site and mutational spectra are shown by sample. Primary samples were selected preferentially for this analysis. One sample where no mutations were detected is not depicted in the oncoplot. b) Mutations found in *PTPRJ*, *ATM*, *NF2* and *RDH5*, for which all mutations are deleterious and are found altered each in two samples.

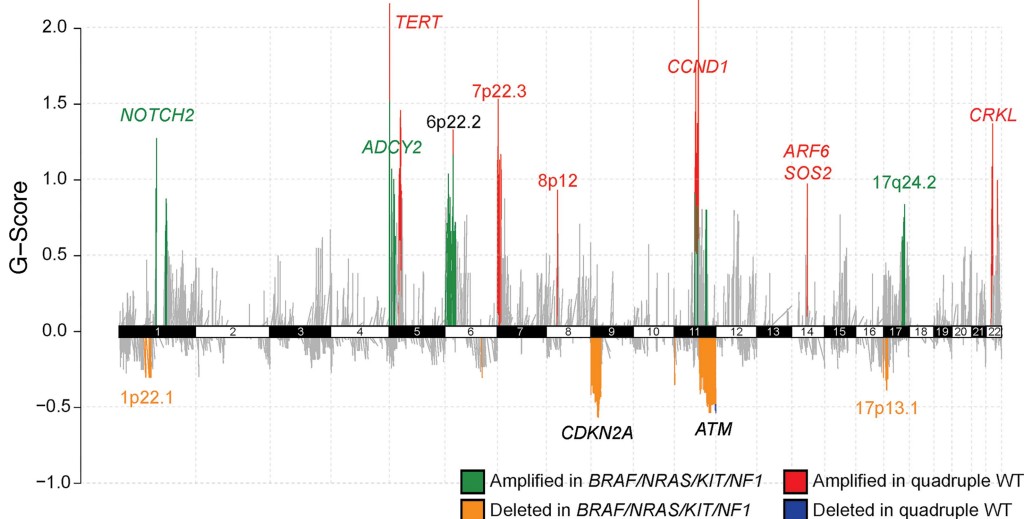

**Extended Data Fig. 3 | Copy number profile of *BRAF/NRAS/KIT/NF1*-mutated tumours vs. quadruple WT.** All depicted regions have been identified by GISTIC2 analysis per group of samples. Statistically significant differences are marked in colour, green = amplified in mutated tumours, red = amplified in QWT, yellow = deleted in mutated tumours, blue = deleted in QWT tumours. Differences are determined first by assessing the global GISTIC2 output and determining differences between groups by one-sided Fisher's exact test (*P*-value < 0.05). If a region is not found in the global GISTIC2 output, but it is found only in the analysis per group, we have indicated it as statistically different. Number of mutated tumours = 23, number of QWT tumours = 24.

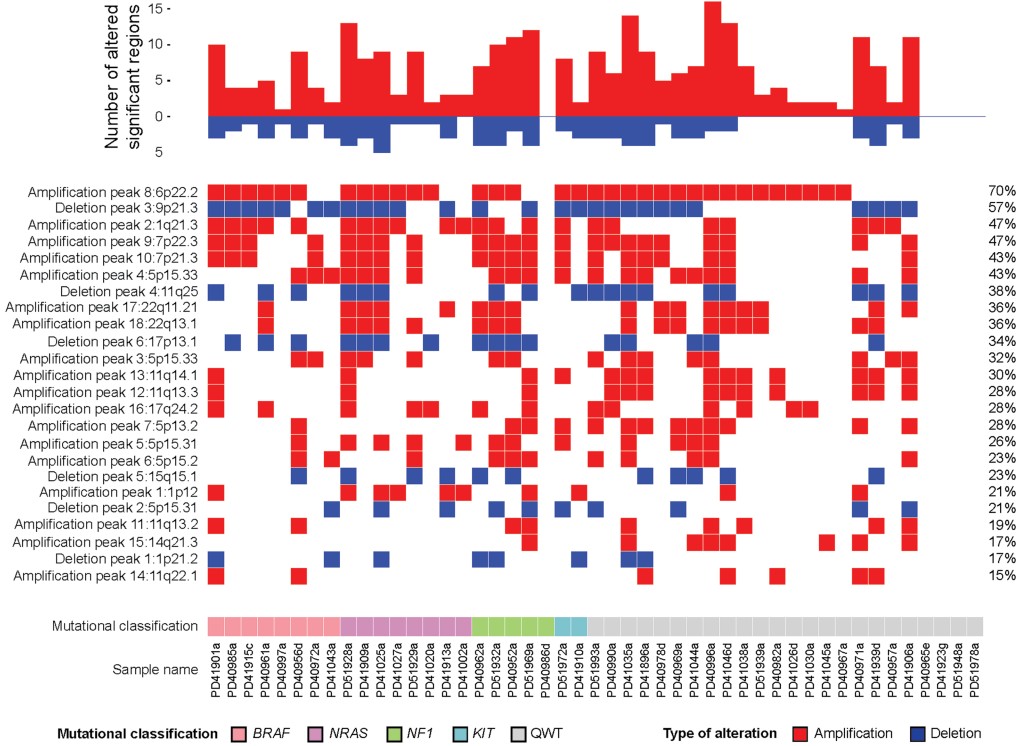

**Extended Data Fig. 4 | Significantly altered regions per sample by GISTIC analysis.** Top panel. Number of significant regions altered by sample. Bottom panel. Binary heatmap showing the significantly altered regions identified by GISTIC2 per sample. One sample per patient is shown. Heatmap is ordered on the X axis by mutational classification and on the Y axis by frequency of alterations per region. As a note, sample PD41002a does not show a deletion in *CDKN2A* (Deletion peak 3) by GISTIC2 analysis, but a deletion was detected by CNVkit (Methods, Extended Data Fig. 1).

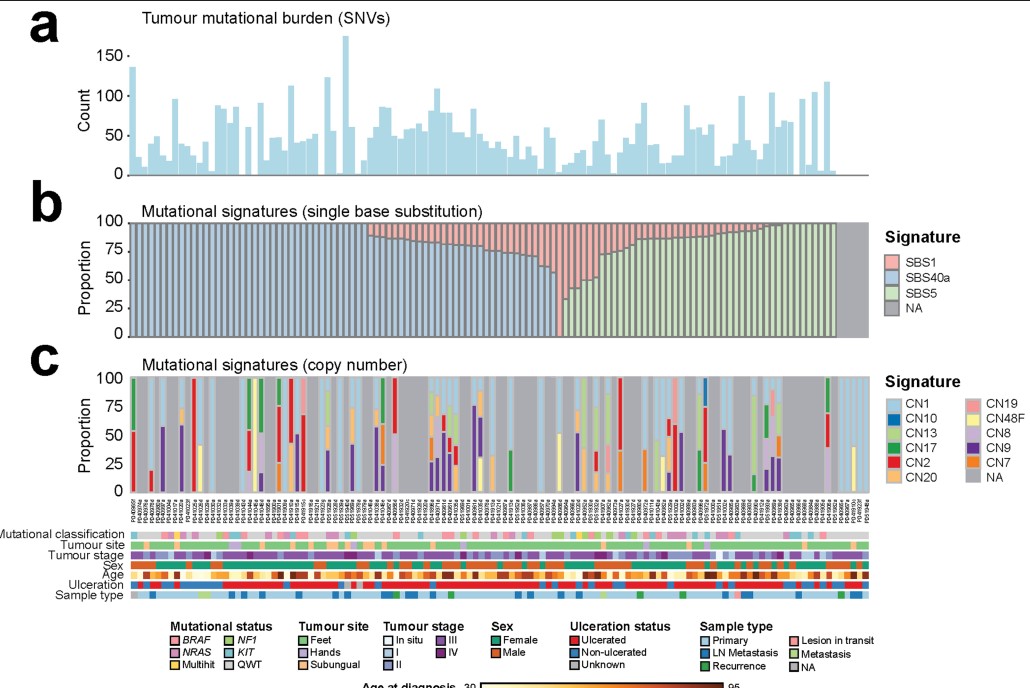

**Extended Data Fig. 5 | Mutational signatures found in acral melanoma samples from Mexican patients.** a) The SNV component of tumour mutational burden per sample. b-c) Proportions of mutational signatures per sample are shown in stacked bars for single base substitutions (b), and copy-number aberrations (c). In b) and c), samples with a light gray background did not have data available. Genomic subtypes and clinical characteristics are plotted at the bottom. As a note, mutational signature CN48F in c) is a *de novo* mutational signature that was not successfully reconstructed by COSMIC reference mutational signatures, and was therefore considered as novel.

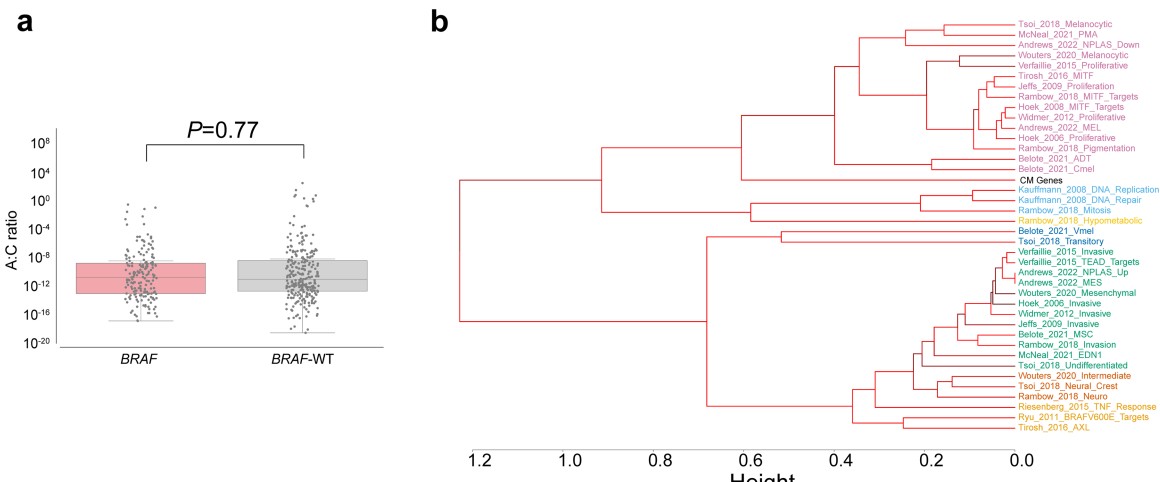

**a**

**b**

**Extended Data Fig. 6 | Analyses testing the association of the transcriptional signature found in acral *BRAF*-mutated tumours with downstream *BRAF* signalling.** A) Scatter plot comparison of Acral:Cutaneous gene expression ratio in cutaneous melanoma samples from The Cancer Genome Atlas (TCGA) stratified by activating mutation status. Samples were compared in two groups: non-*BRAF* activated tumours and *BRAF*-activating tumours. Statistical significance was assessed using individual Mann-Whitney U test. B) Hierarchical clustering dendrogram generated using the WIMMS platform to compare the cutaneous melanoma classifier genes to other published molecular signatures, including a signature of genes activated by mutant *BRAF*$^{V600E}$ in melanoma cells (Ryu_2011_BRAFV600E_Targets)[42].

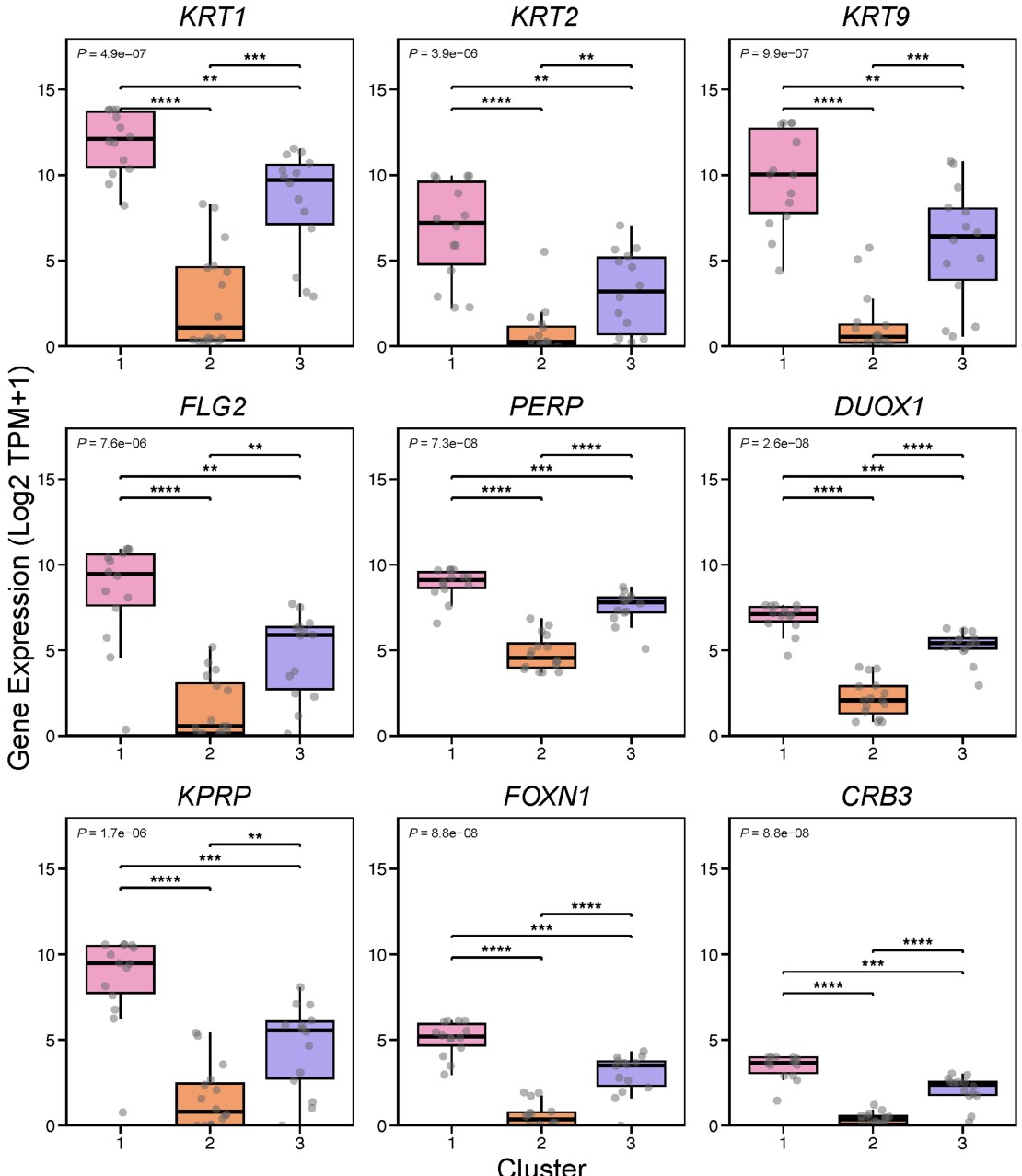

**Extended Data Fig. 7 | Gene expression for a selection of genes found associated to Cluster 1.** *P*-values were estimated with Kruskal-Wallis tests.

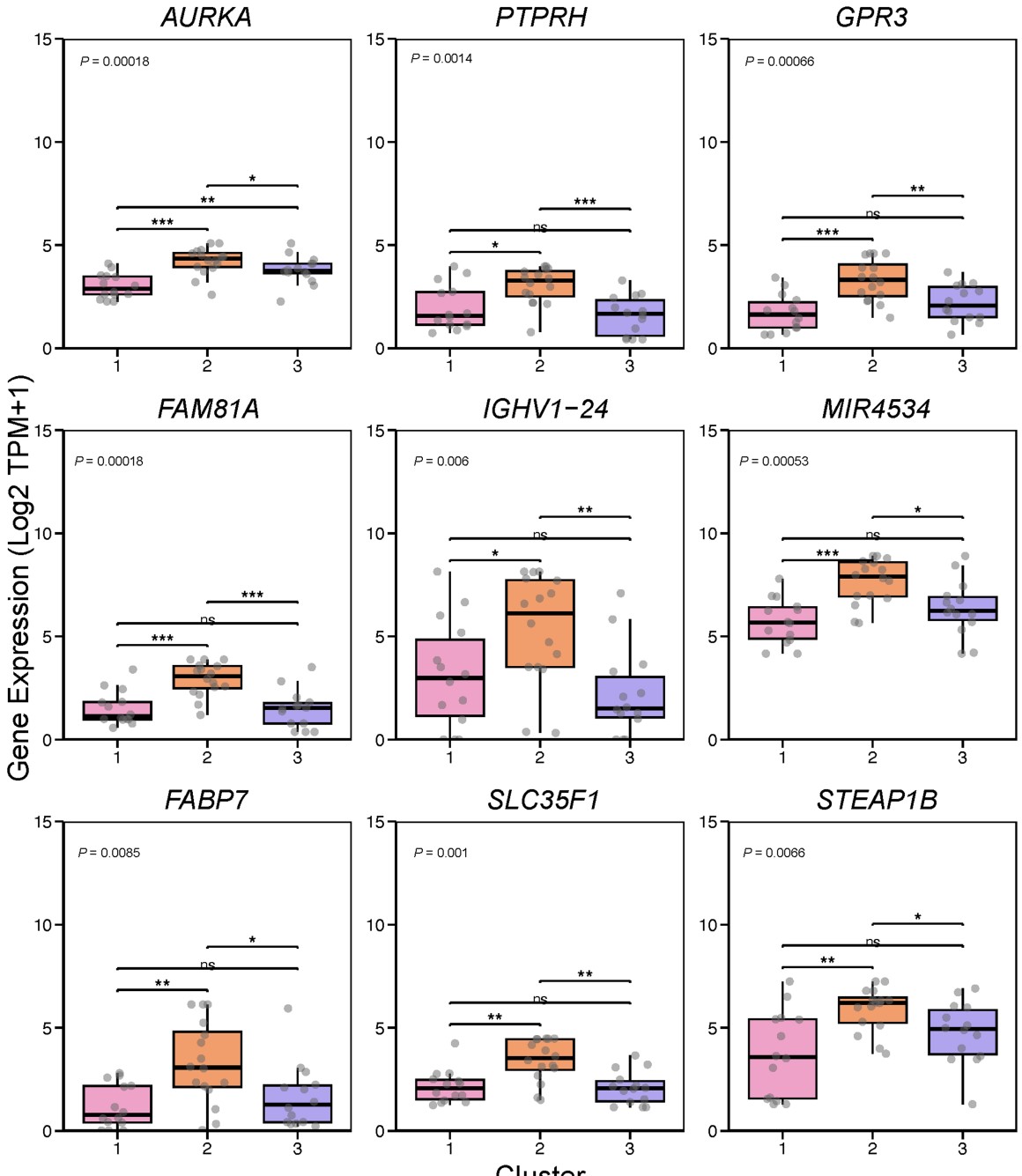

**Extended Data Fig. 8 | Gene expression for a selection of genes found associated to Cluster 2.** *P*-values were estimated with Kruskal-Wallis tests.

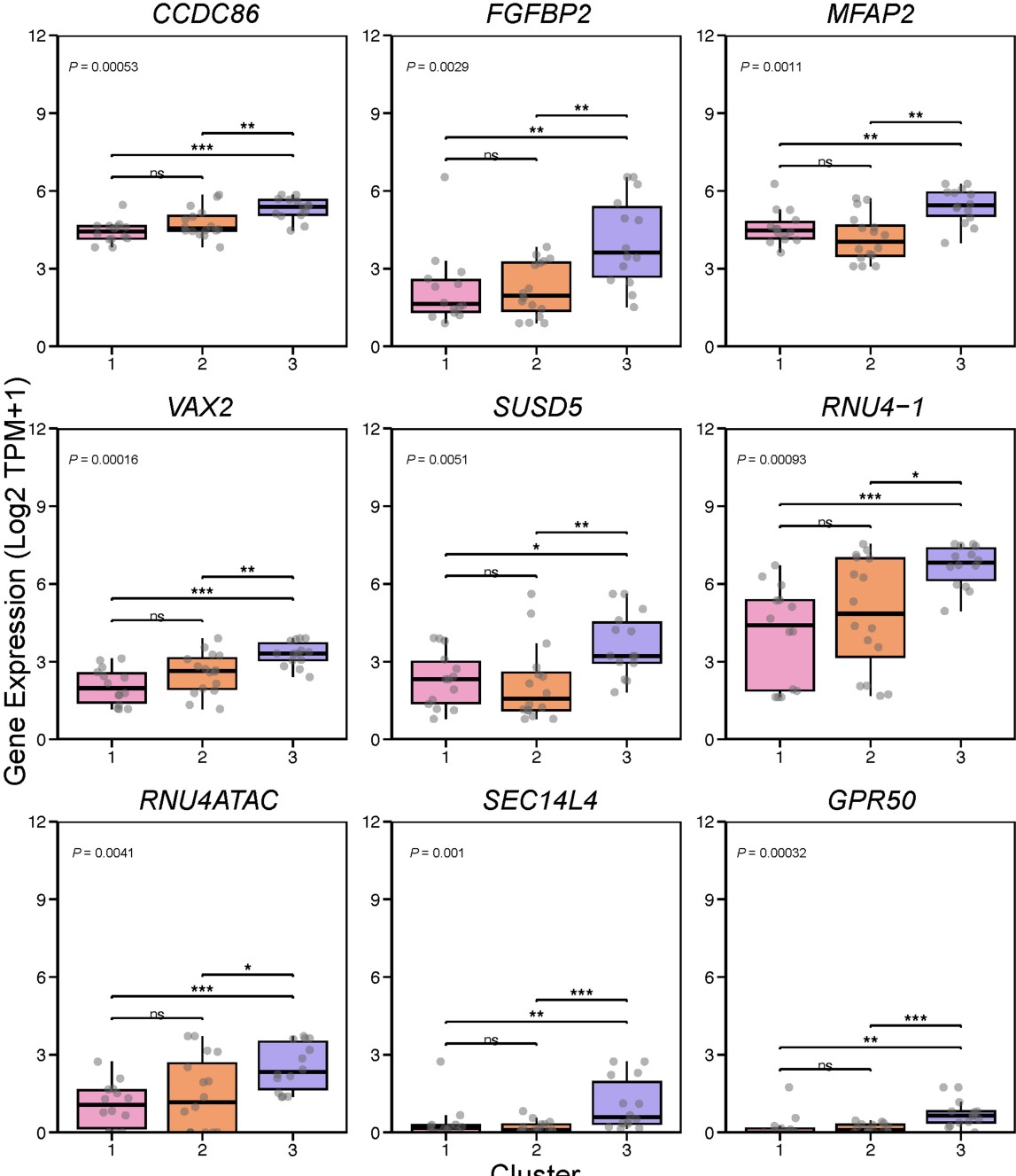

**Extended Data Fig. 9 | Gene expression for a selection of genes found associated to Cluster 3.** *P*-values were estimated with Kruskal-Wallis tests.

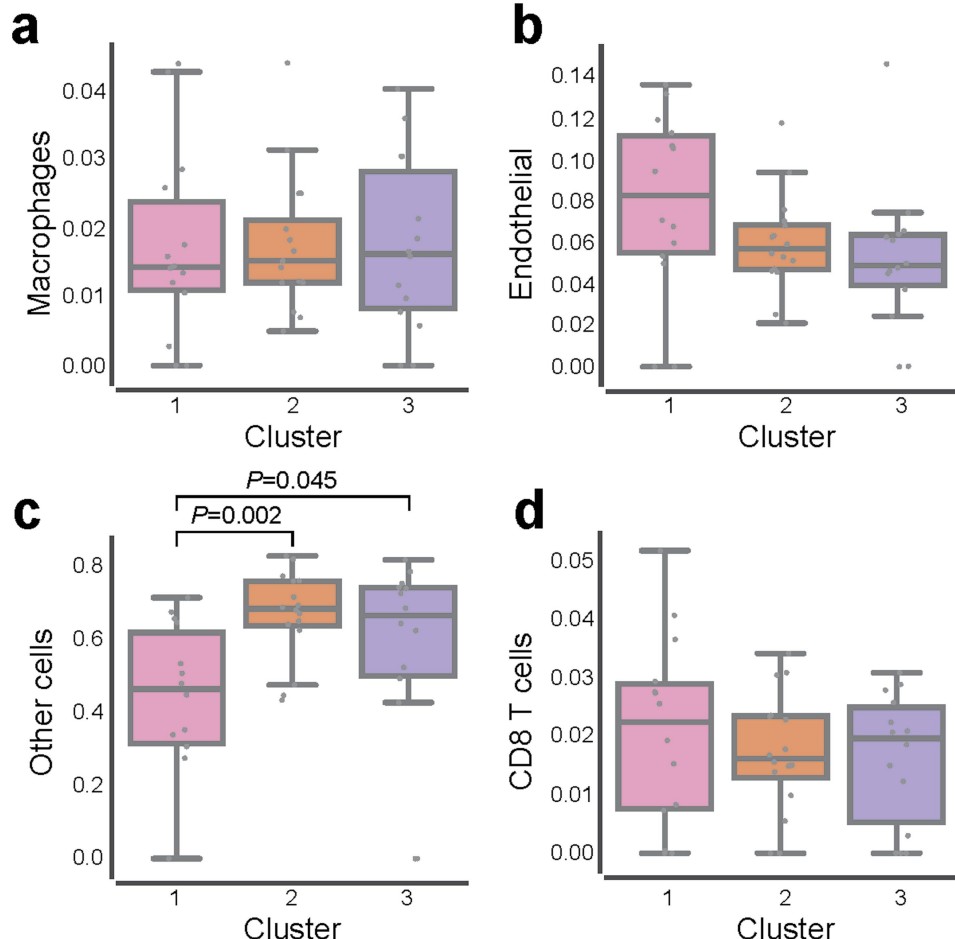

**Extended Data Fig. 10 | Deconvolution results for macrophages, endothelial cells, CD8 T cells and other cells.** A) Box plot of macrophage proportion (Y axis), as calculated by deconvolution, per sample classified by transcriptional cluster. B) Box plot of endothelial cell proportion (Y axis), as calculated by deconvolution, per sample classified by transcriptional cluster. C) Box plot of other cells (non-immune), as calculated by deconvolution, per sample classified by transcriptional cluster. D) Box plot of CD8 T cell proportion (Y axis), as calculated by deconvolution, per sample classified by transcriptional cluster. The central line within each box represents the median value, the box boundaries represent the interquartile range (IQR), and the whiskers extend to the lowest or highest data point still within 1.5xIQR. Individual data points are plotted as dots. Wilcoxon-Mann-Whitney paired tests were performed.

# Reporting Summary

## Statistics

For all statistical analyses, confirm that the following items are present in the figure legend, table legend, main text, or Methods section.

| n/a | Confirmed | |
|---|---|---|
| ☐ | ☒ | The exact sample size (*n*) for each experimental group/condition, given as a discrete number and unit of measurement |
| ☐ | ☒ | A statement on whether measurements were taken from distinct samples or whether the same sample was measured repeatedly |
| ☐ | ☒ | The statistical test(s) used AND whether they are one- or two-sided *Only common tests should be described solely by name; describe more complex techniques in the Methods section.* |
| ☐ | ☒ | A description of all covariates tested |
| ☐ | ☒ | A description of any assumptions or corrections, such as tests of normality and adjustment for multiple comparisons |
| ☐ | ☒ | A full description of the statistical parameters including central tendency (e.g. means) or other basic estimates (e.g. regression coefficient) AND variation (e.g. standard deviation) or associated estimates of uncertainty (e.g. confidence intervals) |
| ☐ | ☒ | For null hypothesis testing, the test statistic (e.g. *F*, *t*, *r*) with confidence intervals, effect sizes, degrees of freedom and *P* value noted *Give P values as exact values whenever suitable.* |
| ☒ | ☐ | For Bayesian analysis, information on the choice of priors and Markov chain Monte Carlo settings |
| ☒ | ☐ | For hierarchical and complex designs, identification of the appropriate level for tests and full reporting of outcomes |
| ☐ | ☒ | Estimates of effect sizes (e.g. Cohen's *d*, Pearson's *r*), indicating how they were calculated |

*Our web collection on statistics for biologists contains articles on many of the points above.*

## Software and code

Policy information about availability of computer code

| Data collection | Clinical data was collected using Microsoft Excel v16. |
|---|---|
| Data analysis | ADMIXTURE v1.3.0, ancestry estimation |
| | PLINK v1.9, quality control and merging acral dataset with 1000 Genome Project for ancestry estimation |
| | BWA-mem v0.7.17-r1188, sequencing read alignment |
| | samtools v1.9, sequencing data analysis |
| | bcftools v1.9, vcf file manipulations |
| | fastqc v0.11.3, sequencing data quality control |
| | GATK v4.2.3.0, genome analysis toolkit including Mutect2 v4.1.0.0, tools for quality, variant calling and data filtering |
| | maftools v2.2.10, visualisation of somatic mutation data |
| | Conpair v0.2, sample concordance estimation |
| | Varscan2 v2.3.9, somatic variant calling |
| | dNdScv v0.0.1.0, driver gene identification |
| | Sequenza v3.0.0, somatic copy number estimation |
| | GISTIC2 v2.0.23, genomic identification of significantly altered regions |
| | CNApp , analysis and comparison of copy number alterations |
| | SigProfiler v1.1.24, mutational signature identification and analysis |
| | STAR 2.5.0c, transcriptome sequencing read alignment |
| | DESeq2 v1.44.0, differential gene expression |
| | limma v3.64.1, batch effect correction |
| | nSolver Analysis Software, normalisation of NanoString data |

```
Prism v10.2.1, plotting software
cola R 2.10.0, consensus clustering of transcriptome data
EPIC v1.1.7, transcriptome data deconvolution
cgpCaVEMan (v1.15.2), for mutation calling
SmartPhase (v1.2.1), for mutation calling together with cgpCaVEMan
cgpPindel (v.3.10.0), for indel calling
ASCAT (v3.1.2), for copy number calling
cnvkit 0.9.10, for copy number estimation
STATA 19.5, for survival analyses
What Is My Melanocytic Signature (WIMMS), https://wimms.tanlab.org
clusterProfiler (v4.12.6), for functional enrichment analysis
org.Hs.eg.db (3.19.1), for gene annotation

Python and R were used to perform the different bioinformatic analyses. Scripts used for analysis are at https://github.com/CGBio-Lab/Mex-acral-exomes-transcriptomes
```

For manuscripts utilizing custom algorithms or software that are central to the research but not yet described in published literature, software must be made available to editors and reviewers. We strongly encourage code deposition in a community repository (e.g. GitHub). See the Nature Portfolio guidelines for submitting code & software for further information.

## Data

Policy information about availability of data

All manuscripts must include a data availability statement. This statement should provide the following information, where applicable:
- Accession codes, unique identifiers, or web links for publicly available datasets
- A description of any restrictions on data availability
- For clinical datasets or third party data, please ensure that the statement adheres to our policy

Sequencing data are available at the European Genome-Phenome Archive (EGA). DNA sequencing data are available under ENA accession number EGAD00001015755 and RNA sequencing data under ENA accession number EGAD00001015756. The 1000 Genomes Project datasets can be dowloaded from https://www.internationalgenome.org/data. The GRCh38 reference genome can be downloaded from https://www.ncbi.nlm.nih.gov/datasets/genome/GCF_000001405.40/. Sequencing data for the Newell. et al study is available from the European Genome-Phenome Archive (EGA) under study accession EGAS00001001552 and dataset accession EGAD00001005500. Access to the data can be gained through application to the Data Access Committee for the dataset. Information on how to apply for access is available at the EGA dataset link: https://ega-archive.org/datasets/EGAD00001005500. The Cancer Genome Atlas (TCGA) Skin Cutaneous Melanoma Firehose Legacy cohort data can be downloaded from cBioPortal (https://www.cbioportal.org/). RNA sequencing data from NcNeil et al is available from GEO under accession number GSE150849.

## Research involving human participants, their data, or biological material

Policy information about studies with human participants or human data. See also policy information about sex, gender (identity/presentation), and sexual orientation and race, ethnicity and racism.

| | |
|---|---|
| Reporting on sex and gender | Samples for both sexes were included in this study. No gender (identity/presentation) or sexual orientation information is presented. |
| Reporting on race, ethnicity, or other socially relevant groupings | Throughout this manuscript, we use the term ''ancestry' to refer to genetically inferred ancestry (through whole-genome genotyping and mapping onto the 1000 Genomes superpopulations). The socioeconomic status data is self-reported, as is indicated in the manuscript. No mentions to race or ethnicity, or other social groupings of the participants, is made. |
| Population characteristics | All this information is provided in the manuscript, briefly, the median age of diagnosis was 60 years of age, 58.7% of participants were female, with the majority of tumours diagnosed Stage III (51%), ulcerated (68%), and in the feet (82.6%). Most patients (90%) had predominantly Amerindian ancestry (median 81%). |
| Recruitment | Consenting and recruitment of patients started in 2017 and ended in 2019. Because of the challenges of recruiting significant numbers of participants with AM, patients diagnosed in earlier years who were still attending follow-up clinics were recruited. This fact is taken into account in the survival analyses. |
| Ethics oversight | The protocol for sample collection was approved by the Mexican National Cancer Institute's (Instituto Nacional de Cancerología, INCan, México) Ethics and Research committees (017/041/PBI;CEI/1209/17) and the United Kingdom's National Health Services (NHS, UK) (18/EE/00076). |

Note that full information on the approval of the study protocol must also be provided in the manuscript.

## Field-specific reporting

Please select the one below that is the best fit for your research. If you are not sure, read the appropriate sections before making your selection.

☒ Life sciences     ☐ Behavioural & social sciences     ☐ Ecological, evolutionary & environmental sciences

For a reference copy of the document with all sections, see nature.com/documents/nr-reporting-summary-flat.pdf

# Life sciences study design

All studies must disclose on these points even when the disclosure is negative.

| | |
|---|---|
| Sample size | Sample size was determined by all patients that fulfilled the inclusion criteria and signed an informed consent form between 2017 and 2019. Inclusion criteria were:<br>1. Being treated at the National Cancer Institute of Mexico with a histological diagnosis of acral melanoma,<br>2. Patients which have samples available for analysis: Slides or paraffin blocks with enough material for diagnosis and sequencing, and/or the presence of primary or metastatic tumour which is accessible for sampling,<br>3. Patients that have signed the informed consent form for diagnosis, treatment and follow-up, and<br>4. Patients whose data had sufficient quality for follow-up analyses. |
| Data exclusions | Samples were excluded when not enough nucleic acids where available after extraction for DNA or RNA sequencing. Samples were also excluded if paired samples failed concordance or there was presence of contamination as stated in the text. The rationale behind this exclusion was to just include samples were somatic variant calling and RNAseq analysis could be performed. For copy number analysis, samples with discrepant results between ASCAT and Sequenza tools, or whose goodness of fit was lower than 95 were excluded to keep results only from samples with high confidence copy number estimates. |
| Replication | The analyses belonging to the transcriptional score caluclations were replicated once in an independent dataset (Newell et al, 2020). For the cell culture experiments for assessing gene expression, three biological replicates were conducted for each condition. All other analyses replication was not performed as this is a descriptive, observational study. |
| Randomization | No randomisation was done as this is an observational study. This study had mainly a descriptive approach (observational study), without any interventions. As the cancer type limited the samples available, all samples that met criteria were included. |
| Blinding | Blinding was not applicable as this study was observational with a descriptive approach. |

# Reporting for specific materials, systems and methods

We require information from authors about some types of materials, experimental systems and methods used in many studies. Here, indicate whether each material, system or method listed is relevant to your study. If you are not sure if a list item applies to your research, read the appropriate section before selecting a response.

### Materials & experimental systems

| n/a | Involved in the study |
|---|---|
| ☒ | ☐ Antibodies |
| ☒ | ☐ Eukaryotic cell lines |
| ☒ | ☐ Palaeontology and archaeology |
| ☒ | ☐ Animals and other organisms |
| ☒ | ☐ Clinical data |
| ☒ | ☐ Dual use research of concern |
| ☒ | ☐ Plants |

### Methods

| n/a | Involved in the study |
|---|---|
| ☒ | ☐ ChIP-seq |
| ☒ | ☐ Flow cytometry |
| ☒ | ☐ MRI-based neuroimaging |

## Plants

| | |
|---|---|
| Seed stocks | *Report on the source of all seed stocks or other plant material used. If applicable, state the seed stock centre and catalogue number. If plant specimens were collected from the field, describe the collection location, date and sampling procedures.* |
| Novel plant genotypes | *Describe the methods by which all novel plant genotypes were produced. This includes those generated by transgenic approaches, gene editing, chemical/radiation-based mutagenesis and hybridization. For transgenic lines, describe the transformation method, the number of independent lines analyzed and the generation upon which experiments were performed. For gene-edited lines, describe the editor used, the endogenous sequence targeted for editing, the targeting guide RNA sequence (if applicable) and how the editor was applied.* |
| Authentication | *Describe any authentication procedures for each seed stock used or novel genotype generated. Describe any experiments used to assess the effect of a mutation and, where applicable, how potential secondary effects (e.g. second site T-DNA insertions, mosiacism, off-target gene editing) were examined.* |

