## [Peer Review File · Nature]

Ancestry and somatic profile indicate acral melanoma origin and prognosis

Corresponding Author: Dr Carla Robles-Espinoza

Version 0:

Reviewer comments:

Referee #1

(Remarks to the Author)

Basurto-Lozada and collaborators evaluated 96 acral melanoma patients using several genomics approaches. This subtype of melanoma is very important in Latin America, Africa, and Asia, but it needs comprehensive studies. I want to congratulate the authors for this significant study! The study is unique due to the tumor type and, mainly, the recruited population. Using genotyping, exome, RNAseq, and Nonstring, the authors showed that acral melanoma in Mexico has a different profile from regular melanoma, with fewer BRAF mutations. The lower BRAF mutation frequency and the molecular profile closer to non-acral melanoma are known from previous work in the EUR population. Although this is undoubtedly a vital story, some points need to be addressed:

1. The central point of the paper is the influence of ancestry on AM biology. The authors showed a correlation between BRAF and KIT frequency in ancestry. However, in several parts of the results, there is no explicit mention of ancestry evaluated, such as somatic CNV, MutSig, transcriptional signature, etc. Since this is the central point of the paper, the impact of ancestry on those biological alterations should be appropriately evaluated and reported in each section.
2. As this is a retrospective dataset that has issues getting all samples for the analysis, it should be clear how many samples could be used in each section. For example, there are few mutations in the mutational signature section, but how many samples? Is this analysis powered enough?
3. The authors reported that the recruitment date was unobtainable for two participants. The recruitment date is standard information on IRB consent forms; how is this not available? Is this the date of recruitment for the study or the treatment? It should be clarified to avoid ethical problems.
4. A few recent papers evaluating AM found TERT mutation (<https://www.nature.com/articles/s41467-024-50233-z>). Did the authors check for TERT mutation and other molecular features that were recently published?
5. One of the significant points here is understanding AM molecular features and Latin American molecular features. The lack of samples from other ancestries (and locations) limits some conclusions. At least, the authors should use literature data to compare previous results on other ancestries. What is unique about AM in Mexico that makes the findings more interesting?
6. The results in the Mutation Signature section should be more descriptive and reach a clear conclusion. For example, about 1/4 of the samples have UV mutational signatures. This result can be seen in the Sup Fig 5, but what are these samples? UV is not expected here. How much of the mutations can be attributed to UV? Any specific trait for this? The same is true for other SBSs.
7. The paper defines triple and quadruple wild-type tumors on page #5. However, A) this needs to be clarified: What is the triple and the quadruple wt: N/H/KRAS, BRAF, and NF1? TP53, HRAS, and KRAS? Also, if I understood the rest of the paper and the analysis, these classifications would not be used anymore on the paper. If so, this needs to be more helpful and should be removed. If those are indeed used, this must be better explicated where.
8. The authors used two DNA sequencing pipelines: low-DNA and standard one. However, just one was found on the methods. What was the difference between them? Which is the report on the paper?
9. I do not understand the following phrase: "For recurrence, all participants with a recurrence prior to consent were excluded from the analysis and treated as a prospective cohort starting at recruitment." This study is not an intervention trial to set a time for initial recruitment (or, in other words, the start of treatment) as a zero mark. Recurrence before the recruitment should be used in the survival analysis since the groups are classified on a genetic basis. There is no way to treat this set as a "prospective cohort." This statement should not be stated to avoid confusion.

10. Regarding the survival analysis and the other analysis, it is essential to point out the limitations of the paper. As a retrospective oncology study, survival bias has a significant influence. This point is essential since the authors wrote that patients have been followed up for over 10 years. The aggressive ones are probably dead, unfortunately, at this point and could not participate in the study. This bias is probably the reason why most of the patients were classified as Stage III.

11. The authors compare non-acral to acral melanoma results in the middle of the results. What samples are those? Are there non-acral melanoma samples on the analysis? I assume this is the Utah dataset. Is this right? It should be clarified. Those samples, although from an area with Mexican immigration, do not have genotyping to evaluate ancestry information. This information limits the results of that analysis since the central point of the paper is the relation of the findings with ancestry.

12. All datasets, the Mexican and the Utah, should be presented in the 'Patient recruitment and sample collection' section to better clarify the sets.

13. The normal adjacent tissue sample and saliva were used in the study according to the methods. But when? Was it for genotyping? Exome? It should be similar, but did the authors check differences or preferences for ancestry regarding those methods?

14. To identify SNV and driver genes, the authors gave preference for the primary tumor but also used the Metz. Are the TMB mutation frequencies similar? Did the authors use these parameters on the regressions for BRAF, survival, and others?

15. In the discussion, the authors stated that the relation between gene mutation and the worst prognosis is unknown. Are the authors referring to AM? For non-acral melanoma, this relation has been known for several years (see below a few quick examples – none of those are my papers), and it also includes AM patients for some papers. The authors should discuss their results and compare them with the literature.

<https://pubmed.ncbi.nlm.nih.gov/21343559/>
<https://pubmed.ncbi.nlm.nih.gov/21606968/>
<https://pmc.ncbi.nlm.nih.gov/articles/PMC9431958/>

(Remarks on code availability)

Referee #2

(Remarks to the Author)

In this study, the authors analyzed 128 acral melanoma (AM) samples from 96 Mexican patients through genotyping, exome sequencing, and gene expression profiling. This is one of the largest genetic analyses of AM. By specifically analyzing melanomas from patients of Latin American ancestry, this study improves our understanding of the genetic basis of the most common melanoma histological subtype found in low and middle-income countries, which has been poorly addressed in previous large-scale sequencing studies. The authors should be commended for this effort.

Regarding novelty, although this study is a tour de force effort in its scale and provides an important resource for the melanoma/cancer genetics fields, the major finding of the increased frequency of BRAF mutations in patients of European ancestry has been previously described by Yeh et al., 2019. The authors do identify new low-frequency significantly mutated genes (e.g. SPHKAP, POU3F3, RDH5, MED12), but the authors provide little evidence that these genes play a role in AM. Finally, in the analysis that demonstrates BRAF mutated AM exhibit a more 'cutaneous melanoma-like' transcriptome suggesting a potential distinct melanocyte cell of origin compared to other AMs, I do have some concerns on this specific analysis (described in more detail below).

I have no major concerns regarding the statistics employed in the analysis, and the code related to this study is well-organized.

Comments/suggested edits:

1. Ancestry and clinical characteristics of Mexican AM patients

1.1 Do the authors have information related to pigmentation of the biospecimens to be included in clinical data file, and does this differ depending on patient ancestry?

2. Genomic profiling of AM samples identifies correlations of ancestry and age with somatic alterations

2.1. In Figure 1a, how were the genes in the oncoplot selected? In Supplementary Figure 2 for quadruple wild-type patients, were these the only genes with impactful mutations? If there are others with similar mutation frequency, what was the rationale/selection criteria? If mutation frequency was used as the selection criteria and not significantly mutated genes, is it possible that genes with SNVs highlighted in the quadruple WT are enriched for passenger events and that CNAs primarily drive the quadruple WT tumors instead? Can the authors please provide the complete output of the dNdScv analysis?

2.2. Of particular importance, the authors have nominated several novel driver mutations in SPHKAP, POU3F3, RDH5, MED12 in this cohort. How confident are the authors that these are oncogenes or tumor suppressors in melanoma? Are the loss of function mutations in SPHKAP or POU3F3 associated with loss of heterozygosity, or decrease in its mRNA expression? Are any of these genes mutated other cancers? To my knowledge, I don't think they have been reported in any other acral melanoma or pan-cancer mutation significance analysis (e.g. Bailey et al., 2018, Cell

<https://doi.org/10.1016/j.cell.2018.02.060>). Is there any functional validation of these genes in large screening studies (e.g. DepMap)? As these findings are part of the more novel aspects of this study, providing supporting evidence would strengthen the manuscript.

2.3. It would be useful for the readers to know what are the specific mutation types that are found for each highlighted mutated gene (for example what are the number of C>T transitions that are found for the highlighted genes in Figure 1a or Supplementary Figure 2?). It would also be useful to compare the frequency of mutations in these genes in melanomas from sun-exposed skin or other acral melanoma studies.

2.4. Figure 1c. The authors trained a logistic regression model to predict the mutation preferences in each ancestry with the control of potential confounders. Can the authors provide a more comprehensive evaluation of the model's performance? For example, how well can the model perform when testing the data with a leave-one-out cross-validation?

2.5 Figure 1d. The authors provided data that NRAS mutations were strongly associated with younger age at diagnosis. However, one concern is that the correlation might be confounded by the ulceration status of the tumor. NRAS mutated tumors appear to have less ulceration compared with the samples harboring other mutations (shown in Fig 1a). Can the authors prove that the association is independent from ulceration status?

2.6. General methods: Can the authors clarify for the somatic SNV calling, why they used both Mutect and Mutect2 before applying the filter of a minimum of two callers. Does this bias the variant calling to Mutect1/2? Why not just use Mutect2 if it is an improvement over mutect1 and take variants called by both Mutect2 and varscan? Mutect also does indel calling. Why use Strelka2 for indels and ignore Mutect2 calls? Was indel realignment performed?

3. Somatic copy number landscape of AM samples identifies correlations with somatic alterations

3.1. Can the authors comment on their confidence in the Sequenza tool for copy number alterations (CNAs), cellularity and ploidy? Have the authors compared their results to other tools (i.e. PURPLE)?

3.2. Figure 2b. The figure for the copy number variations is not entirely clear. As stated by the authors, more amplifications than deletions were observed and wondering whether these values are relative to tumor ploidy or relative to 2? A plot with higher genomic resolution showing the frequency of alteration at each genomic locus in each subtype might be easier to visualize than the heatmap (for example, something similar to the Figure 4 from the Newell et al., 2020 study <https://doi.org/10.1038/s41467-020-18988-3>).

3.3. Previous studies have shown that AM have a lower SNV burden and higher numbers of structural rearrangements and focal copy number events compared to cutaneous melanoma (e.g. Turajlic et al., 2012; Hayward et al. 2017; Newell et al., 2020; Meng Wang et al., 2024). Can the authors comment or perform a comparative analysis to provide readership an understanding of differences between the cohort of Mexican patients?

3.4 Can the authors please provide a more detailed explanation of GCS scores to be included in the manuscript?

3.5 On line 207: Regarding the suggestion that BRAF/NRAS-mutated might be of different aetiology because they have distinct SCNA profiles: What if the cell of origin is the same but the evolutionary trajectory of tumors with these mutations tend to be different?

5. BRAF-mutated acral melanomas exhibit a transcriptional signature more characteristic of non-acral cutaneous melanomas

5.1. A critical insight/suggestion from this work is that BRAF mutant acral melanomas have a different cell of origin from BRAF/NRAS WT acral melanomas based on the finding that "BRAF-mutated acral melanomas exhibit a transcriptional signature more characteristic of non-acral cutaneous melanomas". To arrive at this conclusion, my understanding is the authors first derived an acral vs. non-acral cutaneous melanoma score by measuring the expression of genes in acral and non-acral cutaneous melanomas in a separate cohort of 20 patients. The authors looked at a limited set of genes determined previously to be differentially expressed between glabrous and non-glabrous melanocytes. Of these genes, using 20 genes most discriminative of acral vs. non-acral melanomas, determined by principal component analysis, they computed multiplicative score in 80 patients from their cohort (the acral:cutaneous (A:C) ratio) and determined that the score separates BRAF mutant acral melanoma from BRAF/NRAS WT acral melanomas (Fig 3C). They validated this finding in a separate cohort of 63 patients from Newell et al., 2020 (Fig. 3D).

The authors suggest that because this score has the power to separate acral from non-acral cutaneous melanomas, its ability to separate BRAF mutant acral melanomas from BRAF/NRAS WT acral melanomas indicates that BRAF mutant acral melanomas likely have a different cell of origin, similar to melanocytes enriched in limbs (c-type). Whereas that authors suggest that BRAF/NRAS WT acral melanomas may originate from v-type melanocytes enriched in volar regions.

A major caveat is that the genes used to produce this score may simply be distinguishing BRAF mutant melanomas from BRAF WT melanomas, as opposed to cell of origin. This is because the genes were initially selected based on their ability to distinguish non-acral cutaneous melanomas (which are highly enriched in BRAF mutants) from acral cutaneous melanomas (which are depleted of BRAF mutants). Previous studies have shown that BRAF mutations can be associated

with their own transcriptional signature. Furthermore, BRAF activating mutations are often associated with chromosome 7 amplification. Therefore, in cells where BRAF is mutated, many other genes on chromosome 7 including BRAF exhibit increase expression, along with altered expression of the downstream effectors of those genes.

The authors acknowledge this caveat, yet still suggest that their signature is distinguishing acral melanomas of different cell origin as opposed to simply distinguishing BRAF mutant and WT melanomas of the same cell of origin. To help address this issue, can the authors demonstrate that their signature is unable to separate BRAF mutants from BRAF/NRAS WTs in non-acral cutaneous melanomas, which the authors suggest originate from c-type melanocytes? Furthermore, the differences found between BRAF/NRAS mutated group and WT group can be confounded by many factors such as sample purity. Can the authors find a significant difference in A:C ratio in the CCLE dataset or RNA seq data from CM and AM cell lines?

5.2 Figure 3a, please add the percentage of explained variances based on PC2

5.3 Figure 3d. The authors show 3 mutated subtypes (WT, BRAF and NRAS) in their cohort (shown in fig 3c), but why is the NRAS-mutated subgroup excluded in the Newell et al. cohort?

6. Transcriptional landscape of AM tumors identifies three subgroups with distinct clinical and prognostic characteristics

6.1 Figure 4a. To better understand the transcriptome differences, can the author include some differentially expressed marker genes on the figure for each cluster and add sample purity as one of the column annotations?

6.2 The authors provide differences in cell type proportions in the TMEs shown in Fig 4c-e across the three clusters. Can the authors provide a supplementary figure to show the differences in all the inferred cell types among the three clusters?

6.3 To better understand the distinct features of each cluster, can authors include the pathway enrichment results of the DEGs (differentially expressed genes) in each cluster?

6.4. Cluster 1 appears to be related to lower-stage tumors, which may explain the observed improved prognosis, lower ulceration and Breslow thickness and could explain differences in expression. Can the authors explain why this was not a confounding variable? Furthermore, are there sex differences in acral melanoma? 9 out of 11 patients in Cluster 1 are female. Authors may wish to discuss this imbalance as compared to Cluster 2 and 3.

7. Somatic and gene expression profile influence recurrence-free survival

7.1 The authors state that "Those with a driver mutation (BRAF, NRAS, KIT, NF1 or multihit) had a significantly higher probability of having earlier recurrences (Log-rank test P-value < 0.05) (Supplementary Table 8, Figure 5a), with NF1 mutations likely having a stronger effect (Supplementary Table 9, Figure 5b)." Can the authors validate the same trend (MAPK pathway driver mutations leading to a higher risk of recurrence) in a larger cohort of acral melanoma or in non-volar cutaneous melanoma? Additionally, is this trend independent of their ancestries?

(Remarks on code availability)

code related to this study is well-organized

Referee #3

(Remarks to the Author)

In this manuscript, Basurto-Lozado and colleagues performed a genomic and transcriptomic analysis of acral melanomas from Mexico. Acral melanoma is the most common type of melanoma in skin of color and yet previous studies have largely overlooked Latin American samples. As such, this study addresses an important gap in knowledge.

Acral melanomas are typically diagnosed based on their anatomic location – the non-hair bearing skin of the foot soles, palms of hand, or nail bed. The main finding in this manuscript is that two subtypes of melanoma appear on these body sites – a type of melanoma with BRAF mutations, similar to cutaneous melanoma from the trunk, and a type of melanoma with more classical features of acral lentiginous melanoma. The authors propose that these melanomas arise from distinct sets of melanocytes. Many of these ideas have been proposed and partially shown by others. The strength of the present study is the unique population, in which participants displayed a range of genetic admixture – some with more European descent and some predominantly of AmerIndean descent. This unique cohort allowed the investigators to observe correlations between somatic mutation profiles and ancestry. For instance, the BRAF-mutant subtype of melanoma was predominantly found in participants of European descent. Overall, the manuscript cements the idea that melanomas on volar skin are heterogeneous, where people of European descent sometimes get UV-radiation-induced melanomas, akin to cutaneous melanoma, while in skin of color, the non-UV driven form of acral lentiginous melanoma dominates.

Several technical concerns were noted, which will, once addressed, improve reporting and hopefully strengthen/broaden the main conclusions.

1. In Figure 1a, the authors listed only 59 samples out of 96, it would be informative to show all cases at least once in the main figure. What is the rationale for listing genes, such as ZNF793, KRTAP4-16, and CSN2? These genes have a very low frequency (e.g. 1%) of mutations and unknown roles in melanoma progression. Moreover, these genes were not mentioned

anywhere in the paper.

2. What is the purity and ploidy of each sample? Although the manuscript mentioned that tumors with purity < 0.2 were excluded, in 8 tumors, less than 10 SNVs were detected, and we could not find CNA scores for these tumors. Could these tumors have low purity, thus affecting mutation/CNA calling? More broadly, is tumor purity associated with the number of SNVs across the study?

3. It is unclear if the authors scrutinized copy number calls for focal events, such as homozygous deletions and amplifications. Focal copy number alterations are a cardinal feature of acral melanoma. Specific genes to focus upon are discussed below.

a. NF1 can be homozygously deleted in acral melanomas, which has the same effect as truncating mutations. Therefore, tumors with homozygous deletion of NF1 should also be considered in the NF1-mutated group. For some of the homozygous deletions, the deeply deleted regions can be rather narrow, affecting only a few exons. We recommend scrutinizing pre-segment level data to ensure that these alterations are not missed.

b. What is the status of TERT (promoter mutations, amplification, etc.) in the cohort? This gene is a well-known melanoma driver gene but was not mentioned in this study.

c. For CDKN2A, the authors mentioned that 66% of samples showed deletion. Are these all homozygous deletions?

4. When analyzing genetic subtypes of acral melanoma, it would be informative to separate BRAF V600E with other BRAF mutations as well as the different NRAS hotspot mutations (G12 and Q61 codons).

a. This is especially important in the setting of BRAF, where there are three classes of mutations, each associated with different subtypes of melanoma. In the present manuscript, on Page 7 line 209, the authors state "it has been previously postulated that BRAF-mutated acral melanomas might be more biologically like melanomas from non-acral sites than to other acral melanomas". This is likely true for BRAF V600E mutation but not for other BRAF mutations.

5. It was recently reported that during the evolution of acral melanoma, MAPK pathway mutations can occur after tumor initiation. For the cases in which multiple tumor samples were available, did the authors observe heterogeneous/subclonal NRAS/KIT/BRAF/NF1 mutations?

6. The use of TMB (tumor mutation burden) is not consistent throughout the manuscript. It was used to indicate both "mutations per megabase" and total mutation counts (such as in Figure S4).

7. The number of indels per tumor appears extraordinarily high, on par with the number of somatic SNVs. There is a concern that these calls include numerous false positives. In our experience with Strelka, additional filters are needed, beyond the calls provided directly by the software.

8. For the clustering in Figure 4, do tumor purities differ across different clusters?

9. How was tumor ploidy estimated? And why was a cutoff of 3.6 used to distinguish genome doubling events?

(Remarks on code availability)

No concerns noted

Referee #4

(Remarks to the Author)

Nature is committed to facilitate training in peer-review and to ensure that everyone involved in our peer review process is appropriately recognised. This reviewer co-reviewed one of the listed reports.

(Remarks on code availability)

Version 1:

Reviewer comments:

Referee #1

(Remarks to the Author)

The authors answered all my questions and concerns!

The new analysis and data reinforced their findings and showed the importance of their work.

Congratulations on the amazing work.

Referee #2

(Remarks to the Author)

The authors have made a strong and constructive effort to address all points raised in the initial review. Notably, they re-called their variants using an additional caller and re-ran their driver analysis. This resulted in substantial changes to the list

of significantly mutated genes (SMGs). Rather than overlooking these discrepancies, the authors revised the manuscript accordingly and curated the driver list with greater care, incorporating both literature support and orthogonal evidence. Overall, the responses are thorough and transparent. The revisions strengthen the rigor and reliability of the findings included in this study, and I find the authors' approach appropriate and commendable. I have no further concerns. The authors should be congratulated for this tour de force study.

Referee #3

(Remarks to the Author)

We thank the authors for their revision, which was generally responsive to our critique. Overall, we believe the main conclusions of the manuscript sound, particularly that there are differences in the biological subtypes of melanoma on acral body sites associated with ancestry. However, there are a couple unresolved issues, as discussed below.

In our previous critique, we suggested that the authors inspect the pre-segmented data to identify focal deletions. In response, they manually scrutinized alignments in Integrative Genomics Viewer (IGV) – this is not exactly what we had in mind, and we do not believe this strategy is a reliable way to infer copy number. Please have a look at supplemental figure 10 of PMID: 35706047 for examples of focal deletions of NF1 in acral melanoma. As you will see, the figure shows bin level data (grey data points) as well as copy number segments (yellow lines). In some tumors, there was a focal deletion that the segmentation algorithm did not pick up, but when viewed at the bin level, and armed with the biological knowledge that these are bona fide tumor suppressors, it is reasonable to call them focally deleted. Focal deletions of CDKN2A, NF1, and other tumor suppressors are not uncommon in acral melanoma, likely due to the distinct mutagenic forces that operate on this tumor. While Basurto-Lozada et. al. did not use the same software as PMID 35706047 to infer copy number, hopefully, bin-level copy number estimates are available that they can scrutinize. We anticipate that the frequency of deep deletions affecting tumor suppressors will increase after such a review. Indeed, the authors cite the Newell and Wang studies as examples of other studies that find a similar frequency of homozygous CDKN2A deletions, however, we disagree with the statement because those studies actually found much higher frequencies of homozygous CDKN2A deletion (around 30-40%).

Thank you for confirming that tumor purity was lower in Cluster 1 tumors and updating a figure to show tumor purity. Do the authors believe this might explain why tumors with no driver mutations tend to have a better prognosis? It is more difficult to detect somatic mutations in tumors with low neoplastic cell content, which likely explains the association with good prognosis. We noticed that reviewer 2 also picked up on this association, referring to it as a “confounding variable”. In the point-by-point rebuttal, the authors seemed to agree with reviewer 2 that the “good prognosis” tumors are likely earlier stage with lower tumor cell content, yet these limitations are not sufficiently reflected in the manuscript itself. It is probably true that a tumor with fewer driver mutations will have better prognosis, but given all the technical challenges to detecting driver mutations in tumors with low neoplastic cell content, we advise the authors to tone down this conclusion.

As an extension of the point above, we encourage the authors to revisit whether all tumors should be included in this study. As an extreme example, we saw one tumor (PD51948a) with no somatic point mutations, no somatic copy number alterations, and it was predicted to have a tumor cellularity of 100%. While acral melanomas have low mutation burdens, it seems implausible for a tumor to have no somatic alterations – not even some passenger mutations.

Minor points

We suggest altering the language when the authors say KIT mutations are “lost” in metastases. It is unlikely that the mutation was “lost”. A more plausible scenario is that cells seeded metastatic sites before the KIT mutation occurred and underwent a clonal sweep in the primary tumor. To be sure, the authors do suggest, later in the paragraph, that metastases are likely seeded early, but we would advise against implying reversion of the KIT mutation.

We do not believe a full paragraph in the discussion is needed to acknowledge limitations of studying FFPE tissues. It is true that formalin fixation fragments and damages DNA, and in theory, FFPE-induced damage can be misread as mutations in sequencing data. However, in practice, FFPE-induced artifacts tend to be randomly distributed across the genome and therefore have low allele frequency. When coverage and tumor cellularity are sufficiently high, FFPE-induced artifacts are rare. It is ok to acknowledge the challenges of studying FFPE material, but this could be moved to the methods and does not require precious real estate in the discussion section.

We suggest removing the word “frequent” in this sentence from the abstract. “we found fewer frequent mutations in classical driver genes such as BRAF, NRAS or NF1.”

In the authors response, they said that they homogenized their use of tumor mutation burden, but there still is at least one example where the term is used inconsistently. In the text of the manuscript, the authors refer to tumor mutation burden as mutations per megabase, but in figure 2d, they appear to plot absolute mutation counts (unless there are some tumors with 120 mutations/Mb!). It is less important to be consistent, but most important to define the units whenever the term is used.

Referee #4

(Remarks to the Author)

I co-reviewed this manuscript with one of the reviewers who provided the listed reports.

Version 2:

Reviewer comments:

Referee #3

(Remarks to the Author)

The authors have done an excellent job in addressing our final points and are congratulated on a well-executed study!

Referee #4

(Remarks to the Author)

I co-reviewed this manuscript with one of the reviewers who provided the listed reports.

Dear Dr. Burgess and Reviewers,

We are very grateful for the time and consideration that you have put into our manuscript. We are especially thankful for your appreciation of this work and consideration of the difficulties of undertaking such genomic studies of a rare cancer in LMIC countries such as Mexico. In response to the Reviewers' comments, we have replicated large parts of our analyses, including a manual re-evaluation of all cases included in this manuscript. We also re-performed mutation calling (now using the CaVEMan mutation caller instead of Mutect v1), sample purity and copy number estimation (now using ASCAT together with Sequenza), and mutational signature analysis with these more accurate and curated calls. We have also performed additional studies, including an analysis of the relationships of ancestry proportions with copy number aberrations, *TERT* promoter region sequencing, analyses of samples from other cohorts and other bioinformatics studies to investigate the cell of origin hypothesis. Reassuringly, our conclusions stay the same, and have been enhanced by this work. We believe that our dataset is now cleaner and more accurate. We have also revised parts of the Discussion and have clarified several points throughout the manuscript to reflect comments made by the Reviewers.

Below we have provided a point-by-point response to the Reviewers' comments. We hope that our manuscript is suitable to move forward toward publication.

Referee #1 (Remarks to the Author):

Basurto-Lozada and collaborators evaluated 96 acral melanoma patients using several genomics approaches. This subtype of melanoma is very important in Latin America, Africa, and Asia, but it needs comprehensive studies. I want to congratulate the authors for this significant study! The study is unique due to the tumor type and, mainly, the recruited population. Using genotyping, exome, RNAseq, and Nonstring, the authors showed that acral melanoma in Mexico has a different profile from regular melanoma, with fewer BRAF mutations. The lower BRAF mutation frequency and the molecular profile closer to non-acral melanoma are known from previous work in the EUR population. Although this is undoubtedly a vital story, some points need to be addressed:

We are delighted with comments from Referee 1 and we thank them for their helpful comments. Below we have responded to each point they have made and detailed how we have addressed them in the revised manuscript.

1. The central point of the paper is the influence of ancestry on AM biology. The authors showed a correlation between BRAF and KIT frequency in ancestry. However, in several parts of the results, there is no explicit mention of ancestry evaluated, such as somatic CNV, MutSig, etc. Since this is the central point of the paper, the impact of ancestry on those biological alterations should be appropriately evaluated and reported in each section.

R. We thank the Reviewer for this comment. Indeed, we did find a link between *BRAF* mutation and ancestry. To answer this question, we have tried testing the relationship between specific CNV sites (all significant sites outputted by GISTIC) and ancestry proportions (specifically, European ancestry), but we did not find any statistically significant association (**Figure R1**).

Likewise, we tested the relationship of ancestry with mutational signature proportion, again finding no significance (**Table R1**). We also attempted to control for ancestry proportions in the survival analyses, again, without finding any significant effects. However, we now make this clear after each section of the manuscript where these analyses are performed (lines 510-511, 625-626, 860, 889 of the track-changes manuscript). We have also controlled for ancestry now in the analyses testing the association between mutational status and survival.

Figure R1. Relationship of ancestry with copy number alterations (CNAs). Logistic regression model controlling for age, sex, and TMB was fitted to predict the presence or absence of a copy number alteration in the AM samples using the inferred ADMIXTURE cluster related to the European ancestry component. Log odds estimate and confidence intervals are depicted.

Description: Influence of ancestry proportions (EUR, Q5) on mutational signature counts lm(formula = "Q5 ~ SBS1 + SBS5 + SBS40a", data = mut_sign)					
Min	1Q	Median	3Q	Max	
-0.22978	-0.15652	-0.04533	0.15756	0.46263	
Term	Estimate	Std. Error	t value	p-value	Significance
(Intercept)	0.2087	0.03572	5.843	1.28e-07	***
SBS1	-0.004129	0.003647	-1.132	0.261	
SBS5	0.00006992	0.0008427	0.083	0.934	
SBS40a	0.00026	0.0007195	0.361	0.719	

Residual standard error: 0.1776 on 74 degrees of freedom
Multiple R-squared: 0.01888, Adjusted R-squared: -0.02089
F-statistic: 0.4747 on 3 and 74 DF, p-value: 0.7008

Description: Influence of ancestry proportions (EUR, Q5) on mutational signature proportions lm(formula = "Q5 ~ SBS1_prop + SBS5_prop", data = mut_sign)					
Min	1Q	Median	3Q	Max	
-0.22402	-0.15501	-0.02657	0.14308	0.45594	
Term	Estimate	Std. Error	t value	p-value	Significance
(Intercept)	0.218282	0.029517	7.395	1.67e-10	***
SBS1_prop	-0.191156	0.136148	-1.404	0.164	
SBS5_prop	0.005746	0.046526	0.124	0.902	

Residual standard error: 0.1758 on 75 degrees of freedom
Multiple R-squared: 0.0257, Adjusted R-squared: -0.0002801
F-statistic: 0.9892 on 2 and 75 DF, p-value: 0.3767

Table R1. Relationship of ancestry with mutational signature proportion. Output of a logistic regression model was fitted to predict the presence or absence of a signature (SBS1, SBS5, SBS40a) in the AM samples using the inferred ADMIXTURE cluster related to the European ancestry component. Upper table: Absolute mutation counts, lower table: proportion of signature.

2. As this is a retrospective dataset that has issues getting all samples for the analysis, it should be clear how many samples could be used in each section. For example, there are few mutations in the mutational signature section, but how many samples? Is this analysis powered enough?

R. We agree with the Reviewer that the number of samples used in each section was confusing, as this project entailed a number of different analyses that required different quality

filters. We have now added **Supplementary Figure 14**, with a flowchart describing the different steps taken in each analysis, and made clear the number of samples in each section in the text. We agree that the lower number of mutations may limit the signatures that we can identify. We have indicated this in lines 624-627 of the track-changes manuscript.

3. The authors reported that the recruitment date was unobtainable for two participants. The recruitment date is standard information on IRB consent forms; how is this not available? Is this the date of recruitment for the study or the treatment? It should be clarified to avoid ethical problems.

R. All patients signed informed consent in keeping with our IRB. Of note, INCAN, as a medical facility of the Mexican Secretariat of Health, rigidly follows the Helsinki guidelines. During an office move the two consent forms for these two patients could not be located, however, we now have the consent forms for both patients and have added the missing recruitment dates to the table and the analyses.

4. A few recent papers evaluating AM found *TERT* mutation (<https://www.nature.com/articles/s41467-024-50233-z>). Did the authors check for *TERT* mutation and other molecular features that were recently published?

R. The *TERT* mutation was not originally included in our exome dataset as the mutation is usually in the promoter. However, in response to this comment, we went back to our stock DNA and successfully amplified and sequenced via Sanger sequencing the *TERT* promoter region (including the -124 and -146 positions) in 76 samples belonging to 64 patients. We found that six out of 64 patients carry the -124 mutation (9.3%) and two out of 59 patients for which the -146 position was successfully amplified carry a mutation in this position (3.4%). In total, this means that for this population (as no sample carries both mutations), we estimate that 10.5% of patients have an activating *TERT* promoter mutation. This is in line with what has been described in other studies, specifically Wang *et al*¹ (~13.5%) and Newell, *et al*² (10.3%). We also found that, whenever a *TERT* mutation is found in a sample, it is found in all samples belonging to the same patient, supporting Wang *et al*'s previous finding that these lesions arise early in acral melanoma evolution. This information is now included in the manuscript (lines 357-364 of the track changes manuscript).

5. One of the significant points here is understanding AM molecular features and Latin American molecular features. The lack of samples from other ancestries (and locations) limits some conclusions. At least, the authors should use literature data to compare previous results on other ancestries. What is unique about AM in Mexico that makes the findings more interesting?

R. We thank the Reviewer for this pertinent suggestion and agree that a comparison with data from patients from other geographical regions and ancestries will enhance conclusions. We have reviewed other studies that report genome, exome or targeted sequencing for samples from acral melanoma patients (specifically, Yeh *et al*, 2019³, Newell *et al*, 2020², Farshidfar *et al*, 2022⁴, Birkealv *et al*, 2023⁵, Liu *et al*, 2024⁶, and Wang *et al*, 2024¹), and we believe that the Mexican population is unique or at least different to the majority of these studies because:

1. The fraction of activating *BRAF* mutations is lower than in the majority of these studies, probably due to the positive relationship between *BRAF* mutation and European ancestry. Interestingly, and in keeping with this hypothesis, the fraction of *BRAF*-mutated samples in the Liu *et al* study, which profiled Chinese patients, is lower than in our study, and is comparable to the Wang *et al* study, which profiled patients from Bolivia, China and Japan. We tried to replicate the ancestry analysis in the Newell *et al*. study, but most patients in that study are of European ancestry, and this analysis is therefore uninformative.
2. The ancestry composition of the patients in our study includes a significant proportion of Native American ancestry, which is severely underrepresented in published cancer genomics studies and permits the identification of relationships of specific ancestries with somatic characteristics.
3. Access to treatment. The most common treatment in our cohort was chemotherapeutic drugs only (26% of patients, including dacarbazine, carboplatin, and/or paclitaxel), with only four patients (4.3%) being able to access immunotherapy (nivolumab and/or ipilimumab). This is quite different than what is reported in other cohorts (Newell *et al* include only two patients with chemotherapy only, whereas more than 20% of patients were able to access immunotherapy and/or *BRAF/MEKi* targeted inhibitors).

We were also able to confirm that clinical characteristics such as stage at diagnosis and ulceration rates are comparable across cohorts. We believe that the profiling of samples from different geographical locations and cohorts is essential for understanding cancer development and is crucial for detecting novel risk associations. We completely agree with the Reviewer and have added these observations to the Results (lines 435-442 of the track changes manuscript). We have also added a panel in Figure 1 (**Figure 1d**) showing the comparison between this study and these other studies.

Figure R3. Comparison of genomic profile in acral vs cutaneous melanoma according to different published studies. The number of samples in each study are indicated, along with the sample classification.

6. The results in the Mutation Signature section should be more descriptive and reach a clear conclusion. For example, about 1/4 of the samples have UV mutational signatures. This

results can be seen in the Sup Fig 5, but what are these samples? UV is not expected here. How much of the mutations can be attributed to UV? Any specific trait for this? The same is true for other SBSs.

R. We agree with the Reviewer, and they are correct that UV signature was not expected to be found in these samples. In our revision of the manuscript in response to the Reviewers' comments, we re-performed the variant calling, keeping the '2 out of 3 callers' approach but swapping Mutect for Caveman. We chose this mutation caller as it was part of the main analysis pipeline in the Pan-Cancer Analysis of Whole Genomes (PCAWG) project, and was developed to be highly specific^{7,8}. We then re-performed the mutational signature analysis. We believe we have improved our calling, as no artifactual signatures were found, and the indel count is now much lower than we originally reported. With this new analysis, we no longer see the UV signatures, suggesting that the mutation call set is much cleaner.

7. The paper defines triple and quadruple wild-type tumors on page #5. However, A) this needs to be clarified: What is the triple and the quadruple wt: N/H/KRAS, BRAF, and NF1? TP53, HRAS, and KRAS? Also, if I understood the rest of the paper and the analysis, these classifications would not be used anymore on the paper. If so, this needs to be more helpful and should be removed. If those are indeed used, this must be better explicated where.

R. We thank the Reviewer for raising the fact that this nomenclature was confusing, and indeed our oversight in naming the same group of tumours differently in different parts of the manuscript. We were referring to *BRAF/NRAS/NF1/KIT*-wild type tumours as quadruple-wild type (meaning, those that do not carry any mutations in these genes). We have now clearly defined this characteristic (lines 371-372 of the track changes manuscript) and have renamed tumours in the 'Other' group as 'Quadruple wild-type' (**Figures 1, 2, 4 and 5, Supplementary Figures 2, 3, 4, 5, 6, 7, and related analyses**).

8. The authors used two DNA sequencing pipelines: low-DNA and standard one. However, just one was found on the methods. What was the difference between them? Which is the report on the paper?

R. The Reviewer is right, and we are thankful to them for pointing this out. We wrote in the Methods what we originally did, which was to amplify and sequence a larger cohort of samples by both pipelines. However, after quality control, only samples that were processed by the standard DNA pipeline remained for further analyses. We apologise for the confusion, and have now removed the reference to the low DNA pipeline from the Methods as it's not relevant to the current study.

9. I do not understand the following phrase: "For recurrence, all participants with a recurrence prior to consent were excluded from the analysis and treated as a prospective cohort starting at recruitment." This study is not an intervention trial to set a time for initial recruitment (or, in other words, the start of treatment) as a zero mark. Recurrence before the recruitment should be used in the survival analysis since the groups are classified on a genetic basis. There is no way to treat this set as a "prospective cohort." This statement should not be stated to avoid confusion.

R. We have now removed any reference to this cohort as “prospective”. For these individuals who had a recurrence prior to recruitment, the recruitment happened because of the re-connection with the medical services after the recurrence was diagnosed. Essentially, then the recruitment happened because of the recurrence, so our approach is to consider this cohort of patients as having been recruited at the time of recurrence (as recurrence is a major risk factor for prognosis) and then following them from that point as would happen in a prospective trial. We agree that, ideally, a prospective analysis would commence with recruitment date being close to diagnosis date, and then followed up. However, due to the low prevalence of this disease, and the fact that we are working in a resource-challenging medical environment, this was not feasible in our cohort. However, in response to this comment, we have adjusted for the time between diagnosis and recruitment data, finding that this makes no difference to our results, and have considered only tumours diagnosed closest to the time of recruitment, again, finding qualitatively and quantitatively similar results (See lines 1084-1091 in the track changes manuscript). So, we believe this is a reliable result, but we have pointed out these concerns in the Discussion.

10. Regarding the survival analysis and the other analysis, it is essential to point out the limitations of the paper. As a retrospective oncology study, survival bias has a significant influence. This point is essential since the authors wrote that patients have been followed up for over 10 years. The aggressive ones are probably dead, unfortunately, at this point and could not participate in the study. This bias is probably the reason why most of the patients were classified as Stage III.

R. We agree with the Reviewer and we think they make a quite reasonable point. We have acknowledged the limitations of our study in the Discussion, lines 1071-1091 of the track changes manuscript.

11. The authors compare non-acral to acral melanoma results in the middle of the results. What samples are those? Are there non-acral melanoma samples on the analysis? I assume this is the Utah dataset. Is this right? It should be clarified. Those samples, although from an area with Mexican immigration, do not have genotyping to evaluate ancestry information. This information limits the results of that analysis since the central point of the paper is the relation of the findings with ancestry.

R. We apologise for the confusion. These samples are indeed from the Utah dataset. We were not assuming that these samples had any Amerindian ancestry; they were used only to derive the transcriptional score (comparing to non-acral melanoma samples collected by the same team, led by R. Judson-Torres) that we subsequently used to distinguish between acral and non-acral melanomas. We have now specified this point in line 641 of the track changes document.

12. All datasets, the Mexican and the Utah, should be presented in the ‘Patient recruitment and sample collection’ section to better clarify the sets.

R. We have now added the information on the Utah cohort to the ‘Patient recruitment and sample collection’ section (lines 1107-1108 of the track changes document).

13. The normal adjacent tissue sample and saliva were used in the study according to the methods. But when? Was it for genotyping? Exome? It should be similar, but did the authors check differences or preferences for ancestry regarding those methods?

R. We appreciate the Reviewer raising the point that this information was unclear. We have now clarified this in the Methods, lines 1125-1126 of the track-changes document. To explain what happened, when we started this study, we were collecting saliva as the normal/control tissue as that was what our IRB approved us to do. As such, approximately 75% of patients have saliva as their normal sample. However, when the pandemic emergency started, we were barred from collecting saliva due to the risk of contagion and our protocol was amended to collect adjacent tissue instead (the recruitment dates and tissue collection dates may not be the same as these tissues can be collected after the patient signs the informed consent form). These samples were used for exome sequencing comparisons, and, for genotyping, only saliva samples were used. We have compared the proportions of all ancestry components between samples that had saliva as their normal tissue vs those which had adjacent normal, and there are no significant differences (Wilcoxon Mann-Whitney P -value for European ancestry proportion = 0.13, Amerindian ancestry proportion = 0.19). We believe the Reviewer may be questioning if a potential ancestry proportion imbalance between these two groups may lead to a different ability to call driver mutations. Because of this, we have also further reviewed manually each adjacent normal sample to make sure there were no *BRAF* V600E mutations, and found none.

14. To identify SNV and driver genes, the authors gave preference for the primary tumor but also used the Metz. Are the TMB mutation frequencies similar? Did the authors use these parameters on the regressions for BRAF, survival, and others?

R. For driver identification, all samples were used per dNdScv⁹ best practices, but the same mutation for samples coming from the same patient is used only once in the calculation. For example, if a primary and a metastasis both had a *BRAF*^{V600E} mutation, this mutation was listed only once in the input data used for driver identification. For follow up analyses, including ancestry, copy number correlations, cell of origin investigations, transcriptomic and survival studies, we used only one sample per patient, preferably the primary when possible. Indeed, most primaries and their associated metastases share driver mutations (now shown as **Supplementary Table 6** and depicted here as **Table R2**). For clarity, our methodology is now depicted in the flowchart which is now **Supplementary Figure 14**. TMB estimations between primaries and metastasis are quite similar (**Figure R4**). TMB was used as a covariate in the regression testing for the relationship between the presence of driver mutations and European ancestry.

Patient	Classification	Primary	Primary 2	LN metastasis	LN metastasis 2	Recurrence	Lesion in transit	Other
PD40961	BRAF	V600E	V600E			V600E		
PD40965	KIT	K642E				No driver		K642E
PD40966	KIT	K642E		K642E	No driver			
PD40967	QWT	No driver		No driver				
PD40969	QWT	No driver		No driver				
PD40971	QWT	No driver		No driver				
PD40978	QWT	No driver		No driver				
PD40980	QWT	No driver				No driver		
PD40983	KIT	K642E		No driver			K642E	
PD40986	QWT	No driver		No driver				
PD40987	BRAF	V600E		V600E				
PD41002	NRAS	Q61R		Q61R		Q61R		
PD41020	NRAS	Q61L		Q61L/ NF1 focal deletion				
PD41025	NRAS	Q61R		No driver				
PD41035	QWT	No driver	No driver					
PD41039	KIT	K642E		K642E				
PD41043	BRAF	V600E		V600E				
PD41046	QWT	No driver		No driver	No driver			
PD41910	KIT	R634Q		No driver				
PD41913	NRAS	Q61R	Q61R					
PD41915	BRAF	V600E		V600E				
PD41920	QWT	No driver		No driver				
PD41923	QWT	No driver		No driver				
PD51928	NRAS	G12R		G12R				
PD51969	NF1	Y1292*	Y1292*					

Table R2. Patients with multiple samples and their driver mutational status.

Figure R4. Comparison of TMB (SNVs + indels) for all samples in this study according to sample type. No significant differences are found among sample types.

15. In the discussion, the authors stated that the relation between gene mutation and the worst prognosis is unknown. Are the authors referring to AM? For non-acral melanoma, this relation has been known for several years (see below a few quick examples – none of those are my papers), and it also includes AM patients for some papers. The authors should discuss their results and compare them with the literature.

<https://pubmed.ncbi.nlm.nih.gov/21343559/>

<https://pubmed.ncbi.nlm.nih.gov/21606968/>

<https://pmc.ncbi.nlm.nih.gov/articles/PMC9431958/>

R. We thank the Reviewer for this suggestion, and have added the following paragraph to the discussion regarding the survival results (lines 1027-1040 of the track changes manuscript): “The observation that any mutation in a driver gene (*BRAF/NRAS/KIT/NF1*) leads to worse prognosis is intriguing. Similar observations, namely that mutation in a MAPK pathway gene

leads to worse prognosis, have been made in cutaneous melanoma. For example, Long and collaborators reported in 2011 that untreated patients with *BRAF*^{V600} mutations have worse overall survival than patients with wild-type *BRAF* tumours¹⁰, an observation that has been replicated in other studies¹¹⁻¹³, with these observations supporting the use of *BRAF* inhibitors in the clinic¹⁴. Other mutations in the MAPK pathway also led to shorter overall survival, sometimes in combination with other prognostic factors^{15,16}. A previous study also found that mutations in the MAPK pathway predict worse survival in acral melanoma¹⁷. Regarding *NF1* specifically, previous reports in cutaneous melanoma have suggested that *NF1*-mutated tumours have differing characteristics, sometimes also associated with worse survival and differential response to immune checkpoint inhibition^{18,19}. Our results, where we find that patients carrying *NF1* mutations have worse recurrence-free survival than patients with no mutations in this gene, support and expand these previous observations.”

We thank Reviewer 1 for their constructive comments.

Referee #2 (Remarks to the Author):

In this study, the authors analyzed 128 acral melanoma (AM) samples from 96 Mexican patients through genotyping, exome sequencing, and gene expression profiling. This is one of the largest genetic analyses of AM. By specifically analyzing melanomas from patients of Latin American ancestry, this study improves our understanding of the genetic basis of the most common melanoma histological subtype found in low and middle-income countries, which has been poorly addressed in previous large-scale sequencing studies. The authors should be commended for this effort.

Regarding novelty, although this study is a tour de force effort in its scale and provides an important resource for the melanoma/cancer genetics fields, the major finding of the increased frequency of *BRAF* mutations in patients of European ancestry has been previously described by Yeh et al., 2019. The authors do identify new low-frequency significantly mutated genes (e.g. *SPHKAP*, *POU3F3*, *RDH5*, *MED12*), but the authors provide little evidence that these genes play a role in AM. Finally, in the analysis that demonstrates *BRAF* mutated AM exhibit a more ‘cutaneous melanoma-like’ transcriptome suggesting a potential distinct melanocyte cell of origin compared to other AMs, I do have some concerns on this specific analysis (described in more detail below).

I have no major concerns regarding the statistics employed in the analysis, and the code related to this study is well-organized.

R. We thank Reviewer 2 for the comments above and in the responses below we detail the new insights we have gleaned for our analysis.

Comments/suggested edits:

1. Ancestry and clinical characteristics of Mexican AM patients

1.1 Do the authors have information related to pigmentation of the biospecimens to be included in clinical data file, and does this differ depending on patient ancestry?

R. We agree that the addition of skin pigmentation information would be a valuable addition to the clinical files. Unfortunately, we do not have any information on pigmentation of these samples. A previous, large GWAS study did not find an effect of skin pigmentation on acral melanoma risk²⁰, but we would indeed expect skin pigmentation to be related to patient ancestry^{21,22}.

2. Genomic profiling of AM samples identifies correlations of ancestry and age with somatic alterations

2.1. In Figure 1a, how were the genes in the oncoplot selected? In Supplementary Figure 2 for quadruple wild-type patients, were these the only genes with impactful mutations? If there are others with similar mutation frequency, what was the rationale/selection criteria? If mutation frequency was used as the selection criteria and not significantly mutated genes, is it possible that genes with SNVs highlighted in the quadruple WT are enriched for passenger events and that CNAs primarily drive the quadruple WT tumors instead? Can the authors please provide the complete output of the dNdScv analysis?

R. We understand the point raised by the Reviewer and we have amended our Figures so this information is clearer, and have also specified in Figure legends. The genes shown on the oncoplot in Figure 1 are the genes with the lowest q values as identified by dNdScv, and we have only depicted genes with enough evidence, both from this study and substantial literature, for being drivers (seven genes) in Figure 1. We do agree this improves the clarity in the manuscript flow. In **Supplementary Figure 2**, we now depict all samples, including those without mutations in the genes depicted in the oncoplot, in addition to genes that are mutated in our cohort and already have published evidence of being low-frequency drivers. We believe this study adds evidence to the hypothesis that these are true driver events. We have now included the full output from dNdScv in **Supplementary Table 3**.

Regarding the point “is it possible that genes with SNVs highlighted in the quadruple WT are enriched for passenger events and that CNAs primarily drive the quadruple WT tumors instead?”, we indeed find differences in the CNA landscape between *BRAF/NRAS/KIT/NF1*-mutated and quadruple WT tumours. Specifically, a focal peak including only seven genes (*ARF6*, *SOS2*, *VCPKMT*, *L2HGDH*, *LINC01599*, *LINC01588*, and *MIR6076*) is selectively amplified in quadruple WT tumours ($P = 0.043$, one-sided exact Fisher test). Likewise, the 1p12 region is selectively amplified in the mutated tumours ($P = 0.021$, one-sided exact Fisher test). This region includes *NOTCH2* and another 30 genes. Co-occurring *NOTCH2* and *NRAS* events have been previously reported in a melanoma PDX collection²³, in our collection, 5 out of 8 *NRAS*-mutated tumours showed this amplification. In addition, the amplification peak containing *CCND1* is preferentially amplified in QWT tumours ($P = 0.044$, one-sided exact Fisher test) (**Supplementary Figure 4**, depicted here below as **Figure R5**). A plot and explanation regarding these findings are found in the Results, lines 513-520 of the track changes manuscript.

Figure R5. Copy number profile of *BRAF/NRAS/KIT/NF1*-mutated tumours vs. quadruple WT. All depicted regions have been identified by GISTIC2 analysis per group of samples. Statistically significant differences are marked in colour, yellow = amplified in mutated tumours, red = amplified in QWT, green = deleted in mutated tumours, blue = deleted in QWT tumours. Differences are determined first by assessing the global GISTIC2 output and determining differences between groups by one-sided Fisher's exact test ($P < 0.05$). If a region is not found in the global GISTIC2 output, but it is found only in the analysis per group, we have indicated it as statistically different. Number of mutated tumours = 22, number of QWT tumours = 25.

2.2. Of particular importance, the authors have nominated several novel driver mutations in SPHKAP, POU3F3, RDH5, MED12 in this cohort. How confident are the authors that these are oncogenes or tumor suppressors in melanoma? Are the loss of function mutations in SPHKAP or POU3F3 associated with loss of heterozygosity, or decrease in its mRNA expression? Are any of these genes mutated other cancers? To my knowledge, I don't think they have been reported in any other acral melanoma or pan-cancer mutation significance analysis (e.g. Bailey et al., 2018, Cell <https://doi.org/10.1016/j.cell.2018.02.060>). Is there any functional validation of these genes in large screening studies (e.g. DepMap)? As these findings are part of the more novel aspects of this study, providing supporting evidence would strengthen the manuscript.

R. We are grateful to the Reviewer for this comment, and we agree that more clarity and evidence is needed to nominate novel driver genes. These candidate genes, with the exception of *RDH5*, have now gone down the ranked list, (i.e., the output from dNdScv), after we have re-performed our mutation and indel calling after the suggestions from the Reviewers. In our revised strategy, we have decided to consider genes mutated at low frequency in our cohort which have published evidence of their pathogenicity. We now, with support from the literature, nominate and/or add support to the following genes as low-frequency drivers in acral melanoma:

1. *PTPRJ*. In our cohort, this gene presents two frameshift deletions in different patients. This gene has already been found as part of a long tail of potentially uncommon drivers by Wang et al (2022)²⁴ and in two studies of canine melanoma, one by Wong et al

(2019)²⁵ and another one by Hendricks *et al* (2018)²⁶. In all cases, mutations are truncating suggesting a tumour suppressor function. Interestingly, according to this literature, this low-frequency driver seems to be shared between acral and mucosal melanoma. Reports that find this gene as a driver were published after the Bailey *et al.* (2018) paper²⁷.

2. *ATM*. In our cohort, this gene presents with two nonsense mutations. *ATM* is a well-known cancer-associated gene, and has been reported in acral melanoma before by Boris Bastian's group both with low-frequency mutations somatically and germinally^{1,3,24}. The mutations in our cohort support the involvement of this gene in a minority of quadruple wild-type melanomas (non *BRAF/NRAS/KIT/NF1*). *ATM* is listed as a tumour suppressor in the Bailey *et al.* (2018) paper²⁷.
3. *NF2*. In the Mexican cohort presented here, this gene presented with one acceptor splice site and one frameshift insertion. Truncating mutations at low frequencies in *NF2* have also been found by Wang *et al* (2022)²⁴ and by Turner *et al* (2024)²⁸, again both in acral and mucosal melanomas. *NF2* is listed as a tumour suppressor in the Bailey *et al.* (2018) paper²⁷.
4. *RDH5*. In our cohort, this gene has one frameshift deletion and one nonsense mutation (W95* and P25Ifs*32), so these mutations are truncating. We believe this may be a low-frequency driver because: a) This gene is short, with only 318 amino acids, which lowers the probability of accumulating mutations (and indeed, we find zero synonymous mutations in it), and, 2) it has been found associated with cancer progression, specifically: in hepatocarcinoma, it has been reported that low expression is associated with metastasis via functional assays²⁹, in breast cancer it has also been found downregulated in comparison with normal tissue³⁰, and in colon cancer cell lines it lacked expression and was associated to a poor rate of conversion of retinol into retinoic acid³¹.

We have also performed the suggested DepMap analysis for these four putative drivers, and *BAP1*, a known tumour suppressor which we include for comparison (**Figure R6**). These results were generated using the 24Q4 release of DepMap, containing over 900 cell lines. Our analysis shows that *NF2* and *ATM* are context-dependent tumour suppressors, as the fitness of some cell lines is greatly promoted by their disruption - this is in keeping with the literature on these genes across a range of histologies. *PTPRJ* does not show essentiality across the DepMap cell line collection, indicating that its disruption is tolerated and not profoundly growth promoting, but this gene is well established as a driver, including in mucosal melanoma, as we have previously described. It is possible that the biology of this gene and its

Figure R6. Gene effect scores for putative tumour suppressor genes, and *BAP1*. Data are taken from the DepMap project.

contribution of tumorigenesis is not well captured by a CRISPR screen. *RDH5* has the lowest median scores, and influences cell fitness in a way similar to *BAP1*, again pointing to an important role of context in the contribution of these genes to tumorigenesis. We now discuss the evidence for the involvement of these genes as low-frequency drivers in the Results (lines 370-376 of the track changes manuscript).

2.3. It would be useful for the readers to know what are the specific mutation types that are found for each highlighted mutated gene (for example what are the number of C>T transitions that are found for the highlighted genes in Figure 1a or Supplementary Figure 2?). It would also be useful to compare the frequency of mutations in these genes in melanomas from sun-exposed skin or other acral melanoma studies.

R. We agree with the Reviewer, and we have now added a horizontal bar plot to the oncoplot in **Figure 1** to show the spectra of mutation types for each highlighted gene. Likewise, we have added a figure inset (**Figure 1d** in the manuscript and **Figure R3** in this document) where we show the comparison of the frequency of mutations in melanomas from sun-exposed sites and other published acral melanoma studies.

2.4. Figure 1c. The authors trained a logistic regression model to predict the mutation preferences in each ancestry with the control of potential confounders. Can the authors provide a more comprehensive evaluation of the model's performance? For example, how well can the model perform when testing the data with a leave-one-out cross-validation?

R. We computed the leave-one-out cross-validation (LOO-CV) of each of the four models, using AUC score since this is a classification problem. The results show that the model that best predicts the mutation is the one trained to predict *NRAS* mutations. However, the only model that shows ancestry as a relevant predictor is the one predicting a *BRAF* mutation, with an AUC of 0.62.

However, it is important to note that these models were not constructed to be predictive, since we would not expect to explain the presence of a driver mutation only with a single indicator of ancestry and a few demographic covariates. Here, the purpose of these models was to find whether there existed a relationship with a specific ancestry, while controlling for possible predictors. So the conclusion should be only that, with the given predictors, ancestry still explains some of the variance on the mutations only for mutations in *BRAF*.

Mutation	Lower	Upper	Estimate	P value	LOO CV
KIT	-6.2873	2.2742	-2.0065	0.3582	0.5628
BRAF	0.6639	9.4406	5.0522	0.02403	0.6157
NRAS	-3.4425	7.0075	1.7824	0.5037	0.7871
NF1	-2.5576	7.5625	2.5024	0.3324	0.6453

Table R3. Leave-one-out cross-validation (LOO CV) analysis model per mutation. ‘Lower’, and ‘upper’ refer to the bounds of the 95% confidence interval for the estimate. The estimate column indicates the effect size (e.g., regression coefficient) of each mutation on the outcome variable. The LOO-CV column indicates the AUC (area under the curve) of the classifier constructed using the logistic model.

2.5 Figure 1d. The authors provided data that NRAS mutations were strongly associated with younger age at diagnosis. However, one concern is that the correlation might be confounded by the ulceration status of the tumor. NRAS mutated tumors appear to have less ulceration compared with the samples harboring other mutations (shown in Fig 1a). Can the authors prove that the association is independent from ulceration status?

R. We thank the Reviewer for raising this point - we have re-performed the linear regression analysis now controlling for ulceration. Adding this covariable indeed caused the association between *NRAS* and age at diagnosis to disappear. Subsequently, we tested the association of *NRAS* mutation with ulceration status, finding a significant association between *NRAS* mutation and lower ulceration rates (Two-tailed Fisher’s Exact test $P=0.016$). The observation that *NRAS* mutation is associated to lower ulceration rates than *BRAF* mutation had already been reported for cutaneous melanoma³², but this is the first time that it is reported for acral melanoma (to the best of our knowledge). Therefore, after re-doing this analysis, we do not find any independent association with age at diagnosis anymore. We have clarified this in the text now, lines 420-424 of the track changes manuscript.

2.6. General methods: Can the authors clarify for the somatic SNV calling, why they used both Mutect and Mutect2 before applying the filter of a minimum of two callers. Does this bias the variant calling to Mutect1/2? Why not just use Mutect2 if it is an improvement over mutect1 and take variants called by both Mutect2 and varscan? Mutect also does indel calling. Why use Strelka2 for indels and ignore Mutect2 calls? Was indel realignment performed?

R. We thank the Reviewer for their observation. In our previous discussions and analyses deciding what mutation callers to use, we observed that Mutect and Mutect2 called different sets of mutations (**Figure R6**). This makes sense as Mutect2 uses a Bayesian somatic genotyping model that differs from the original MuTect (GATK Team, 2025³³). However, when we re-analysed our results, we acknowledge that the Reviewer is correct that Mutect and Mutect2 overlap more than any other two callers, and thus, we have changed our strategy in response to this observation. In this Resubmission, we have swapped Mutect for CaVEMan⁸. The overlap between CaVEMan and the other two remaining tools (Mutect2 and Varscan2) is similar and we believe we have refined the accuracy of our calls (for example, by the resulting mutational signature analysis, which eliminated all known artifacts).

Regarding indels, indel realignment was performed as part of the best practices protocol for Mutect2 and VarScan2. We initially decided against using Mutect2 for indel calling as we re-sequenced some of the resulting indel calls and found that most were artifacts. We do agree however with Reviewers 2 and 3 that we originally had called an excess of indels, and we have now used Pindel^{34,35} as used by the PCAWG study⁷. This strategy greatly reduced the number of indels, improving their quality, in line with expectations (**Supplementary Table 1**).

Figure R6. Upset plot showing the overlap of mutations called by four different tools. Mutations unique to Mutect and Mutect2 are shown in the first two bars. The overlap between Mutect and Mutect2 is shown in the fifth bar from left to right.

3. Somatic copy number landscape of AM samples identifies correlations with somatic alterations

3.1. Can the authors comment on their confidence in the Sequenza tool for copy number alterations (CNAs), cellularity and ploidy? Have the authors compared their results to other tools (i.e. PURPLE)?

R. This is a great question and indeed one that has led to many discussions among our group and our collaborators. The PURPLE tool is mostly recommended for whole-genome analyses, and we at the moment are unaware of any reports using it for whole-exome datasets, but other reports have used tools such as ASCAT³⁶ for the estimation of CNA, cellularity and ploidy for whole-exome datasets. We have evidence that Sequenza is superior at estimating lower sample purities than other tools such as ASCAT (**Figure R7, right panel**), and has been used in other whole-exome acral melanoma studies⁶, but we acknowledge that ASCAT is more widely used and was the tool utilised in the primary analysis pipeline by the PCAWG consortium. Therefore, we have settled on a mixed strategy: 1) We calculated tumour purity with both Sequenza and ASCAT. For the majority of tumours, estimates mostly agree (92%). However, for the remaining 11 samples, estimates are wildly different, with Sequenza estimating a low sample purity (therefore, having led us to exclude them from follow-up copy number analyses) and ASCAT estimating a sample purity of 1. In this case, we observe that these tumours have fewer mutations (**Figure R7**), which would agree with them having a low sample purity. This is also evident in the histology slides for these tumours, which we reviewed. 2) We removed the tumours for which ASCAT estimated a sample purity of 1 and Sequenza estimated <0.15 , or for which goodness-of-fit was below 0.95 as indicated by ASCAT, as these are potentially unreliable samples for copy number analysis. However, for all other samples (those that had similar purities estimated by both tools), we kept the ASCAT estimates. 3) For follow-up analyses, we kept samples that had an ASCAT-estimated goodness-of-fit equal of

higher than 95. In the final dataset, used for the copy number analyses, 60 samples from 47 patients remained.

Figure R7. Correlation between TMB and sample purity as called by ASCAT. A number of samples with very low TMB can be seen where ASCAT estimates a tumour purity of 1 (left panel, lower right). The majority of these samples were excluded due to low purity estimates by Sequenza and/or lower goodness-of-fit by ASCAT (table at the right).

3.2. Figure 2b. The figure for the copy number variations is not entirely clear. As stated by the authors, more amplifications than deletions were observed and wondering whether these values are relative to tumor ploidy or relative to 2? A plot with higher genomic resolution showing the frequency of alteration at each genomic locus in each subtype might be easier to visualize than the heatmap (for example, something similar to the Figure 4 from the Newell et al., 2020 study <https://doi.org/10.1038/s41467-020-18988-3>).

R. We thank the Reviewer for their observation. Figure 2a assumes a baseline ploidy of 2 as it is plotting the copy number changes with respect to the normal sample. However, we agree that showing the copy number changes relative to sample ploidy would be as informative. Therefore, we have now modified Figure 2b to show changes relative to each tumour sample ploidy, and have added an explanation for what exactly each figure is showing in the Figure legend. Additionally, following the Newell *et al.* Figure 4, we have also added a binary matrix representing whether specific genomic regions are amplified or deleted, as well as their quantification in each sample, as **Supplementary Figure 5**.

3.3. Previous studies have shown that AM have a lower SNV burden and higher numbers of structural rearrangements and focal copy number events compared to cutaneous melanoma (e.g. Turajlic et al., 2012; Hayward et al. 2017; Newell et al., 2020; Meng Wang et al., 2024). Can the authors comment or perform a comparative analysis to provide readership an understanding of differences between the cohort of Mexican patients?

R. In response to this comment, we have performed a thorough review of the literature and have performed an analysis of the Newell et al. 2020 dataset. When compared to cutaneous

melanoma, we do see, as others have shown, differences in SNV burden and copy number aberrations. However, the Mexican dataset shows differences with these other mostly European-descent cohorts in several points:

- 1) The ancestry composition of the patients in our study includes a significant proportion of Native American ancestry, which is severely underrepresented in already published cancer genomics studies and permits the identification of relationships of specific ancestries with somatic characteristics. The elucidation of the relationship between ancestry and somatic mutation rate was possible to observe due to this, which is mostly absent (or not reported) in other datasets (**Figure 1c**). We tried to replicate this analysis in the Newell *et al.* study, but most patients are of European ancestry, and is therefore uninformative.
- 2) The fraction of activating *BRAF* mutations is lower than in the studies with predominantly European-descent patients, and more similar to those with Chinese patients, probably due to the positive relationship between *BRAF* mutation and European ancestry
- 3) Access to treatment. The most common treatment in our cohort was cytotoxic chemotherapy only (26% of patients, including dacarbazine, carboplatin, and/or paclitaxel), with only four patients (4.3%) being able to access immunotherapy (nivolumab and/or ipilimumab). This is markedly different than what is reported in other cohorts (For example, Newell *et al* include only two patients with chemotherapy only, whereas more than 20% of patients were able to access immunotherapy and/or *BRAF/MEK*i targeted inhibitors).

We agree with the Reviewer that these are important points and have included them in the Results (lines 435-442 of the track changes manuscript).

3.4 Can the authors please provide a more detailed explanation of GCS scores to be included in the manuscript?

R. We thank the Reviewer for this suggestion and we apologise for the lack of clarity in the earlier version of the manuscript. We have added a clearer explanation of GCS scores to the Methods section, lines 1212-1248 of the track changes manuscript: "GCS (Global copy number alteration score) is a number quantifying the copy number aberration level in each sample provided by the CNApp tool³⁷. Higher GCS scores indicate a higher burden of copy number aberrations compared with all other samples in the cohort. GCS is the sum of the normalised BCS (Broad copy number alteration score) and FCS (focal copy number alteration score), which are calculated considering broad (chromosome and arm-level) and focal (weighted focal CNAs corrected by the amplitude and length of the segment) aberrations per sample. These values are calculated using as input the number of DNA copies normalised by sample ploidy. A more detailed explanation can be found in the original publication."

3.5 On line 207: Regarding the suggestion that *BRAF/ NRAS*-mutated might be of different aetiology because they have distinct SCNA profiles: What if the cell of origin is the same but the evolutionary trajectory of tumors with these mutations tend to be different?

The Reviewer raises an important point. Indeed, differences in the transcriptomes of advanced disease could reflect distinct cells of origin or divergent evolutionary trajectories, with oncogene-specific expression programs shaping tumor development. It is important to clarify

that the hypothesis that acral melanomas may arise from different cells of origin is not solely based on this study but is supported by prior work. Our previous research has demonstrated transcriptional diversity among melanocytes in different anatomic locations, including distinct populations of epidermal melanocytes in the palms and soles³⁸. Additionally, our zebrafish model studies have shown that AM-associated drivers preferentially (though not exclusively) induce tumors in the limbs (fins), whereas CM-associated *BRAF* mutations preferentially (but not exclusively) lead to tumors in the trunk³⁹. Furthermore, we have demonstrated that *BRAF* mutations selectively drive hyperproliferation in only a subset of less-pigmented primary human melanocytes⁴⁰.

Taken together, these findings suggest a model in which melanocyte populations vary by anatomic site, potentially influencing tumorigenesis in a mutation-dependent manner. In the present study, we have genetically and transcriptionally characterised a sizeable cohort of AM clinical specimens, and our analyses are consistent with this hypothesis that emerged from these prior observations of model systems. However, the present study design is not suited for addressing temporal questions such as evolutionary trajectories, nor for systematically testing alternative hypotheses through cellular or molecular manipulation. While our findings align with the cell-of-origin model, we recognize that they do not exclude alternative explanations in isolation. However, when integrated with prior work and the additional experiments provided in this revision (see response to 5.1 below), the collective evidence most strongly supports the cell-of-origin model. We have added these points to the discussion (lines 998-1012 of the track changes document).

5. BRAF-mutated acral melanomas exhibit a transcriptional signature more characteristic of non-acral cutaneous melanomas

5.1. A critical insight/suggestion from this work is that BRAF mutant acral melanomas have a different cell of origin from BRAF/NRAS WT acral melanomas based on the finding that “BRAF-mutated acral melanomas exhibit a transcriptional signature more characteristic of non-acral cutaneous melanomas”. To arrive at this conclusion, my understanding is the authors first derived an acral vs. non-acral cutaneous melanoma score by measuring the expression of genes in acral and non-acral cutaneous melanomas in a separate cohort of 20 patients. The authors looked at a limited set of genes determined previously to be differentially expressed between glabrous and non-glabrous melanocytes. Of these genes, using 20 genes most discriminative of acral vs. non-acral melanomas, determined by principal component analysis, they computed multiplicative score in 80 patients from their cohort (the acral:cutaneous (A:C) ratio) and determined that the score separates BRAF mutant acral melanoma from BRAF/NRAS WT acral melanomas (Fig 3C). They validated this finding in a separate cohort of 63 patients from Newell et al., 2020 (Fig. 3D).

The authors suggest that because this score has the power to separate acral from non-acral cutaneous melanomas, its ability to separate BRAF mutant acral melanomas from BRAF/NRAS WT acral melanomas indicates that BRAF mutant acral melanomas likely have a different cell of origin, similar to melanocytes enriched in limbs (c-type). Whereas that authors suggest that BRAF/NRAS WT acral melanomas may originate from v-type melanocytes enriched in volar regions.

A major caveat is that the genes used to produce this score may simply be distinguishing BRAF mutant melanomas from BRAF WT melanomas, as opposed to cell of origin. This is because the genes were initially selected based on their ability to distinguish non-acral cutaneous melanomas (which are highly enriched in BRAF mutants) from acral cutaneous melanomas (which are depleted of BRAF mutants). Previous studies have shown that BRAF mutations can be associated with their own transcriptional signature. Furthermore, BRAF activating mutations are often associated with chromosome 7 amplification. Therefore, in cells where BRAF is mutated, many other genes on chromosome 7 including BRAF exhibit increase expression, along with altered expression of the downstream effectors of those genes.

The authors acknowledge this caveat, yet still suggest that their signature is distinguishing acral melanomas of different cell origin as opposed to simply distinguishing BRAF mutant and WT melanomas of the same cell of origin. To help address this issue, can the authors demonstrate that their signature is unable to separate BRAF mutants from BRAF/NRAS WTs in non-acral cutaneous melanomas, which the authors suggest originate from c-type melanocytes? Furthermore, the differences found between BRAF/NRAS mutated group and WT group can be confounded by many factors such as sample purity. Can the authors find a significant difference in A:C ratio in the CCLE dataset or RNA seq data from CM and AM cell lines?

The Reviewer raises a critical and thoughtful point regarding the challenge of distinguishing whether the observed transcriptional differences in *BRAF*-mutated acral melanomas reflect differences in cell of origin or oncogene-specific programs. We fully acknowledge that this type of study, which relies on analysis of clinical specimens, is not inherently suited to definitively resolving this distinction (see discussion in point 3.5, above). However, we have now employed multiple approaches to evaluate the plausibility of each model, within the limitations imposed by available data and technical constraints.

First, to directly address the Reviewer's suggestion, we have now performed the requested analysis using TCGA data for non-acral cutaneous melanomas. We do not observe the difference we see in acral melanomas – suggesting that in non-acral cutaneous melanomas, *BRAF*-driven tumors are not different from *BRAF* wildtype tumors with respect to our classifier (**Supplementary Figure 9a**). Although this observation is consistent with our model that *BRAF* mutation status specifically distinguishes AMs with a distinct origin, but not necessarily non-acral cutaneous melanomas, we cannot rule out the possibility that the negative result of no difference is not artifactual, for example, due to greater noise in the TCGA samples. We therefore sought additional complementary approaches to address this question.

We appreciate the reviewer's suggestion to utilize cell line datasets such as CCLE. However, acral melanomas are notably resistant to the establishment of stable cell lines due to a bottlenecking effect, making such datasets of limited utility for this analysis. For example, CCLE contains only a single cell line classified as acral melanoma. Given the stark contrast in derivation efficiency between non-acral cutaneous melanomas (>90%) and acral melanomas (extremely poor—unpublished data from multiple labs, as reflected in the scarcity of available lines), even if a sufficient number of transcriptomes were available for a statistically powered comparison, it remains uncertain how well these rare established lines (by definition the outliers) capture the true gene expression profiles of the disease.

We therefore conducted additional analyses that we had not previously considered. As the Reviewer notes, 'previous studies have shown that *BRAF* mutations can be associated with their own transcriptional signature.' To further explore whether the A:C classifier primarily reflects a *BRAF*-associated transcriptional signature, we performed two additional analyses:

- 1) Using a recently developed method for assessing gene signature similarity⁴¹, we compared the c-mel gene signature from our classifier to a previously published set of genes directly activated by mutant *BRAF* in melanoma cells. We found no significant correlation between these signatures (**Supplementary Figure 9b**).
- 2) We examined datasets in which mutant *BRAF* was introduced into primary melanocytes in a doxycycline-inducible manner⁴⁰. Again, we found that the c-mel signature genes were not activated downstream of mutant *BRAF*, further suggesting that the classifier does not simply reflect *BRAF*-driven transcriptional changes (**Figure 3d**).

Because studies of clinical specimens alone cannot fully resolve this question, we still feel it is important to acknowledge and discuss both cell-of-origin hypothesis and the oncogene-signature hypothesis in the manuscript. While our additional analyses do not strongly support an oncogene signature as the primary etiology, thus favoring the cell-of-origin model, it is possible that in some cases these could be intertwined. Recent data has shown that some acral melanomas harboring amplifications of the *CRKL* oncogene depend upon *HOX13* positional identify programs already present in the cell of origin, suggesting that oncogenes and cell-of-origin programs can synergize³⁹. Given these collective observations, we believe it is reasonable to favor the cell of origin model in our conclusions, while making clear that both interpretations exist and that this is an important and ongoing area of investigation. We have now made this clear in the Discussion, lines 998-1012 of the track changes document.

5.2 Figure 3a, please add the percentage of explained variances based on PC2

R. We have added the percentage of explained variances based on PC2 in Figure 3a.

5.3 Figure 3d. The authors show 3 mutated subtypes (WT, *BRAF* and *NRAS*) in their cohort (shown in fig 3c), but why is the *NRAS*-mutated subgroup excluded in the Newell et al. cohort?

R. We had decided to only depict the *BRAF* vs *BRAF*-WT comparison in the plot from the Newell dataset due to our previous observation that only *BRAF* mutated tumours showed significant differences in score against the other categories, but we agree that this was unclear. We have now re-done the Figure to depict only the *BRAF* vs *BRAF*-WT comparison for both datasets as it illustrates better the point of the analysis (**Figure 3c**, and replicated here below as **Figure R8**). We replicate our original observation in both datasets.

Figure R8. Transcriptional score comparison per mutational group in the Mexican dataset (left) and in Newell *et al* (right).

6. Transcriptional landscape of AM tumors identifies three subgroups with distinct clinical and prognostic characteristics

6.1 Figure 4a. To better understand the transcriptome differences, can the author include some differentially expressed marker genes on the figure for each cluster and add sample purity as one of the column annotations?

R. We think this is a great suggestion. We have now included marker genes in the main cluster Figure (Figure 4a), and have added **Supplementary Figures 10-12**, as well as **Supplementary Table 17** quantifying the levels of distinct marker genes across the different clusters.

6.2 The authors provide differences in cell type proportions in the TMEs shown in Fig 4c-e across the three clusters. Can the authors provide a supplementary figure to show the differences in all the inferred cell types among the three clusters?

R. Following the Reviewer's suggestion, we have added a supplementary figure showing the quantification of cell types [including B cells, cancer-associated fibroblasts (CAFs), CD4 T cells, CD8 T cells, endothelial cells, macrophages, NK cells and other cells (Figure 4d-f and **Supplementary Figure 13**). Our results after re-doing the clustering analysis as a result of this revision are maintained, but now we also report a significant association of B cell infiltration with cluster 2, possibly associated with the ulcerated state of the majority of these samples.

6.3 To better understand the distinct features of each cluster, can authors include the pathway enrichment results of the DEGs (differentially expressed genes) in each cluster?

R. We have now included a Supplementary Table per cluster with the differentially expressed genes and pathways (**Supplementary Tables 18-20**). Cluster 1 is associated with keratinocyte and epidermal processes (indicating, as the Reviewer has noted below, lower-

stage tumours), Cluster 2 with a highly proliferative, cell cycle-active profile, and associated with lower survival, and Cluster 3 with an overexpression of OXPPOS and respiration processes, potentially indicating non-proliferative, metabolically active tumours.

6.4. Cluster 1 appears to be related to lower-stage tumors, which may explain the observed improved prognosis, lower ulceration and Breslow thickness and could explain differences in expression. Can the authors explain why this was not a confounding variable? Furthermore, are there sex differences in acral melanoma? 9 out of 11 patients in Cluster 1 are female. Authors may wish to discuss this imbalance as compared to Cluster 2 and 3.

R. The Reviewer is right in that Cluster 1 tumours are lower stage, ulceration and Breslow thickness could lead to differences in expression. However, we think that this is a characteristic of these tumours - they are more infiltrated with immune and other cells (as shown by deconvolution analyses in **Supplementary Figure 13c**) and this will affect the detected gene expression. We hope that this is clear in lines 769-776 of the track changes manuscript. The sex differences among expression clusters are indeed very interesting, although these are not statistically significant. This replicates what has been observed in other cutaneous melanoma datasets, in which women have earlier age at diagnosis and stage^{42,43} perhaps due to social factors. Inspired by this question, we have performed a more thorough analysis of the sex differences across our cohort, and have observed, interestingly, that mutation in any driver (*BRAF/NRAS/KIT/NF1*) is statistically significantly associated with sex (with women having more MAPK pathway mutations). This, in our view, is intriguing and replicates observations in other studies^{17,18,44}. We are especially grateful for the Reviewer's insight here as this analysis allowed us to describe another interesting association (**Table R4**). This is now included in the Results, lines 376-380 of the track changes document, and as **Supplementary Table 5**.

Characteristic		Female (n=54) (100%) (58%)	Male (n=38) (100%) (42%)	Total (100%)
Age	Mean	60.79	61.82	61.21
	Range	32-98	43-85	32-98
Breslow depth	Mean	5.98	7.48	6.61
	Range	0.45-50 (n=52)	0.6-38 (n=37)	0.45-50
TMB	Mean	50.57	45.05	48.29
	Range	0-176	1-124	0-176
Stage	0	1 (1.85%) (100%)	0 (0%) (0%)	1 (100%)
	I	8 (14.81%) (80%)	2 (5.26%) (20%)	10 (100%)
	II	14 (25.93%) (56%)	11 (28.95%) (44%)	25 (100%)
	III	26 (48.15%) (55%)	21 (55.26%) (46%)	47 (100%)
	IV	5 (9.26%) (55%)	4 (10.53%) (45%)	9 (100%)
Primary location	Foot	42 (77.78%) (55%)	34 (89.47%) (45%)	76 (100%)
	Hand	4 (7.4%) (100%)	0 (0%) (0%)	4 (100%)
	Subungual	8 (14.81%) (66%)	4 (10.53%) (34%)	12 (100%)
Classification	BRAF	10 (18.52%) (83%)	2 (5.26%) (17%)	12 (100%)
	NRAS	10 (18.52%) (83%)	2 (5.26%) (17%)	12 (100%)
	NF1	3 (5.56%) (43%)	4 (10.53%) (57%)	7 (100%)
	KIT	9 (16.67%) (75%)	3 (7.89%) (25%)	12 (100%)
	Multihit	1 (1.85%) (100%)	0 (0%) (0%)	1 (100%)
	QWT	21 (38.89%) (44%)	27 (71.05%) (56%)	48 (100%)
			Female (n= 25) (100%) (56%)	Male (n = 19) (100%) (44%)
RNAseq Clusters (n=44)	Cluster1	8 (32%) (57%)	6 (31.57%) (43%)	14 (100%)
	Cluster2	9 (36%) (56%)	7 (36.84%) (56%)	16 (100%)
	Cluster3	8 (32%) (57%)	6 (31.57%) (43%)	14 (100%)
Sample mean coverage		45.05	44.56	44.86

Table R4. Clinical, mutational and transcriptional characteristics by sex. Two percentages are shown in each cell, the first one is by column and second one by row. Differences in mutational classification are significant between men and women.

7. Somatic and gene expression profile influence recurrence-free survival

7.1 The authors state that “Those with a driver mutation (BRAF, NRAS, KIT, NF1 or multihit) had a significantly higher probability of having earlier recurrences (Log-rank test P-value < 0.05) (Supplementary Table 8, Figure 5a), with NF1 mutations likely having a stronger effect (Supplementary Table 9, Figure 5b).” Can the authors validate the same trend (MAPK pathway driver mutations leading to a higher risk of recurrence) in a larger cohort of acral melanoma or in non-volar cutaneous melanoma? Additionally, is this trend independent of their ancestries?

R. As part of this revision, we have re-done variant and indel calling, and excluded four samples due to extensive reviewing of clinical and sequencing information (Three due to representing different subtypes of melanoma, and another one due to low coverage). After re-classifying tumours by driver mutation, we re-performed all survival analyses, and we now observe a higher risk of death with any MAPK mutation (*BRAF*, *NRAS*, *NF1*, and *KIT*). We have added the following paragraph to the discussion: “The observation that any mutation in a driver gene (*BRAF/NRAS/KIT/NF1*) leads to worse prognosis is intriguing. Similar observations, namely that mutation in a MAPK pathway gene leads to worse prognosis, have been made in cutaneous melanoma. For example, Long and collaborators reported in 2011 that untreated patients with *BRAF*^{V600} mutations have worse overall survival than patients with wild-type *BRAF* tumours¹⁰, an observation that has been replicated in other studies^{11–13}, with these observations supporting the use of *BRAF* inhibitors in the clinic¹⁴. Other mutations in the MAPK pathway also led to shorter survival, sometimes in combination with other prognostic factors^{15,16}. A previous study also found that mutations in the MAPK pathway predict worse survival in acral melanoma¹⁷. Regarding *NF1* specifically, where we find that patients carrying *NF1* mutations have worse recurrence-free survival than patients with no mutations in this gene, previous reports in cutaneous melanoma have suggested that *NF1*-mutated tumours have differing characteristics, sometimes also associated with worse survival and differential response to immune checkpoint inhibition^{18,19}. Our results support and expand these previous observations.” (lines 1027-1040 of the track changes document).

The finding of patients with *NF1* mutations having earlier recurrences is maintained in the new analysis, however, we reviewed a number of studies in the literature for datasets that had cohorts with both *NF1* mutation status and recurrence appearance (**Table R5**), but we were unable to find any reports where we could address this question more specifically.

X - Data are available with publication

(-)- Data are not available

Table R5. List of studies reviewed for *NF1* mutation and recurrence data.

No	Reference number	Recurrence Data	Mutational Status Data	Correlation Analysis	Notes
1	Wang et al. 2021 ⁴⁵	x	-	-	Patient tables are not available. (211 ALM patients with follow-up data)
2	Tas & Erturk 2018 ⁴⁶	x	-	-	Patient tables are not

					available. (102 ALM patients)
3	Hayward et al. 2017 ⁴⁷	-	x	-	Survival data unavailable. (35 ALM patients)
4	Hida et al. 2024 ⁴⁸	-	x	-	Survival data unavailable (46 ALM samples)
5	Dai et al. 2013 ⁴⁹	x	KIT	Poor correlation between KIT and recurrence	Performed a screen for KIT mutation and expression only (39 AML samples)
6	Holman et al. 2020 ⁵⁰	-	NF1, KRAS, HRAS, KIT, BRAF, NRAS	Samples classified into subungual and acral. Recurrence/ mets classification is ambiguous	Patient tables are not available. (54 cases of SUM and 78 cases of non subungual acral melanoma)
7	Newell et al. 2020 ²	-	x	-	Small sample size (36 primary tumors)
8	Krauthammer et al. 2012 ⁵¹	-	KRAS, HRAS, BRAF, NRAS	-	sun-exposed melanomas and sun-shielded types, including acral melanomas, but does not specify the number of acral samples.
9	Farshidfar et al. 2022 ⁵²	-	x	-	(104 AML patients)
10	Liu et al. 2024 ⁵³	-	NRAS, KRAS, NF1, KIT	-	(147 AML patients)
11		X	NRAS,	-	Molecular

	Ahn et al. 2023 ⁵⁴		KRAS, HRAS, NF1, KIT		testing for common melanoma driver genes. Data are not published
12	Huang et al. 2020 ⁵⁵	X	-	-	
13	Kim et al. 2018 ⁵⁶	X	-	-	
14	Dika et al. 2020 ⁵⁷	-	BRAF, KIT, and NRAS	-	Small sample size (31 samples)

Referee #2 (Remarks on code availability):

code related to this study is well-organized

Referee #3 (Remarks to the Author):

In this manuscript, Basurto-Lozado and colleagues performed a genomic and transcriptomic analysis of acral melanomas from Mexico. Acral melanoma is the most common type of melanoma in skin of color and yet previous studies have largely overlooked Latin American samples. As such, this study addresses an important gap in knowledge.

Acral melanomas are typically diagnosed based on their anatomic location – the non-hair bearing skin of the foot soles, palms of hand, or nail bed. The main finding in this manuscript is that two subtypes of melanoma appear on these body sites – a type of melanoma with BRAF mutations, similar to cutaneous melanoma from the trunk, and a type of melanoma with more classical features of acral lentiginous melanoma. The authors propose that these melanomas arise from distinct sets of melanocytes. Many of these ideas have been proposed and partially shown by others. The strength of the present study is the unique population, in which participants displayed a range of genetic admixture – some with more European descent and some predominantly of AmerIndian descent. This unique cohort allowed the investigators to observe correlations between somatic mutation profiles and ancestry. For instance, the BRAF-mutant subtype of melanoma was predominantly found in participants of European descent. Overall, the manuscript cements the idea that melanomas on volar skin are heterogeneous, where people of European descent sometimes get UV-radiation-induced melanomas, akin to cutaneous melanoma, while in skin of color, the non-UV driven form of acral lentiginous melanoma dominates.

Several technical concerns were noted, which will, once addressed, improve reporting and hopefully strengthen/broaden the main conclusions.

R. We are very thankful to Reviewer 3 for their encouraging comments on our submission. We have addressed all their comments here below.

1. In Figure 1a, the authors listed only 59 samples out of 96, it would be informative to show all cases at least once in the main figure. What is the rationale for listing genes, such as ZNF793, KRTAP4-16, and CSN2? These genes have a very low frequency (e.g. 1%) of mutations and unknown roles in melanoma progression. Moreover, these genes were not mentioned anywhere in the paper.

R. Originally, the rationale was to show only samples that had mutations in the depicted genes, however, we have now added a full figure (with all samples) in **Supplementary Figure 2**. We had originally listed the top 21 genes as they were calculated by dNdScv, but it is true that we did not mention these genes further. We have now re-done this Figure in response to your and Reviewer 2's comments and we show only the top seven genes according to dNdScv (Those with evidence of being drivers).

2. What is the purity and ploidy of each sample? Although the manuscript mentioned that tumors with purity < 0.2 were excluded, in 8 tumors, less than 10 SNVs were detected, and we could not find CNA scores for these tumors. Could these tumors have low purity, thus affecting mutation/CNA calling? More broadly, is tumor purity associated with the number of SNVs across the study?

R. The Reviewer is correct, we excluded tumours with purity < 0.2 only from the copy number estimation, not the mutation calling. We have now provided an estimate of the purity and ploidy of each sample in **Supplementary Table 28**. Indeed, in this new iteration of the manuscript, we have re-done several analyses including copy number calling. Now, we have estimated purity with two tools, ASCAT and Sequenza, and those where they greatly disagreed (*i.e.*, ASCAT estimated a purity of 1 and Sequenza estimated < 0.15 , 10 samples, or where the ASCAT-estimated goodness-of-fit was below 0.95, one sample) have been excluded. These indeed had very low SNV counts. After our re-analysis, in which we retained ASCAT purity and ploidy estimates instead of Sequenza, we see samples from six different patients having < 10 mutations. Most of these have an estimated purity < 0.75 , but for a few, ASCAT estimated a purity of 1 (likely an overestimate due to the low number of mutations). However, these do carry copy number aberrations. Other studies have found a number of samples with low numbers of mutations as well^{2,53}. Additionally, we have stringently filtered our mutation calls to increase specificity while minimising false positives, a non-trivial endeavour given our samples were from FFPE tissue. Tumour purity is indeed associated with the number of SNVs across the study (Please see **Figure R7**).

3. It is unclear if the authors scrutinized copy number calls for focal events, such as homozygous deletions and amplifications. Focal copy number alterations are a cardinal feature of acral melanoma. Specific genes to focus upon are discussed below.

R. We thank the Reviewer for their comment. We scrutinised specific driver events as suggested by the Reviewer and have altered our sample classification as a result. We add the details below, after each question.

a. NF1 can be homozygously deleted in acral melanomas, which has the same effect as truncating mutations. Therefore, tumors with homozygous deletion of NF1 should also be considered in the NF1-mutated group. For some of the homozygous deletions, the deeply deleted regions can be rather narrow, affecting only a few exons. We recommend scrutinizing pre-segment level data to ensure that these alterations are not missed.

R. We thank the Reviewer for their comment, and we agree with their statement. Scrutiny of segmented data showed only two samples with homozygous deletions of *NF1*. These alterations have now been considered in the mutational status of the samples, with these now included in the *NF1*-mutated subgroup, and therefore included in all subsequent analyses (e.g., copy number and ancestry comparisons).

Additionally, we have scrutinised pre-segmented data as suggested and considered the following conditions to identify focal *NF1* deletions:

1. That the sample has a mean coverage higher than 15x,
2. That at least one full exon has a coverage of 0.

We chose this strategy due to the variable coverage among exons, which can make it difficult to know if a specific exon is deleted or just has lower coverage than other surrounding exons. In this way, we did identify one sample that carries a focal deletion of *NF1* exon 13 (**Figure R9**).

Figure R9. Coverage of *NF1* exon 13 (in the center) in sample PD41020d. A close up (top panel) and a broader region (bottom) are shown. Three samples per patient are shown, at the top is the normal sample, the middle window shows the primary tumour, and the bottom window shows the metastasis (which has a focal deletion).

This sample has now been classified as *NF1*-null, and we are very thankful for the Reviewer's suggestion as we were able to improve our sample classification.

b. What is the status of *TERT* (promoter mutations, amplification, etc.) in the cohort? This gene is a well-known melanoma driver gene but was not mentioned in this study.

R. We had not originally included the *TERT* promoter in our study as we had been unable to assess the mutational state due to this being an exome dataset, where the promoter was not sequenced. However, in response to this and the other Reviewers' comments, we attempted to amplify by capillary sequencing the *TERT* promoter in all samples, including the -124 and -146 positions. We were successful in assessing the *TERT* mutational state in 76 samples belonging to 64 patients. We found that six out of 64 patients carry the -124 mutation (9.3%) and two out of 59 patients for which the -146 position was successfully amplified carry a mutation in this position (3.4%). In total, this means that for this population, and as no sample carries both mutations, we estimate that 10.5% of patients have an activating *TERT* promoter mutation. This is in line with what has been described in other studies, specifically Wang et al, 2024¹ (~13.5%) and Newell, *et al* 2020² (10.3%). We also found that, whenever a *TERT* mutation is found in a sample, it is found in all samples belonging to the same patient, supporting Wang *et al*'s previous finding that these lesions arise early in acral melanoma evolution. This information is now included in the manuscript (lines 357-364 of the track changes manuscript, and **Supplementary Table 4**).

Additionally, after re-doing our copy number analyses with ASCAT instead of Sequenza, we now see a significant amplification peak containing *TERT* (**Figure 2a**), with 20 samples now showing *TERT* amplification, of which three also had carried *TERT* promoter mutations (PD41035, PD41896, PD51932). So, in summary, for all patients where we could measure either *TERT* promoter or copy number status, 61% of patients were *TERT* mutant, which is in line with what has been reported previously (Wang et al, 2024¹). This is now shown in **Supplementary Table 4**.

c. For *CDKN2A*, the authors mentioned that 66% of samples showed deletion. Are these all homozygous deletions?

R. We have looked at segmentation data and we detected only two samples that carried homozygous deletions in *CDKN2A*, according to our new analyses with ASCAT. The rest of the samples show heterozygous deletion of this gene. We now mention this in line 509-510 of the track-changes document.

4. When analyzing genetic subtypes of acral melanoma, it would be informative to separate BRAF V600E with other BRAF mutations as well as the different NRAS hotspot mutations (G12 and Q61 codons).

Mention the types of hotspot mutations in the text.

R. We now mention the types of *BRAF* mutations in the main text (line 350 of the track changes document). Out of the 12 patients with *BRAF* mutations, 11 carry a *BRAF*^{V600E} mutation and one carries a *BRAF*^{L597R} mutation. According to previous literature⁵⁸, this mutation is also *BRAF*-activating.

a. This is especially important in the setting of *BRAF*, where there are three classes of mutations, each associated with different subtypes of melanoma. In the present manuscript, on Page 7 line 209, the authors state “it has been previously postulated that *BRAF*-mutated acral melanomas might be more biologically like melanomas from non-acral sites than to other acral melanomas”. This is likely true for *BRAF* V600E mutation but not for other *BRAF* mutations.

R. We thank the Reviewer for this reflection, and we agree. However, only one of the *BRAF* mutations is not a *BRAF*^{V600E}, and this one (*BRAF*^{L597R}) has also been associated with activation⁵⁸. We have redone the ancestry and transcriptomic signature analyses changing the category of the sample with the *BRAF*^{L597R} mutation to the *BRAF*-WT, and results look similar for the ancestry analysis ($P=0.01$) and for the transcriptomic analysis, although for this one we lose statistical significance ($P=0.1$) perhaps due to the small sample size. Therefore, we have left this sample in the *BRAF*-mutated set for completeness, but we now clarify this fact in the Methods, line 1331 of the track changes manuscript).

5. It was recently reported that during the evolution of acral melanoma, MAPK pathway mutations can occur after tumor initiation. For the cases in which multiple tumor samples were available, did the authors observe heterogeneous/subclonal *NRAS*/*KIT*/*BRAF*/*NF1* mutations?

R. We thank the Reviewer for this suggestion, and we have now looked closer into the data from patients with multiple tumour samples (**Table R2**). Out of 22 patients for whom we sequenced at least two different samples (e.g., a primary and a metastasis), 13 have mutations in *NRAS*/*KIT*/*BRAF*/*NF1*. For *BRAF*, all four patients have mutations across all samples, and for *NRAS*, three out of four patients have a *NRAS* mutation in both the primary and lymph node metastasis. These data suggest that these mutations appear as an early event in tumour evolution. For patients with *KIT* mutations, about half of the times the mutation is found in the primary and lost in the metastasis. These data agree with Wang *et al*, which suggests that metastases are seeded before the emergence of these mutations (Figure 5, Wang *et al*)¹. We have now included this information in **Supplementary Table 6**.

6. The use of TMB (tumor mutation burden) is not consistent throughout the manuscript. It was used to indicate both “mutations per megabase” and total mutation counts (such as in Figure S4).

R. We have now homogenised the term to refer to TMB - using this term throughout the manuscript. We have specified in the now Supplementary Figure 6 that this is plotting total TMB.

7. The number of indels per tumor appears extraordinarily high, on par with the number of somatic SNVs. There is a concern that these calls include numerous false positives. In our experience with Strelka, additional filters are needed, beyond the calls provided directly by the software.

R. We thank the Reviewer for this important observation. Indeed, we agree, and in response to this comment we have re-done our indel calling with Pindel, which was the algorithm used by the primary analysis workflow in the PCAWG studies⁷. Our new results indeed are much more conservative and we have much fewer numbers of indels across samples (**Figure R10**).

Figure R10. Comparison of the number of indels detected with Strelka2 (our previous strategy, in orange) and Pindel (our current strategy, in blue). Two adjacent lines are shown by sample.

8. For the clustering in Figure 4, do tumor purities differ across different clusters?

R. Yes, they differ, with Cluster 1 tumours being less pure than the other clusters. According to a RNA deconvolution analysis (**Figure 4d-f** and **Supplementary Figure 13c**), we see infiltration by endothelial and CD4+ T cells, as well as CAFs. This cluster is associated to an earlier stage, with a better prognosis. We have now added a track annotating tumour purity to the cluster figure (**Figure 4a**, although this could only be estimated for samples that passed the copy number analysis as it was estimated at that step).

9. How was tumor ploidy estimated? And why was a cutoff of 3.6 used to distinguish genome doubling events?

R. Originally, we estimated tumour ploidy with Sequenza, but in response to the comments from other Reviewers we have now decided to use ASCAT, as it is more widely used. ASCAT provides a ploidy estimate for each sample, which we have used in this paper. Regarding genome doubling events, we originally had used the 3.6 cut-off based on previous literature⁵⁹ and after discussions with experienced colleagues. However, in this submission, we have

changed our strategy and we now estimate whole genome duplication events with the tool ASCAT, which gives this estimate as a boolean variable (yes/no).

Referee #3 (Remarks on code availability):

No concerns noted

We are very thankful to Reviewer 3 for their helpful comments.

References

1. Wang, M. *et al.* The genetic evolution of acral melanoma. *Nat Commun* **15**, 6146 (2024).
2. Newell, F. *et al.* Whole-genome sequencing of acral melanoma reveals genomic complexity and diversity. *Nat Commun* **11**, 5259 (2020).
3. Yeh, I. *et al.* Targeted Genomic Profiling of Acral Melanoma. *J. Natl. Cancer Inst.* **111**, 1068–1077 (2019).
4. Farshidfar, F. *et al.* Integrative molecular and clinical profiling of acral melanoma links focal amplification of 22q11.21 to metastasis. *Nat Commun* **13**, 898 (2022).
5. Birkeälv, S. *et al.* Mutually exclusive genetic interactions and gene essentiality shape the genomic landscape of primary melanoma. *J Pathol* **259**, 56–68 (2023).
6. Liu, H. *et al.* Integrative molecular and spatial analysis reveals evolutionary dynamics and tumor-immune interplay of in situ and invasive acral melanoma. *Cancer Cell* **42**, 1067-1085.e11 (2024).
7. Campbell, P. J. *et al.* Pan-cancer analysis of whole genomes. *Nature* **578**, 82–93 (2020).
8. Jones, D. *et al.* cgpCaVEManWrapper: Simple Execution of CaVEMan in Order to Detect Somatic Single Nucleotide Variants in NGS Data. *Current Protocols in Bioinformatics* **56**, 15.10.1-15.10.18 (2016).
9. Martincorena, I. *et al.* Universal Patterns of Selection in Cancer and Somatic Tissues. *Cell* **171**, 1029-1041.e21 (2017).
10. Long, G. V. *et al.* Prognostic and clinicopathologic associations of oncogenic BRAF in metastatic melanoma. *J Clin Oncol* **29**, 1239–1246 (2011).
11. Wu, X. *et al.* Mutations in BRAF codons 594 and 596 predict good prognosis in melanoma. *Oncol Lett* **14**, 3601–3605 (2017).

12. Houben, R. *et al.* Constitutive activation of the Ras-Raf signaling pathway in metastatic melanoma is associated with poor prognosis. *J Carcinog* **3**, 6 (2004).
13. Moreau, S. *et al.* Prognostic Value of BRAFV600 Mutations in Melanoma Patients After Resection of Metastatic Lymph Nodes. *Ann Surg Oncol* **19**, 4314–4321 (2012).
14. Ribas, A. & Flaherty, K. T. BRAF targeted therapy changes the treatment paradigm in melanoma. *Nat Rev Clin Oncol* **8**, 426–433 (2011).
15. Jakob, J. A. *et al.* NRAS Mutation Status is an Independent Prognostic Factor in Metastatic Melanoma. *Cancer* **118**, 4014–4023 (2012).
16. Thomas, N. E. *et al.* Association Between NRAS and BRAF Mutational Status and Melanoma-Specific Survival Among Patients With Higher-Risk Primary Melanoma. *JAMA Oncol* **1**, 359–368 (2015).
17. Bai, X. *et al.* MAPK Pathway and TERT Promoter Gene Mutation Pattern and Its Prognostic Value in Melanoma Patients: A Retrospective Study of 2,793 Cases. *Clin Cancer Res* **23**, 6120–6127 (2017).
18. Thielmann, C. M. *et al.* NF1-mutated melanomas reveal distinct clinical characteristics depending on tumor origin and respond favorably to immune-checkpoint inhibitors. *Eur J Cancer* **159**, 113–124 (2021).
19. Cirenajwis, H. *et al.* 1-mutated melanoma tumors harbor distinct clinical and biological characteristics. *Molecular Oncology* **11**, 438–451 (2017).
20. Landi, M. T. *et al.* Genome-wide association meta-analyses combining multiple risk phenotypes provide insights into the genetic architecture of cutaneous melanoma susceptibility. *Nat Genet* **52**, 494–504 (2020).
21. Ang, K. C. *et al.* Native American genetic ancestry and pigmentation allele contributions to skin color in a Caribbean population. *eLife* **12**, e77514 (2023).
22. Bonilla, C., Shriver, M. D., Parra, E. J., Jones, A. & Fernández, J. R. Ancestral proportions and their association with skin pigmentation and bone mineral density in Puerto Rican women from New York city. *Hum Genet* **115**, 57–68 (2004).
23. Krepler, C. *et al.* A Comprehensive Patient-Derived Xenograft Collection Representing the

- Heterogeneity of Melanoma. *Cell Rep* **21**, 1953–1967 (2017).
24. Wang, M., Banik, I., Shain, A. H., Yeh, I. & Bastian, B. C. Integrated genomic analyses of acral and mucosal melanomas nominate novel driver genes. *Genome Medicine* **14**, 65 (2022).
 25. Wong, K. *et al.* Cross-species genomic landscape comparison of human mucosal melanoma with canine oral and equine melanoma. *Nature Communications* **10**, 1–14 (2019).
 26. Hendricks, W. P. D. *et al.* Somatic inactivating PTPRJ mutations and dysregulated pathways identified in canine malignant melanoma by integrated comparative genomic analysis. *PLoS Genet* **14**, e1007589 (2018).
 27. Bailey, M. H. *et al.* Comprehensive Characterization of Cancer Driver Genes and Mutations. *Cell* **173**, 371-385.e18 (2018).
 28. Turner, J. A. *et al.* Expanding the landscape of oncogenic drivers and treatment options in acral and mucosal melanomas by targeted genomic profiling. *International Journal of Cancer* **155**, 1792–1807 (2024).
 29. Hu, H. *et al.* Retinal dehydrogenase 5 (RHD5) attenuates metastasis via regulating HIPPO/YAP signaling pathway in Hepatocellular Carcinoma. *Int J Med Sci* **17**, 1897–1908 (2020).
 30. LinlingWu, Zhang, W., Yang, C. & Chen, H. Bioinformatics analysis of the diagnostic significance and functions of RDH5 in breast cancer. <https://www.ejgo.net/articles/10.22514/ejgo.2023.054> (2023)
doi:10.22514/ejgo.2023.054.
 31. Jette, C. *et al.* The Tumor Suppressor Adenomatous Polyposis Coli and Caudal Related Homeodomain Protein Regulate Expression of Retinol Dehydrogenase L *. *Journal of Biological Chemistry* **279**, 34397–34405 (2004).
 32. Ellerhorst, J. A. *et al.* Clinical Correlates of NRAS and BRAF Mutations in Primary Human Melanoma. *Clinical Cancer Research* **17**, 229–235 (2011).
 33. Mutect2. *GATK* <https://gatk.broadinstitute.org/hc/en-us/articles/360037593851-Mutect2>

(2025).

34. Raine, K. M. *et al.* cgpPindel: Identifying Somatic Acquired Insertion and Deletion Events from Paired End Sequencing. *Current Protocols in Bioinformatics* **52**, 15.7.1-15.7.12 (2015).
35. Ye, K., Schulz, M. H., Long, Q., Apweiler, R. & Ning, Z. Pindel: a pattern growth approach to detect break points of large deletions and medium sized insertions from paired-end short reads. *Bioinformatics* **25**, 2865–2871 (2009).
36. Ross, E. M., Haase, K., Van Loo, P. & Markowitz, F. Allele-specific multi-sample copy number segmentation in ASCAT. *Bioinformatics* **37**, 1909–1911 (2021).
37. Franch-Expósito, S. *et al.* CNApp, a tool for the quantification of copy number alterations and integrative analysis revealing clinical implications. *eLife* **9**, e50267 (2020).
38. Belote, R. L. *et al.* Human melanocyte development and melanoma dedifferentiation at single-cell resolution. *Nat Cell Biol* **23**, 1035–1047 (2021).
39. Weiss, J. M. *et al.* Anatomic position determines oncogenic specificity in melanoma. *Nature* **604**, 354–361 (2022).
40. McNeal, A. S. *et al.* BRAFV600E induces reversible mitotic arrest in human melanocytes via microRNA-mediated suppression of AURKB. *Elife* **10**, e70385 (2021).
41. Hu, M., Coleman, S., Judson-Torres, R. L. & Tan, A. C. The classification of melanocytic gene signatures. *Pigment Cell Melanoma Res* **37**, 854–863 (2024).
42. Smith, A. J., Lambert, P. C. & Rutherford, M. J. Understanding the impact of sex and stage differences on melanoma cancer patient survival: a SEER-based study. *Br J Cancer* **124**, 671–677 (2021).
43. Di Carlo, V. *et al.* Sex differences in survival from melanoma of the skin: The role of age, anatomic location and stage at diagnosis: A CONCORD-3 study in 59 countries. *European Journal of Cancer* **217**, 115213 (2025).
44. Barault, L. *et al.* Mutations in the RAS-MAPK, PI(3)K (phosphatidylinositol-3-OH kinase) signaling network correlate with poor survival in a population-based series of colon cancers. *International Journal of Cancer* **122**, 2255–2259 (2008).

45. Wang, L., Wu, J., Dai, Z., Ji, S. & Jiang, R. Clinical characteristics and prognosis of acral lentiginous melanoma: a single-center series of 211 cases in China. *Int J Dermatology* **60**, 1504–1509 (2021).
46. Tas, F. & Erturk, K. Acral Lentiginous Melanoma Is Associated with Certain Poor Prognostic Histopathological Factors but May Not be Correlated with Nodal Involvement, Recurrence, and a Worse Survival. *Pathobiology* **85**, 227–231 (2018).
47. Hayward, N. K. *et al.* Whole-genome landscapes of major melanoma subtypes. *Nature* **545**, 175–180 (2017).
48. Hida, T., Kato, J., Idogawa, M., Tokino, T. & Uhara, H. Genomic landscape of cutaneous, acral, mucosal, and uveal melanoma in Japan: analysis of clinical comprehensive genomic profiling data. *Int J Clin Oncol* **29**, 1984–1998 (2024).
49. Dai, B. *et al.* Analysis of KIT expression and gene mutation in human acral melanoma: with a comparison between primary tumors and corresponding metastases/recurrences. *Human Pathology* **44**, 1472–1478 (2013).
50. Holman, B. N. *et al.* Clinical and molecular features of subungual melanomas are site-specific and distinct from acral melanomas. *Melanoma Research* **30**, 562–573 (2020).
51. Krauthammer, M. *et al.* Exome sequencing identifies recurrent somatic RAC1 mutations in melanoma. *Nat Genet* **44**, 1006–1014 (2012).
52. Farshidfar, F. *et al.* Integrative molecular and clinical profiling of acral melanoma links focal amplification of 22q11.21 to metastasis. *Nat Commun* **13**, 898 (2022).
53. Liu, H. *et al.* Integrative molecular and spatial analysis reveals evolutionary dynamics and tumor-immune interplay of in situ and invasive acral melanoma. *Cancer Cell* **42**, 1067–1085.e11 (2024).
54. Ahn, S.-Y., Bae, G.-E., Park, S.-Y. & Yeo, M.-K. Differences in the Clinical and Molecular Profiles of Subungual Melanoma and Acral Melanoma in Asian Patients. *Cancers* **15**, 4417 (2023).
55. Huang, K. *et al.* Comparative Analysis of Acral Melanoma in Chinese and Caucasian Patients. *Journal of Skin Cancer* **2020**, 1–8 (2020).

56. Kim, H.-J., Seo, J.-W., Roh, M.-S., Lee, J.-H. & Song, K.-H. Clinical features and prognosis of Asian patients with acral lentiginous melanoma who have nodal nevi in their sentinel lymph node biopsy specimen. *Journal of the American Academy of Dermatology* **79**, 706–713 (2018).
57. Dika, E. *et al.* BRAF, KIT, and NRAS Mutations of Acral Melanoma in White Patients. *American Journal of Clinical Pathology* **153**, 664–671 (2020).
58. Dahlman, K. B. *et al.* BRAF(L597) mutations in melanoma are associated with sensitivity to MEK inhibitors. *Cancer Discov* **2**, 791–797 (2012).
59. Prasad, K. *et al.* Whole-genome duplication shapes the aneuploidy landscape of human cancers. *Cancer Res* **82**, 1736–1752 (2022).

Dear Dr. Burgess and Reviewers,

We are very grateful for the time and consideration that you have put into re-reviewing our manuscript. We are excited with the Reviewers' assessment of our previous responses, and, in this Revised manuscript, we have performed additional analyses in response to the final comments from Reviewer 3. In summary, we ran an additional copy number caller (CNVkit), in order to re-assess homozygous deletions in *NF1* and *CDKN2A*, and have considered how removing samples with a small number of mutations would affect our conclusions. These analyses have strengthened our previous results and have been included in this Revised manuscript.

We have also followed the Editor's recommendations: We have shortened our manuscript (it is now 4,374 words) and have reviewed the 'Guide to Authors', ensuring that the revised manuscript complies with these guidelines.

Below we provide a point-by-point response to the Reviewers' comments. We hope that our manuscript is suitable to move forward toward publication.

Referees' comments:

Referee #1 (Remarks to the Author):

The authors answered all my questions and concerns!
The new analysis and data reinforced their findings and showed the importance of their work. Congratulations on the amazing work.

We are very grateful to Reviewer 1 for their assessment and their encouraging comments on our manuscript.

Referee #2 (Remarks to the Author):

The authors have made a strong and constructive effort to address all points raised in the initial review. Notably, they re-called their variants using an additional caller and re-ran their driver analysis. This resulted in substantial changes to the list of significantly mutated genes (SMGs). Rather than overlooking these discrepancies, the authors revised the manuscript accordingly and curated the driver list with greater care, incorporating both literature support and orthogonal evidence. Overall, the responses are thorough and transparent. The revisions strengthen the rigor and reliability of the findings included in this study, and I find the authors' approach appropriate and commendable. I have no further concerns. The authors should be congratulated for this tour de force study.

We are very grateful to Reviewer 2 for their thorough evaluation of our responses and their encouraging comments on our manuscript.

Referee #3 (Remarks to the Author):

We thank the authors for their revision, which was generally responsive to our critique. Overall, we believe the main conclusions of the manuscript sound, particularly that there are differences in the biological subtypes of melanoma on acral body sites associated with ancestry. However, there are a couple unresolved issues, as discussed below.

In our previous critique, we suggested that the authors inspect the pre-segmented data to identify focal deletions. In response, they manually scrutinized alignments in Integrative Genomics Viewer (IGV) – this is not exactly what we had in mind, and we do not believe this strategy is a reliable way to infer copy number. Please have a look at supplemental figure 10 of PMID: 35706047 for examples of focal deletions of NF1 in acral melanoma. As you will see, the figure shows bin level data (grey data points) as well as copy number segments (yellow lines). In some tumors, there was a focal deletion that the segmentation algorithm did not pick up, but when viewed at the bin level, and armed with the biological knowledge that these are bona fide tumor suppressors, it is reasonable to call them focally deleted. Focal deletions of CDKN2A, NF1, and other tumor suppressors are not uncommon in acral melanoma, likely due to the distinct mutagenic forces that operate on this tumor. While Basurto-Lozada et. al. did not use the same software as PMID 35706047 to infer copy number, hopefully, bin-level copy number estimates are available that they can scrutinize. We anticipate that the frequency of deep deletions affecting tumor suppressors will increase after such a review. Indeed, the authors cite the Newell and Wang studies as examples of other studies that find a similar frequency of homozygous CDKN2A deletions, however, we disagree with the statement because those studies actually found much higher frequencies of homozygous CDKN2A deletion (around 30-40%).

We thank the reviewers for their suggestion on how to scrutinize the pre-segmented data, and we apologise for not having understood what they meant by their comment on our previous revision. We have carefully examined Fig. S10 and the associated information in the referenced paper¹, and we have replicated this analysis using our data. Specifically, in addition to ASCAT, we have now also used CNVkit to obtain pre-segmented, bin level data, for all samples that passed QC for copy number estimations (n=60). First, we followed their strategy to estimate the expected \log_2 ratios for homozygous deletions for each sample based on ASCAT's estimation of ploidy and purity, using the following formula where CNseg=0 for homozygous deletions:

$$\text{(Formula 1) } \log_2 \text{ Ratio} = \log_2 \left(\frac{(1-p) \cdot 2 + p \cdot \text{CNseg}}{(1-p) \cdot 2 + p \cdot \varphi} \right)$$

Where p =tumour purity, CNseg represents the absolute copy number, and φ equals sample ploidy. Subsequently, we identified all bins covering *NF1* and *CDKN2A* that had a \log_2 ratio equal to or below the expected \log_2 ratio for a homozygous deletion.

For *NF1*, a large gene where there are 53 bins with an average length of 492bp, we first looked at the samples we had previously identified with homozygous deletions in the segmented data (two samples, PD40962a and PD51932a). For these samples, about 40% of bins had a \log_2 ratio below the threshold, including several contiguous bins (**Figure R1**). Therefore, to now consider homozygous focal deletions, we considered the case where 1) at least two

contiguous bins had \log_2 ratios at or below the calculated threshold for that sample, or 2) at least one full exon has read coverage equal to zero. Considering these criteria, we identified four samples with deletions in *NF1* (**Figures R1 and R2**, and **new Supplementary Figure 2a**): Three of these samples had already been identified in our study, two in the segmented data analysis (mentioned above) and another one during the previous revision, and newly identified sample PD40986d. We have now considered this sample as *NF1*-null and have re-done all related analyses, as well as replotted all relevant figures (**Figure 2**, **Supplementary Figures 5-8**).

It is worth mentioning that the rate of *NF1* homozygous deletions we find (4 out of 60 samples, ~7%) is not lower than the rates reported in other publications; so we believe we are not undercalling *NF1* deletions. For example, Newell *et al* (2020)² report only one *NF1* homozygous deletion in their dataset, out of 87 samples, and Sousa-Squiavinato *et al* (2025)³ report no samples with *NF1* homozygous deletions.

[REDACTION]

As for *CDKN2A*, this is a small gene with only five bins with an average length of 517bp. We first assessed pre-segmented data for the samples we had already identified as homozygously deleted in the analysis of segmented data with ASCAT (PD51969a and PD41044a). We noticed that while PD51969a showed four out of five bins at or below the calculated threshold, PD41044a had only one, but with a characteristic pattern where most of the other bins were close to the threshold, with a noticeable difference with respect to the bins around the gene (**Figure R3**). Therefore, we considered a sample as having a homozygous deletion if 1) It had at least one bin below the threshold, 2) at least two other bins were close to the threshold, and 3) there was a noticeable difference in \log_2 ratios for bins falling in *CDKN2A* in comparison with its neighbours. (We reassessed whether we would reclassify any samples for *NF1* following these criteria, but none apart from the ones previously identified fulfilled these requirements). This yielded ten samples with homozygous deletions in *CDKN2A* (**Figure R4**, and **new Supplementary Figure 2b**), an increase from our previous version of the manuscript and more in line with the rates of *CDKN2A* deletions in other acral melanoma studies. We have amended the text accordingly (lines 211-212 of the clean manuscript version).

We are thankful to the Reviewer for the attention they have paid to this important point and we believe that these additional analyses have strengthened our conclusions. We have included these analyses in the Methods and as **Supplementary Figure 2**.

Figure R1. Pre-segmented bin-level copy number data in the *NF1* region for all acral melanoma samples that passed QC. Each row represents one sample, with the left panel displaying the whole chromosome 17 and the right panel a close up view at the *NF1* region. Each point represents a bin as calculated by CNVkit, the x-axis represents chromosomal position and the y-axis the \log_2 ratio calculated with CNVkit. Red dots indicate bins falling in the *NF1* region. The purple dotted line represents the expected \log_2 ratio for a homozygous deletion calculated by the formula in (1). The orange lines represent segmented data. Highlighted in green are samples previously identified in our analysis (sample PD41020d has a point below the plotting range; this is the only sample with a bin not visible in the graphs above, and is depicted in full in **Figure R2**). Highlighted in dark pink, the new *NF1*-deleted sample identified by this analysis.

Figure R2. Pre-segmented bin-level copy number data in the *NF1* region for sample PD41020d. This is the only sample with a bin below the plotting range in Figure R1, and had been identified in our previous revision as having a deletion of a full exon by manual coverage evaluation.

Figure R3. Samples with *CDKN2A* homozygous deletions found previously by analysis of segmented data (ASCAT).

Figure R4. Pre-segmented bin-level copy number data in the *CDKN2A* region for all acral melanoma samples that passed QC. Each row represents one sample, with the left panel displaying the whole chromosome 9 and the right panel a close up view at the *CDKN2A* region. Each point represents a bin as calculated by CNVkit, the x-axis represents chromosomal position and the y-axis the \log_2 ratio calculated with CNVkit. Red dots indicate bins falling in the *CDKN2A* region. The purple dotted line represents the expected \log_2 ratio for a homozygous deletion calculated by the formula in (1). The orange lines represent segmented data. Highlighted in green are samples previously identified in our analysis, and highlighted in dark pink, the new *CDKN2A*-deleted samples identified by this analysis.

Thank you for confirming that tumor purity was lower in Cluster 1 tumors and updating a figure to show tumor purity. Do the authors believe this might explain why tumors with no driver mutations tend to have a better prognosis? It is more difficult to detect somatic mutations in tumors with low neoplastic cell content, which likely explains the association with good prognosis. We noticed that reviewer 2 also picked up on this association, referring to it as a “confounding variable”. In the point-by-point rebuttal, the authors seemed to agree with reviewer 2 that the “good prognosis” tumors are likely earlier stage with lower tumor cell content, yet these limitations are not sufficiently reflected in the manuscript itself. It is probably true that a tumor with fewer driver mutations will have better prognosis, but given all the technical challenges to detecting driver mutations in tumors with low neoplastic cell content, we advise the authors to tone down this conclusion.

We understand what the Reviewer is saying and after reflection with our team, we agree with the fact that we are unable to determine whether the tendency for a better prognosis seen for patients with QWT tumours is due to a true absence of driver mutations or to tumours being generally earlier stage, with driver mutations that are undetectable with our current methods. We have therefore removed this paragraph from the Discussion.

As an extension of the point above, we encourage the authors to revisit whether all tumors should be included in this study. As an extreme example, we saw one tumor (PD51948a) with no somatic point mutations, no somatic copy number alterations, and it was predicted to have a tumor cellularity of 100%. While acral melanomas have low mutation burdens, it seems implausible for a tumor to have no somatic alterations – not even some passenger mutations.

This is a really good point, which has sparked many discussions among the manuscript's authors. We agree that some tumours have very low numbers of mutations, but these have still been classified as acral melanomas following histopathological analysis by consultant dermatopathologists, and we believe that leaving them in the dataset captures the range of tumour presentation seen in samples in FFPE biobanks (As an example, see **Figure R5** corresponding to the highlighted sample PD51948a). We believe that the low numbers of mutations in these samples is due to our stringent variant calling (*i.e.*, running three algorithms and selecting those called by two or three tools) and copy number calling (*e.g.*, estimating ploidy and purity with two algorithms followed by extensive manual reassessment, as described in this revision). In support of this, we identified collections of somatic mutations identified by single variant calling tools (**Figure R6**) that were filtered from the final call set (as above), but clearly denote tumour associated events. We have now made available the somatic variants identified per caller in our GitHub for transparency and to facilitate further analysis. Furthermore, RNAseq data shows that *SOX10* and *MITF*, which are classical

melanoma markers, are detectable in samples with a low mutation burden (*i.e.*, three or fewer mutations detected), with expression levels equal to or higher than samples with more mutations (range 0-21 mutations for *MITF*, 0-46 for *SOX10*). These data support the idea that these samples contain acral melanoma, albeit with a low mutational load and lower tumour cellularity.

Figure R5. Histology for sample PD51948a, showing nests of malignant cells, some expressing melanin.

Figure R6. Examples of somatic variants identified by single tools in sample PD51948a (VarScan or mutect2 in this case). Screenshots taken from IGV.

As a further precaution, we have considered our results were we to remove samples that have low numbers of variants (*i.e.*, having three or fewer somatic mutations) from the full manuscript. For this exercise, we removed twelve samples and re-did the most important analyses in this manuscript. The analysis assessing the correlation of *BRAF* mutation with European ancestry remains significant (**Figure R7a**, P -value = 0.02), the transcriptome analysis quantifying the acral:cutaneous score ratio between *BRAF*-activating and *BRAF*-wt samples remains significant (**Figure R7b**, P -value = 0.041), and the clustering analysis reveals that all but one sample remained in their originally assigned clusters (**Figure R7c**). Therefore, were we to be wrong about the inclusion of these lowly-mutated samples, their removal would not alter the conclusions in this manuscript.

Figure R7. Main analyses in the manuscript without the lowly-mutated samples. a) European ancestry relationship with driver mutations. b) Acral:cutaneous (A:C) transcriptional ratio between samples with *BRAF*-activating mutations and those *BRAF*-wt. c) Unsupervised clustering analysis.

Minor points

We suggest altering the language when the authors say KIT mutations are “lost” in metastases. It is unlikely that the mutation was “lost”. A more plausible scenario is that cells seeded metastatic sites before the KIT mutation occurred and underwent a clonal sweep in the primary tumor. To be sure, the authors do suggest, later in the paragraph, that metastases are likely seeded early, but we would advise against implying reversion of the KIT mutation.

We agree, and we intended to convey this message when writing the original manuscript. We have now amended the language in line 189 of the clean manuscript version to reflect this point.

We do not believe a full paragraph in the discussion is needed to acknowledge limitations of studying FFPE tissues. It is true that formalin fixation fragments and damages DNA, and in theory, FFPE-induced damage can be misread as mutations in sequencing data. However, in practice, FFPE-induced artifacts tend to be randomly distributed across the genome and therefore have low allele frequency. When coverage and tumor cellularity are sufficiently high, FFPE-induced artifacts are rare. It is ok to acknowledge the challenges of studying FFPE material, but this could be moved to the methods and does not require precious real estate in the discussion section.

As advised, we have moved the paragraph discussing the challenges of calling mutations in FFPE to the Methods (lines 495-501 of the clean version manuscript).

We suggest removing the word “frequent” in this sentence from the abstract. “we found fewer frequent mutations in classical driver genes such as BRAF, NRAS or NF1.”

We have removed the word “frequent” from the abstract.

In the authors response, they said that they homogenized their use of tumor mutation burden, but there still is at least one example where the term is used inconsistently. In the text of the manuscript, the authors refer to tumor mutation burden as mutations per megabase, but in figure 2d, they appear to plot absolute mutation counts (unless there are some tumors with 120 mutations/Mb!). It is less important to be consistent, but most important to define the units whenever the term is used.

We apologise for the confusion, we have carefully read the manuscript for instances where we use the term tumour mutation burden (TMB), and have amended these as needed.

Referee #4 (Remarks to the Author):

I co-reviewed this manuscript with one of the reviewers who provided the listed reports.

We thank Reviewers 3 and 4 for their help in improving our manuscript.

References

1. Wang, M., Banik, I., Shain, A. H., Yeh, I. & Bastian, B. C. Integrated genomic analyses of acral and mucosal melanomas nominate novel driver genes. *Genome Medicine* **14**, 65 (2022).
2. Newell, F. *et al.* Whole-genome sequencing of acral melanoma reveals genomic complexity and diversity. *Nature Communications* **11**, 5259 (2020).
3. Sousa-Squiavinato, A. C. M. *et al.* Modelling Acral Melanoma in Admixed Brazilians Uncovers Genomic Drivers and Targetable Pathways. 2025.08.08.25332963 Preprint at <https://doi.org/10.1101/2025.08.08.25332963> (2025).
4. Wang, M. *et al.* The genetic evolution of acral melanoma. *Nat Commun* **15**, 6146 (2024).

Dear Dr. Burgess,

We are thrilled that our manuscript has been accepted in principle for publication in *Nature*. Here I attach a list of the final editorial points and our response.

Thank you again for the opportunity to work with you and the Reviewers.

We have included in this submission the following checklists:

1. Biology editorial checklist,
2. Manuscript checklist, and
3. Reporting summary.

We have addressed all comments in the reporting summary, and we have provided both the commented file and the new updated version for comparison. We believe we do not need to submit the third-party rights table and the code and software submission checklists, but please advise if otherwise. We have also included in this submission a Supplementary Table 1 file with rounded down ages and month/year dates to protect patient anonymity, but have provided a statement on how to access the full dataset in the Supplementary Information file.

Below we address the formatting issues:

1. Please add references to the abstract.

R. References have been added to the abstract.

2. The number of main text references should be 60 in total or less - currently there are 77.

R. Thank you, the total number included those of the main text. We have separated now those from the main text from those in the Methods, and these (Abstract + Main text) now number 46 references. We have created a separate reference list for the methods with continuous numbering.

3. Flagging that there are no methods references - please create a separate reference list for the methods with continuous numbering.

R. We have added a separate section within the same document with the methods references, with continuous numbering.

4. Please remove the main figures from the article file.

R. The main figures have been removed from the article file, and have only left the Figure legends.

5. Please reduce subheadings to 40 characters (with spaces) or less.

R. All subheaders have been reduced to 40 characters or less.

6. Please provide a supplementary information guide (That is, as table of contents listing what is in the supplementary information)

R. We have now created a new file with a guide of the supplementary figures and tables and their legends, as well as the actual supplementary figures. Where necessary, tables have been provided separately, as some of these are quite large and do not fit the page width or length.

7. Flagging that there are potential third party rights issues in the figures - please check sources or if permissions are needed for the flowchart illustrations in the figures. If so, please provide a filled-out third party rights table.

R. We have created all figures ourselves, we have not imported any figures or illustrations. For the flowchart in Supp. Fig. 5, we used Lucidchart, for which we purchased a license. Reviewing their Terms of Service (<https://lucid.co/tos>), in point 5, "Sharing features in the Services" it says:

5.1. Sharing Content. The Subscription Services allow you to share information and Content within your account, outside your account, and publicly. You are solely responsible for the Content that you create, transmit, display, or share with others while using the Subscription Services, and for the consequences of your actions.

5.2. Third-Party Content. The Services may contain content provided by third parties (e.g., templates) that is not owned by Lucid and that may be protected by intellectual property rights of those third parties. Such content is the sole responsibility of the person or entity from whom it originated, and you are responsible for your use of it.

We confirm we did not use any templates but only their default drawing tools, so we believe a third party rights table is not necessary (but please advise if this is not the case).

8. Please remove the Supplementary information from the article file and resupply them in a separate file.

R. We have removed the Supplementary information from the main file and have created a separate Supplementary Information file with the guide and the relevant figures and tables.

Editorial requests:

- As mentioned above, please ensure all data accessions are made live.

R. We have now requested and reviewed that all data accessions are live, these are EGAD00001015755 (whole-exome data) and EGAD00001015756 (RNAseq data).

<https://ega-archive.org/datasets/EGAD00001015755>

<https://ega-archive.org/datasets/EGAD00001015756>

- As indicated above, please separate your code and data availability statements into separate statements for 'Data availability' and 'Code availability'

R. We have separated the Data and Code availability statements now.

- Note that we can accommodate up to 10 'Extended Data' display items such as figures. These have greater visibility than SI figures because they are included in the electronic pdf when the article is downloaded. We recommend promoting up to 10 of the most-important SI figs as extended data figs, to enhance their visibility. Please see below for guidelines for SI vs ED display items. Please also ensure that any adjustments are made to the citations of these display items (and their numbering) to account for the changes.

R. Thank you, we have now promoted what we consider the 10 most important supplementary figures to Extended Data items. The main text has been amended to reflect this.

9. Flagging that there are no Extended data items present. (Please see editorial advice below)

R. We have now promoted 10 Supplementary Figures to Extended data items.

10. Please ensure that the text size in all figures is at least 5 pt Arial.

R. We have made sure that all text in the figures is legible.